# Stochastic Smoothed Primal-Dual Algorithms
# for Nonconvex Optimization with Linear Inequality Constraints

**Ruichuan Huang** [1]  **Jiawei Zhang** [* 2 3]  **Ahmet Alacaoglu** [* 1]

## Abstract

We propose smoothed primal-dual algorithms for solving stochastic nonconvex optimization problems with linear *inequality* constraints. Our algorithms are single-loop and only require a single (or two) samples of stochastic gradients at each iteration. A defining feature of our algorithm is that it is based on an inexact gradient descent framework for the Moreau envelope, where the gradient of the Moreau envelope is estimated using one step of a stochastic primal-dual (linearized) augmented Lagrangian algorithm. To handle inequality constraints and stochasticity, we combine the recently established global error bounds in constrained optimization with a Moreau envelope-based analysis of stochastic proximal algorithms. We establish the optimal (in their respective cases) $O(\varepsilon^{-4})$ and $O(\varepsilon^{-3})$ sample complexity guarantees for our algorithms and provide extensions to stochastic linear constraints. Unlike existing methods, iterations of our algorithms are free of subproblems, large batch sizes or increasing penalty parameters in their iterations and they use dual variable updates to ensure feasibility.

## 1. Introduction

We focus on the problem template

$$\min_{\mathbf{x} \in X} f(\mathbf{x}) \text{ subject to } A\mathbf{x} = \mathbf{b}, \qquad (1)$$

where $f \colon \mathbb{R}^n \to \mathbb{R}$ is $L_f$-smooth, the set $X \subseteq \mathbb{R}^n$ is polyhedral, and easy to project. In particular, let $X$ be given as $X = \{\mathbf{x} \colon H\mathbf{x} \le h\}$ for some matrix $H$ and vector $h$. Taking $H = I$, for example, gives this template the ability to model *linear inequality* constraints.

---
[*]Co-last authors [1]University of British Columbia [2]University of Wisconsin–Madison [3]MIT. Correspondence to: Jiawei Zhang <jzhang2924@wisc.edu>, Ahmet Alacaoglu <alacaoglu@math.ubc.ca>.

*Proceedings of the 42$^{nd}$ International Conference on Machine Learning*, Vancouver, Canada. PMLR 267, 2025. Copyright 2025 by the author(s).

In particular, when we have the problem

$$\min_{\mathbf{x} \in \mathbb{R}^n} f(\mathbf{x}) \text{ subject to } A\mathbf{x} \le \mathbf{b}, \qquad (2)$$

we introduce a slack variable $\mathbf{t} = A\mathbf{x} - \mathbf{b}$ so that $A\mathbf{x} - \mathbf{t} = \mathbf{b}$ and our optimization variable becomes $\binom{\mathbf{x}}{\mathbf{t}}$. Then, we can equivalently write the problem in the template (1) by using the constraint $\mathbf{t} \le 0$, where the set $X = \{\binom{\mathbf{x}}{\mathbf{t}} \colon \mathbf{x} \in \mathbb{R}^n, \mathbf{t} \le 0\}$ is easy to project. As such, we focus on (1) and our results directly apply to solving (2) by using this standard slack variable reformulation.

The assumption of $X$ being easy-to-project is without loss of generality. Indeed, when $X$ is not easy to project, we can add a slack variable for $H\mathbf{x} \le h$ similar to the above paragraph, to have a linear equality constrained problem with projectable constraints (cf. (1)). We refer to (Li et al., 2021, Remark 6), for the classical conversion of an $\varepsilon$-stationarity point of the problem with the slack variable to the original inequality constrained problem. Throughout, we assume that we have access to an unbiased oracle $F(\mathbf{x})$ such that

$$\mathbb{E}[F(\mathbf{x})] = \nabla f(\mathbf{x}), \text{ and } \mathbb{E}\|F(\mathbf{x}) - \nabla f(\mathbf{x})\|^2 \le \sigma^2. \quad (3)$$

A common setting is when $f(\mathbf{x}) = \mathbb{E}_{\xi \sim \Xi}[f(\mathbf{x}, \xi)]$ where $\Xi$ is an unknown distribution that we can draw i.i.d. samples from. In this case, it is common to set $F(\mathbf{x}) = \nabla f(\mathbf{x}, \xi)$ where $\mathbb{E}[\nabla f(\mathbf{x}, \xi)] = \nabla f(\mathbf{x})$. This will be our main focus.

Inclusion of the set $X$ in (1) increases the modeling power of (1) significantly, while causing difficulties in the analysis. Many problems fit this template, including constrained and distributed optimization, nonnegative matrix factorization, sparse subspace estimation and collaborative learning, see for example[1] (Zhang et al., 2022; Hong, 2016). Moreover, reformulations of nonconvex problems are also common by using linear inequality constraints (Zhang et al., 2022).

Algorithm development for (1) and related templates with global complexity guarantees, have been active in the last couple of years (Alacaoglu & Wright, 2024; Zhang & Luo, 2020; Zhang et al., 2020; Lu et al., 2024; Li et al., 2021; Lin et al., 2022; Yan & Xu, 2022; Li et al., 2024; Boob

---
[1]Details of some applications are given in Sec. 6 of our extended version: https://arxiv.org/abs/2504.07607

et al., 2023; Hong, 2016), mainly due to the new applications of functionally constrained nonconvex optimization problems in the context of neural network training (Katz-Samuels et al., 2022; Dener et al., 2020). In these applications with problems involving nonconvex functional constraints, stochastic augmented Lagrangian methods (ALM) have found widespread use, whereas their behavior for even *linearly constrained* nonconvex optimization of the form (1) remain poorly understood. Our focus is to improve our understanding of stochastic ALM in the context of nonconvex optimization, by focusing on the fundamental template (1).

Compared to the setting of convex $f$, where the global complexity analysis is mature for ALM and its stochastic version (Yan & Xu, 2022), nonconvexity of $f$ poses significant difficulties in the analysis of ALM. Many works in the literature focus on penalty based algorithms (which will be formally introduced later in this section) that do not perform dual updates (or perform *negligible* dual updates that we clarify later) (Lu et al., 2024; Li et al., 2021; Lin et al., 2022), rather than primal-dual algorithms such as ALM. However, in practice, dual updates are known to be essential for accelerating convergence. Penalty methods are known to be unstable since increasing penalty parameter causes Lipschitz constant of the subproblems to increase and can lead to numerical issues. These differences in behavior between penalty and augmented Lagrangian methods are well-known, see, for example, the classical books (Bertsekas, 2014, Sec. 2.2.5) (Nocedal & Wright, 1999, Sec. 17.5).

For problem (1) with access to full gradients of $f$ and the full matrix $A$, optimal complexity with primal-dual methods are obtained in the work of Zhang & Luo (2022). When one has access to stochastic gradients of $f$ and the matrix $A$, a recent work by Alacaoglu & Wright (2024) showed optimal complexity guarantees under expected smoothness (see Assumption 5.2), for the special case of (1) when $X = \mathbb{R}^n$. However, this latter restriction significantly reduces the generality of the template. For example, modeling standard quadratic programming requires $X$ to be a half-space, which was not supported in the analysis of Alacaoglu & Wright (2024). Our goal is to go beyond these results by handling both the case when $X \neq \mathbb{R}^n$ as well as the case when we do not have access to the matrix $A$ but only to an unbiased estimate of $A$, by keeping optimal complexity guarantees. A more detailed comparison of complexity guarantees will be made in Section 6 and a summary is provided in Table 1.

**Lagrangian, penalty and augmented Lagrangian.** The standard approach to tackle (1) is to design algorithms operating on the Lagrangian, augmented Lagrangian or penalty functions. In particular, the Lagrangian function is given as

$$L(\mathbf{x}, \mathbf{y}) = f(\mathbf{x}) + \langle A\mathbf{x} - \mathbf{b}, \mathbf{y} \rangle,$$

with the dual variables $\mathbf{y}$, whereas the penalty function (or

more precisely the quadratic penalty (QP)) has the form of

$$\text{Pen}_\rho(\mathbf{x}) = f(\mathbf{x}) + \frac{\rho}{2}\|A\mathbf{x} - \mathbf{b}\|^2.$$

It is common for algorithms based on the penalty function to require $\rho \to \infty$ for convergence (Bertsekas, 2014, Sec. 2.2.5). One major disadvantage of this strategy is that $\rho$ getting larger makes the subproblem of minimizing the penalty function more and more ill-conditioned (cf. (4)).

An influential idea was the introduction of the augmented Lagrangian (AL) function which combined the idea of the Lagrangian and penalty formulations (Hestenes, 1969; Powell, 1969). In particular, the AL function is defined as

$$L_\rho(\mathbf{x}, \mathbf{y}) = f(\mathbf{x}) + \langle A\mathbf{x} - \mathbf{b}, \mathbf{y} \rangle + \frac{\rho}{2}\|A\mathbf{x} - \mathbf{b}\|^2.$$

Augmented Lagrangian methods in the classical literature were favoured because they did not require $\rho$ to grow arbitrarily large. In fact, many instances of ALM converge to the optimal solution with fixed $\rho$ since the incorporation of the dual variable updates aids in satisfying feasibility (Bertsekas, 2014, Prop. 2.4, Prop. 2.6).

**Primal vs primal-dual algorithms.** The algorithms based on the penalty function are generally referred to as *penalty algorithms* and are easier to analyze in different settings since they are primal-only algorithms, meaning that they only perform updates on primal variable $\mathbf{x}$ where approximate feasibility is ensured by $\rho \to \infty$. In particular, a classical penalty method iterates for $k = 1, 2, \ldots$ as

$$\begin{aligned}
&\mathbf{x}_{k+1} \approx \arg\min_{\mathbf{x} \in X} f(\mathbf{x}) + \frac{\rho_k}{2}\|A\mathbf{x} - \mathbf{b}\|^2, \\
&\text{Select } \rho_{k+1} > \rho_k.
\end{aligned} \tag{4}$$

The algorithms based on the AL function are generally more difficult to analyze due to the additional dynamics coming from the dual updates which are critical to ensure that the approximate feasibility is attained with constant $\rho$. An ALM iteration proceeds for $k = 1, 2, \ldots$ by updating

$$\mathbf{x}_{k+1} \approx \arg\min_{\mathbf{x} \in X} f(\mathbf{x}) + \langle \mathbf{y}_k, A\mathbf{x} - \mathbf{b} \rangle + \frac{\rho}{2}\|A\mathbf{x} - \mathbf{b}\|^2,$$

$$\mathbf{y}_{k+1} = \mathbf{y}_k + \sigma(A\mathbf{x}_{k+1} - \mathbf{b}).$$

For penalty methods and ALM, different strategies exist to generate $\mathbf{x}_{k+1}$ that approximately minimize the penalty or augmented Lagrangian functions by either iterating multiple steps of gradient descent (GD), known as *inexact* algorithms, or applying one step of GD, known as *linearized* algorithms (Ouyang et al., 2015).

In view of the earlier discussion, when $f$ is nonconvex, most of the literature focuses on either analyzing penalty methods, or analyzing ALM with *negligible* dual updates and increasing penalty parameters $\rho$, due to the inherent

difficulty in analyzing the dual variable and its effect in convergence. In particular, as also highlighted in (Alacaoglu & Wright, 2024), many of the recent analysis of ALM is of the form of a *perturbed penalty analysis*, meaning that the feasibility is driven by increasing penalty parameters, and the dual updates are designed so that they do not deteriorate the estimates too much. Because of this, the dual step sizes are selected to be small to ensure boundedness of the dual variable (or controlling the growth of the dual variable). We refer to such updates as *negligible* dual updates since the analyses do not harness the benefit of such updates in ensuring feasibility. Feasibility is driven by large penalty parameters. Some representative examples are (Lu et al., 2024), (Li et al., 2021), (Lin et al., 2022), (Li et al., 2024).

This is the case even in the deterministic setting and the only method that we are aware that can handle true ALM with fixed penalty parameters and non-negligible dual updates are due to (Zhang & Luo, 2022) that uses a linearized *proximal* AL function with a dynamic adjustment on the proximal center, which will be clarified in Section 2 since it will form the basis of our algorithmic development.

### 1.1. Contributions

In this paper, we propose a stochastic smoothed linearized ALM for solving (1) that only uses a single sample of stochastic gradient at every iteration. This algorithm also works with a constant penalty parameter and incorporates non-negligible dual updates for feasibility where the dual step sizes have the same order as the primal step sizes. We show that this method has its iteration complexity and sample complexity guarantees in the order of $O(\varepsilon^{-4})$. Such a sample complexity result is optimal even in the unconstrained nonconvex case under our assumptions (see Assumption 1.1) (Arjevani et al., 2023). In contrast, the prior results with optimal complexity required large penalty parameters, no dual updates and further assumptions (Lu et al., 2024). We then prove that this complexity can be improved to $O(\varepsilon^{-3})$ with variance reduction when an additional expected smoothness assumption is made (see Assumption 5.2). Under this stronger assumption, this is the optimal complexity even without constraints (Arjevani et al., 2023).

We consider extensions of this framework when we have linear constraints that hold in expectation, that is, when the constraints are given as $\mathbb{E}_\xi[A_\xi \mathbf{x} - \mathbf{b}_\xi] = 0$, with the same complexity guarantees. To our knowledge, this is the first algorithm achieving the optimal $O(\varepsilon^{-4})$ benchmark sample complexity for nonconvex optimization with stochastic constraints using one sample per iteration, going beyond the best-known $O(\varepsilon^{-5})$ complexity that is achieved for a more general problem that does not capture the structure of linear constraints (Li et al., 2024; Alacaoglu & Wright, 2024).

A more detailed comparison with the related works is given

in Section 6. A summary is given in Table 1.

### 1.2. Preliminaries

We denote the indicator function of a convex closed set $X$ as $I_X(\mathbf{z}) = 0$ if $\mathbf{x} \in X$ and $I_X(\mathbf{x}) = \infty$ if $\mathbf{x} \notin X$. The notation $\partial f$ for a convex, closed function denotes the subdifferential set and $\partial I_X(\mathbf{x})$ is the normal cone of $X$ at $\mathbf{x}$, by definition. For matrix $A$, $\|A\|$ denotes its operator norm.

Given closed and convex $X$, projection onto $X$ is given as

$$\mathrm{proj}_X(\mathbf{x}) = \arg\min_{\mathbf{v} \in X} \|\mathbf{x} - \mathbf{v}\|^2.$$

Similarly, we define the proximal operator of $f$ as

$$\mathrm{prox}_f(\mathbf{x}) = \arg\min_{\mathbf{v}} f(\mathbf{v}) + \frac{1}{2}\|\mathbf{v} - \mathbf{x}\|^2.$$

We say that $f$ is $L$-smooth when its gradient is $L$-Lipschitz:

$$\|\nabla f(\mathbf{x}) - \nabla f(\mathbf{y})\| \le L\|\mathbf{x} - \mathbf{y}\|.$$

We say that $f$ is $\rho$-weakly convex when $f + \frac{\rho}{2}\|\cdot\|^2$ is convex. An $L$-smooth function is automatically $L$-weakly convex. Moreau envelope of the weakly convex $f$ is defined as

$$\varphi_\lambda(\mathbf{z}) = \min_{\mathbf{v}} f(\mathbf{v}) + \frac{1}{2\lambda}\|\mathbf{v} - \mathbf{z}\|^2,$$

which can be interpreted as a notion of *smoothing*. Moreau envelope has many useful properties such as being smooth when $f$ is nonsmooth and weakly convex, when $\lambda$ is selected accordingly. Moreover, stationary points of $f$ and the Moreau envelope coincide (Drusvyatskiy & Paquette, 2019, Lemma 4.3). The gradient of the Moreau envelope can be computed as

$$\lambda^{-1}(\mathbf{x} - \mathrm{prox}_{\lambda\varphi}(\mathbf{x})).$$

**Stationary points.** A succinct way of characterizing a stationary point of (1) is the following: $\mathbf{x}^\star$ is a stationary point if there exists $\mathbf{y}^\star$ such that the following hold:

$$0 \in \nabla f(\mathbf{x}^\star) + A^\top \mathbf{y}^\star + \partial I_X(\mathbf{x}^\star) \text{ and } 0 = A\mathbf{x}^\star - \mathbf{b}.$$

One may, for example, refer to (Rockafellar, 2000). Accordingly, we say that $(\mathbf{x}, \mathbf{y})$ is $\varepsilon$-**stationary** if

$$\|A\mathbf{x} - \mathbf{b}\| \le \varepsilon \text{ and}$$
$$\|\mathbf{v}\| \le \varepsilon \text{ where } \mathbf{v} \in \nabla f(\mathbf{x}) + A^\top \mathbf{y} + \partial I_X(\mathbf{x}) \quad (5)$$

which is a common notion used in related works, for example (Zhang & Luo, 2022).

We also use the following related notion of near-stationarity, as used in (Davis & Drusvyatskiy, 2019). We say that $\mathbf{x}$ is $\varepsilon$-**near stationary** if it satisfies

$$\|\nabla \Psi(\mathbf{x})\| \le \varepsilon, \quad (6)$$

where $\Psi(\mathbf{x})$ is the Moreau envelope of the objective function $f(\mathbf{x}) + I_X(\mathbf{x}) + I_{\{\mathbf{v}: A\mathbf{v} = \mathbf{b}\}}(\mathbf{x})$ in (1), see also (7). We refer to (Davis & Drusvyatskiy, 2019) for the precise notion of near stationarity.

| Reference | Constraint | Oracle | Complexity | Loops | Method |
|---|---|---|---|---|---|
| (Alacaoglu & Wright, 2024) | $A\mathbf{x} = \mathbf{b}$ | Eq. (3) and Asmp. 5.2 | $\widetilde{O}(\varepsilon^{-3})$ | 1 | ALM |
| (Alacaoglu & Wright, 2024) | $\mathbb{E}[c(\mathbf{x}, \zeta)] = 0$, and $\mathbf{x} \in X^{\dagger}$ | Eq. (3) and Asmp. 5.2 | $\widetilde{O}(\varepsilon^{-5})$ | 1 | Penalty |
| (Lu et al., 2024) | $c(\mathbf{x}) = 0$, and $\mathbf{x} \in X^{\dagger}$ | Eq. (3) and Asmp. 5.2 | $O(\varepsilon^{-3})$ | 1 | Penalty |
| (Li et al., 2024) | $\mathbb{E}[c(\mathbf{x}, \zeta)] = 0$, and $\mathbf{x} \in X^{\dagger}$ | Eq. (3) and Asmp. 5.2 | $O(\varepsilon^{-5})$ | 2 | Penalty[*] |
| This work | $A\mathbf{x} = \mathbf{b}$, and $\mathbf{x} \in X$ is a polyhedral | Eq. (3) | $O(\varepsilon^{-4})$ | 1 | ALM |
| This work | $\mathbb{E}_{\zeta}[A(\zeta)\mathbf{x} - \mathbf{b}(\zeta)] = 0$, and $\mathbf{x} \in X$ is a polyhedral | Eq. (3) | $O(\varepsilon^{-4})$ | 1 | ALM |
| This work | $A\mathbf{x} = \mathbf{b}$, and $\mathbf{x} \in X$ is a polyhedral | Eq. (3) and Asmp. 5.2 | $O(\varepsilon^{-3})$ | 1 | ALM |

*Table 1.* Comparison of methods. [*]This method is referred to as a penalty method because the penalty parameter is taken to infinity to ensure feasibility and dual updates do not contribute in achieving feasibility. [†]The set $X$ is assumed to have an efficient projection.

### 1.3. Assumptions

We next state the assumptions that will be used throughout. These assumptions are standard and to our knowledge, the weakest, in the literature for both deterministic and stochastic nonconvex problems with linear constraints (Zhang & Luo, 2022; Alacaoglu & Wright, 2024). A more detailed comparison of assumptions will be made in Section 6.

**Assumption 1.1.** For the problem (1), the following holds:

1. The function $f$ is $L_f$-smooth and lower bounded over the feasible set: $f(\mathbf{x}) \geq \underline{f} > -\infty$ for any $\mathbf{x} \in X$ and $A\mathbf{x} = \mathbf{b}$.

2. The set $X$ admits an efficient projection and is polyhedral. That is, it has the form $X = \{\mathbf{x} \colon H\mathbf{x} \leq h\}$ for some $H, h$.

3. We have access to stochastic gradients satisfying (3).

## 2. Algorithm

We introduce Algorithm 1 in this section. To gain a deeper understanding of the algorithm, we will go over two different ways of interpreting it.

**Interpretation 1: Linearized proximal ALM.** Algorithm 1 incorporates a single-step SGD approximation of the proximal AL function. This strategy is also known as the linearized proximal ALM. In particular, the first step of the algorithm approximates the proximal AL function[2], that is,

$$\mathbf{x}_{t+1} \approx \arg\min_{\mathbf{x} \in X} L_\rho(\mathbf{x}, \mathbf{y}_{t+1}) + \frac{\mu}{2}\|\mathbf{x} - \mathbf{z}_t\|^2,$$

by a single step of projected SGD, followed by a dual variable update and updating the proximal center $\mathbf{z}_t$, which

takes average of $\mathbf{z}_t$ and $\mathbf{x}_t$, resulting in the terminology *smoothed* that we use for the algorithm.

**Interpretation 2: Inexact GD on the Moreau envelope.**[3] Algorithm 1 can also be interpreted as an inexact gradient descent step on the Moreau envelope of the function in (1). In particular, this Moreau envelope is given as

$$\Psi(\mathbf{z}_t) = \min_{\mathbf{x} \in X, A\mathbf{x} = \mathbf{b}} \left\{ f(\mathbf{x}) + \frac{\mu}{2}\|\mathbf{x} - \mathbf{z}_t\|^2 \right\}. \quad (7)$$

By observing that minimizing the Moreau envelope helps on obtaining a near-stationary point in view of (6) (cf. (Davis & Drusvyatskiy, 2019)), inexact gradient update on this function requires the computation of

$$\arg\min_{\mathbf{x} \in X, A\mathbf{x} = \mathbf{b}} \left\{ f(\mathbf{x}) + \frac{\mu}{2}\|\mathbf{x} - \mathbf{z}_t\|^2 \right\},$$

which is a nontrivial optimization subproblem. However, it is easier than (1) because the regularization provides us a *strongly convex objective* in the subproblem (given that $\lambda$ is larger than $L_f$). As a result, we can approximate the solution of this problem by applying one iteration of ALM since this problem is a strongly convex optimization problem over linear constraints. We show that just one step of stochastic ALM is sufficient at every iteration by using a stochastic gradient computed with a single sample and one dual update, followed by the update of the proximal center $\mathbf{z}_t$.

On the surface, this algorithm strongly resembles that of Zhang & Luo (2022), from which we draw many ideas. However, in addition to using stochastic gradients, there is another subtle change, on the update of $\mathbf{z}_{t+1}$. Unlike (Zhang & Luo, 2022), we update $\mathbf{z}_{t+1}$ by using $\mathbf{x}_t$ to be

---

[2]Note that this is also a classical function (Rockafellar, 1976).

[3]Let us note that Hu et al. (2024) used a similar idea in a different context.

able to continue the analysis with the bounded variance assumption on $G$ (cf. Algorithm 1) instead of boundedness assumption on $G$, since the latter would require bounded domains. Thanks to this small change in this section, we handle the case with unbounded primal and dual domains.

# 3. Convergence Analysis

In this section, we first provide the main complexity results, then introduce the main analysis tools and a proof sketch.

## 3.1. Main Theorem

In view of the two stationarity notions given in Section 1.2, we start with the result showing that Algorithm 1 outputs a point at which the norm of the gradient of Moreau envelope is small, in expectation.

For the result, we state the algorithmic parameters. To avoid clutter, we write the orders of the parameters by highlighting their dependences on the problem parameters. The explicit forms of the parameters are given in (25), in App. A.

$$\tau \asymp \frac{1}{\sqrt{T}}, \quad \eta \asymp \frac{1}{\sqrt{T}}, \quad \beta \asymp \frac{1}{\sqrt{T}},$$
$$\mu \asymp L_f, \quad \lambda \asymp L_f + \mu(\|A\|^2 + 1). \tag{8}$$

We are now ready to state the first main result.

**Theorem 3.1.** *Let Assumption 1.1 hold and run Alg. 1 with parameters from (8). We have that $\mathbb{E}\|\nabla\Psi(\mathbf{z}_{t^*})\| \le \varepsilon$ where $t^*$ is selected uniformly at random from $\{0,\dots,T-1\}$ with $T = \Omega(\varepsilon^{-4})$. The stochastic oracle complexity is $O(\varepsilon^{-4})$.*

In particular, the above result gives us an $\varepsilon$-near stationary point in view of (Davis & Drusvyatskiy, 2019). To get an $\varepsilon$-stationary point, we perform a post-processing procedure to obtain the following output from the result of Alg. 1:

$$\hat{\mathbf{x}} = \text{proj}_X(\mathbf{x}_{t^*} - \tau\hat{G}(\mathbf{x}_{t^*}, \mathbf{y}_{t^*+1}, \mathbf{z}_{t^*})), \tag{9}$$

with $\tau \le \frac{1}{L_K}$ where $L_K$ is the Lipschitz constant of $L_\rho(\cdot, \mathbf{y}, \mathbf{z}) + \frac{\lambda}{2}\|\cdot -\mathbf{x}\|^2$ (cf. (25)) and

$$\hat{G}(\mathbf{x}_{t^*}, \mathbf{y}_{t^*+1}, \mathbf{z}_{t^*}) = \frac{1}{B}\sum_{i=1}^{B} G(\mathbf{x}_{t^*}, \mathbf{y}_{t^*+1}, \mathbf{z}_{t^*}, \xi_i)$$

for $\xi_i$ i.i.d. and $B = \Theta(\varepsilon^{-2})$. This is the only place where we use a large batch size and Algorithm 1 only runs with a single sample at every iteration. This post processing step is only done once and does not affect the overall complexity. The details are given in Appendix A.3.

**Corollary 3.2.** *Let Assumption 1.1 hold. From the output of Algorithm 1, we can obtain $\hat{\mathbf{x}}$ which is an $\varepsilon$-stationary point. The complexity of the whole procedure is $O(\varepsilon^{-4})$.*

## 3.2. Analysis Tools

In our analysis, Moreau envelope of two functions is critical. The first was the Moreau envelope of the composite objective in (1), defined in (7). We next define the Moreau envelope on the proximal AL which is the main function to analyze projected SGD, cf. (Davis & Drusvyatskiy, 2019)

$$\varphi_{1/\lambda}(\mathbf{x}, \mathbf{y}, \mathbf{z}) = \min_{\mathbf{u} \in X} \left\{ L_\rho(\mathbf{u}, \mathbf{y}) + \frac{\mu}{2}\|\mathbf{u} - \mathbf{z}\|^2 \right.$$
$$\left. + \frac{\lambda}{2}\|\mathbf{u} - \mathbf{x}\|^2 \right\}. \tag{10}$$

Another important quantity that has a significant role in the analysis is the proximal point

$$\mathbf{u}^*(\mathbf{x}, \mathbf{y}, \mathbf{z}) = \arg\min_{\mathbf{u} \in X} L_\rho(\mathbf{u}, \mathbf{y}) + \frac{\mu}{2}\|\mathbf{u} - \mathbf{z}\|^2$$
$$+ \frac{\lambda}{2}\|\mathbf{u} - \mathbf{x}\|^2. \tag{11}$$

With this, we trivially have

$$\varphi_{1/\lambda}(\mathbf{x}, \mathbf{y}, \mathbf{z}) = L_\rho(\mathbf{u}^*(\mathbf{x}, \mathbf{y}, \mathbf{z}), \mathbf{y})$$
$$+ \frac{\mu}{2}\|\mathbf{u}^*(\mathbf{x}, \mathbf{y}, \mathbf{z}) - \mathbf{z}\|^2 + \frac{\lambda}{2}\|\mathbf{u}^*(\mathbf{x}, \mathbf{y}, \mathbf{z}) - \mathbf{x}\|^2.$$

This is the main point of departure from (Zhang & Luo, 2022) where the proximal AL function is used in the analysis, in the potential function. This is because (Zhang & Luo, 2022) used a projected *full* GD step on the proximal AL function for which, a descent inequality follows directly. In our case, because we apply a projected SGD step, to be able to handle updates with single-sample stochastic gradients, we need to use the Moreau envelope of the proximal AL function in our potential. This analysis of projected SGD was pioneered in (Davis & Drusvyatskiy, 2019).

The first result is a descent result on the Moreau envelope.

**Lemma 3.3** (cf. Lemma A.5). *Under Assumption 1.1, for the $\mathbf{x}_{t+1}$ update given in Algorithm 1, we have*

$$16\mathbb{E}\left[\varphi_{1/\lambda}(\mathbf{x}_{t+1}, \mathbf{y}_{t+1}, \mathbf{z}_{t+1})\right]$$
$$\le 16\mathbb{E}\left[\varphi_{1/\lambda}(\mathbf{x}_t, \mathbf{y}_{t+1}, \mathbf{z}_{t+1})\right]$$
$$- \tau\lambda^2\mathbb{E}\|\mathbf{u}^*(\mathbf{x}_t, \mathbf{y}_{t+1}, \mathbf{z}_t) - \mathbf{x}_t\|^2 + 8\lambda\tau^2\sigma^2$$
$$+ 2\left(4\lambda\tau\mu + 16\lambda\tau^2\mu^2 + \tau\lambda^2\mu^2/\gamma_s^2\right)\mathbb{E}\|\mathbf{z}_t - \mathbf{z}_{t+1}\|^2,$$

*where $\gamma_s = 2\mu + \rho\|A\|$.*

This follows mostly from (Davis & Drusvyatskiy, 2019) and handles the transition from $\mathbf{x}_t$ to $\mathbf{x}_{t+1}$ in our analysis. One additional error term we have here is $\|\mathbf{z}_{t+1} - \mathbf{z}_t\|^2$, due to the change in the proximal center $\mathbf{z}_t$, a term that was not involved in the analysis of (Davis & Drusvyatskiy, 2019).

Next, we incorporate the dynamics of the updates on the dual variable $\mathbf{y}_t$ and the proximal center $\mathbf{z}_t$. These results use some ideas from (Zhang & Luo, 2022) with additional insights. This is because Zhang & Luo (2022) use

---

**Algorithm 1** Stochastic smoothed and linearized ALM

---

**Initialize:** $\mathbf{x}_0 = \mathbf{z}_0 \in X$, $\mathbf{y}_0 \in \mathbb{R}^m$ and $\rho \geq 0$.
**for** $t = 0$ **to** $T - 1$ **do**
$\qquad \mathbf{y}_{t+1} = \mathbf{y}_t + \eta(A\mathbf{x}_t - \mathbf{b})$
$\qquad$ Sample $\xi_t \in \Xi$ i.i.d. and let $G(\mathbf{x}_t, \mathbf{y}_{t+1}, \mathbf{z}_t, \xi_t) = \nabla f(\mathbf{x}_t, \xi_t) + A^\top \mathbf{y}_{t+1} + \rho A^\top (A\mathbf{x}_t - \mathbf{b}) + \mu(\mathbf{x}_t - \mathbf{z}_t)$.
$\qquad \mathbf{x}_{t+1} = \mathrm{proj}_X(\mathbf{x}_t - \tau G(\mathbf{x}_t, \mathbf{y}_{t+1}, \mathbf{z}_t, \xi_t))$
$\qquad \mathbf{z}_{t+1} = \mathbf{z}_t + \beta(\mathbf{x}_t - \mathbf{z}_t)$

---

$L_\rho(\mathbf{x}, \mathbf{y}) + \frac{\lambda}{2}\|\mathbf{x} - \mathbf{z}\|^2$ in their potential, so their analysis only characterizes the change in $\mathbf{y}$ and $\mathbf{z}$ in this function. Our analysis however, needs to characterize this change in the Moreau envelope of this function. This requires further estimations using the properties of the Moreau envelope, and the proximal point $\mathbf{u}^*(\mathbf{x}, \mathbf{y}, \mathbf{z})$ (see e.g. Lem. A.6).

**Lemma 3.4.** *(cf. Lemma A.6) Under Assumption 1.1, for the iterates of Alg. 1, we have*

$$2\mathbb{E}\left[\varphi_{1/\lambda}(\mathbf{x}_t, \mathbf{y}_{t+1}, \mathbf{z}_{t+1})\right]$$
$$\leq 2\mathbb{E}\left[\varphi_{1/\lambda}(\mathbf{x}_t, \mathbf{y}_t, \mathbf{z}_t)\right]$$
$$\quad - 2\mathbb{E}\langle \mathbf{y}_{t+1} - \mathbf{y}_t, A\mathbf{u}^*(\mathbf{x}_t, \mathbf{y}_t, \mathbf{z}_t) - \mathbf{b}\rangle$$
$$\quad - \mu\mathbb{E}\langle \mathbf{z}_t - \mathbf{z}_{t+1}, 2\mathbf{u}^*(\mathbf{x}_t, \mathbf{y}_{t+1}, \mathbf{z}_t) - \mathbf{z}_{t+1} - \mathbf{z}_t\rangle.$$

It is easy to notice that combining the last two lemmas will give us a bound on the change of $\varphi_{1/\lambda}$ from $t$ to $t+1$. On the other hand, the inner products appearing on the right-hand side of the last bound will require an intricate analysis after combining with the terms coming from other components in the potential function, introduced next. One aim, is to make sure we get enough slack to be able to cancel error terms coming from $\|\mathbf{z}_{t+1} - \mathbf{z}_t\|^2$ in the previous lemma and further errors that will arise as we handle the inner products.

### 3.3. Proof Sketch

3.3.1. ONE ITERATION INEQUALITY ON THE POTENTIAL

As alluded to earlier, we introduce the potential function we work with, which incorporates the Moreau envelopes defined in (10) and (7):

$$V_t = \varphi_{1/\lambda}(\mathbf{x}_t, \mathbf{y}_t, \mathbf{z}_t) - 2d(\mathbf{y}_t, \mathbf{z}_t) + 2\Psi(\mathbf{z}_t),$$

where we used the new notation

$$d(\mathbf{y}, \mathbf{z}) = \min_{\mathbf{x} \in X} L_\rho(\mathbf{x}, \mathbf{y}) + \frac{\mu}{2}\|\mathbf{x} - \mathbf{z}\|^2. \qquad (12)$$

There are two main changes compared to the analysis of (Zhang & Luo, 2022). The first is that the *primal descent* portion of our analysis investigates the behavior of the Moreau envelope of the proximal AL function (given in (10)) whereas the analysis of (Zhang & Luo, 2022) analyzes the proximal AL function (given in (19)) directly.

The reason for this departure is the well-known difficulty while analyzing SGD for constrained problems with single

sample of stochastic gradients. Hence, it is not clear if it is possible to show a useful inequality with the proximal AL function in the constrained case. In particular, until the work of (Davis & Drusvyatskiy, 2019), convergence analyses of projected SGD required large batches.

In addition to combining the bounds from the previous section on the change of $\varphi_{1/\lambda}$, we have to characterize the change in $d(\mathbf{y}, \mathbf{z})$ and $\Psi(\mathbf{z})$, for which we can use the following estimations, which only use the definition of $\mathbf{y}_{t+1}$ and hence have the same proof as the previous work.

**Lemma 3.5.** *(Zhang & Luo, 2020, Lemma 3.2, Lemma 3.3) For $d(\mathbf{y}, \mathbf{z})$ and $\Psi(\mathbf{z})$ defined in (7) and (12), we have*

$$2d(\mathbf{y}_{t+1}, \mathbf{z}_{t+1}) - 2d(\mathbf{y}_t, \mathbf{z}_t)$$
$$\geq 2\eta\langle A\mathbf{x}_t - \mathbf{b}, A\mathbf{x}^*(\mathbf{y}_{t+1}, \mathbf{z}_t) - \mathbf{b}\rangle$$
$$\quad + \mu\langle \mathbf{z}_{t+1} - \mathbf{z}_t, \mathbf{z}_{t+1} + \mathbf{z}_t - 2\mathbf{x}^*(\mathbf{y}_{t+1}, \mathbf{z}_{t+1})\rangle,$$

*and*

$$\Psi(\mathbf{z}_{t+1}) - \Psi(\mathbf{z}_t) \leq \mu\langle \mathbf{z}_{t+1} - \mathbf{z}_t, \mathbf{z}_t - \bar{\mathbf{x}}^*(\mathbf{z}_t)\rangle$$
$$\quad + \frac{\mu}{2\sigma_4}\|\mathbf{z}_t - \mathbf{z}_{t+1}\|^2,$$

*where $\sigma_4 = \frac{\mu - L_f}{\mu}$ and*

$$\mathbf{x}^*(\mathbf{y}, \mathbf{z}) = \arg\min_{\mathbf{x} \in X} L_\rho(\mathbf{x}, \mathbf{y}) + \frac{\mu}{2}\|\mathbf{x} - \mathbf{z}\|^2, \qquad (13)$$

$$\bar{\mathbf{x}}^*(\mathbf{z}) = \arg\min_{\mathbf{x} \in X, A\mathbf{x} = \mathbf{b}} f(\mathbf{x}) + \frac{\mu}{2}\|\mathbf{x} - \mathbf{z}\|^2. \qquad (14)$$

We continue with the main inequality on the potential function with one iteration of Alg. 1. The proof of this lemma is rather intricate and requires a careful combination of the inner products coming from the previous lemmas, and uses the particular update of the proximal center $\mathbf{z}_{t+1}$ as well as parameter selections. Recall that $\mathbf{u}^*(\mathbf{x}, \mathbf{y}, \mathbf{z})$ and $\mathbf{x}^*(\mathbf{y}, \mathbf{z})$ appearing in the lemma are defined in (11) and (13).

**Lemma 3.6** (cf. Lemma A.9). *With Assumption 1.1 and parameters in (8) (see (25)), we have for Alg. 1 that*

$$\mathbb{E}V_t - \mathbb{E}V_{t+1} \geq c_\beta\mathbb{E}\|\mathbf{z}_{t+1} - \mathbf{z}_t\|^2 - \lambda\tau^2\sigma^2/2$$
$$\quad + c_\tau\mathbb{E}\|\mathbf{u}^*(\mathbf{x}_t, \mathbf{y}_{t+1}, \mathbf{z}_t) - \mathbf{x}_t\|^2$$
$$\quad + c_\eta\mathbb{E}\|A\mathbf{x}^*(\mathbf{y}_{t+1}, \mathbf{z}_t) - \mathbf{b}\|^2, \qquad (15)$$

*where $c_\tau = \Theta(1/\sqrt{T})$, $c_\eta = \Theta(1/\sqrt{T})$, $c_\beta = \Theta(1/\sqrt{T})$ with their precise definitions given in Lemma A.9.*

One novelty in our analysis is to show that this potential function is still lower bounded and decreases, in expectation, up to an error term depends on $\tau^2$ and the variance. To integrate this change into the framework of (Zhang & Luo, 2022) under reasonable assumptions on the stochastic oracle as mentioned earlier in Section 2, we also slightly changed the definition of $\mathbf{z}_{t+1}$ in the algorithm, due to technical reasons. In particular, in our case, we lose the control over $\|\mathbf{x}_{t+1} - \mathbf{x}_t\|^2$ (since we do not assume bounded domains in this section), whereas the deterministic analysis of (Zhang & Luo, 2022) have a natural control over such terms.

The other change is the error coming from the variance of stochastic gradients. This causes the complexity to deteriorate compared to the deterministic case, which is an effect common with algorithms based on SGD. In particular, with a correctly selected step size, we obtain a sample complexity with the same-order as SGD, which is optimal even for unconstrained nonconvex problems (Arjevani et al., 2023).

### 3.3.2. COMPLEXITY ANALYSIS

After Lemma 3.6, it is straightforward to obtain

$$\mathbb{E}\|\mathbf{z}_{t+1} - \mathbf{z}_t\|^2 \le \varepsilon^2,$$
$$\mathbb{E}\|A\mathbf{x}^*(\mathbf{y}_{t+1}, \mathbf{z}_t) - \mathbf{b}\|^2 \le \varepsilon^2,$$
$$\mathbb{E}\|\mathbf{u}^*(\mathbf{x}_t, \mathbf{y}_{t+1}, \mathbf{z}_t) - \mathbf{x}_t\|^2 \le \varepsilon^2,$$

when $T = \Theta(\varepsilon^{-4})$. Then, by tedious but straightforward calculations, we can directly get the bound on the norm of the gradient of the Moreau envelope, $\nabla \Psi(\mathbf{z}_t)$, obtaining near-stationarity. The details appear in Appendix A.2.

A couple more steps let us go from this result to an $\varepsilon$-stationary point. The idea is simple: since we know that small $\|\nabla\Psi(\mathbf{z}_t)\|$ means that we are near a stationary point, we can perform just *one* more iteration of SGD with batch size $\approx \varepsilon^{-2}$ to get an $\varepsilon$-stationary point, without changing the worst-case complexity. The details are in App. A.3.

## 4. Extension to Random Linear Constraints

We turn to the case when constraints are sampled, that is, we do not have access to the full matrix $A$, or vector $\mathbf{b}$ but only unbiased samples of them. This is a suitable setting, when, for example, we have a large matrix $A$. In particular, we have $A = \mathbb{E}_{\zeta \sim P}[A_\zeta], \mathbf{b} = \mathbb{E}_{\zeta \sim P}[\mathbf{b}_\zeta]$ and use $A_\zeta, \mathbf{b}_\zeta$ in the algorithm. We rewrite the template for convenience, as

$$\min_{\mathbf{x} \in X} f(\mathbf{x}) \text{ subject to } \mathbb{E}_{\zeta \sim P}[A_\zeta \mathbf{x} - \mathbf{b}_\zeta] = 0. \quad (16)$$

In this case, to get an unbiased stochastic gradient for proximal AL, we need to sample two i.i.d. samples of $\zeta$:

$$
\begin{aligned}
G(\mathbf{x}, \mathbf{y}, \mathbf{z}, \xi) = &\nabla f(\mathbf{x}, \xi) \\
&+ A_{\zeta^1}^\top \mathbf{y} + \rho A_{\zeta^1}^\top (A_{\zeta^2}\mathbf{x} - \mathbf{b}_{\zeta^2}) + \mu(\mathbf{x} - \mathbf{z}).
\end{aligned} \quad (17)
$$

An immediate issue here is that the variance of stochastic gradients of the proximal AL function scales linearly with $\mathbf{x}$ and $\mathbf{y}$. Hence, assuming bounded variance would require assuming bounded dual variables, which is a strong assumption that is not satisfied in practice. To go around this difficulty, we have two adjustments, *(i)* we assume a constraint qualification (CQ) and compactness of $X$ and *(ii)* we include a safeguarding procedure in the algorithm to monitor when the dual variable gets too large. Under these two modifications, we obtain the same complexity guarantees as our previous setting with deterministic constraints.

**Assumption 4.1.** For problem (16), the following holds:

1. The feasible set $\{\mathbf{x} : \mathbf{x} \in X, A\mathbf{x} = \mathbf{b}\}$ is bounded.

2. The origin is in the relative interior of the set $\{A\mathbf{x} - \mathbf{b} : \mathbf{x} \in X\}$.

3. $A$ has full row-rank.

In addition to the assumptions in the earlier setting, we require a Slater's condition as well as compact domains to ensure boundedness of the dual variable. Slater's condition is a classical CQ, see e.g., (Bertsekas et al., 2003, Sec. 5.3.1).

*Remark* 4.2. The choice of $M_y$ is given next, which admittedly can be difficult in practice. Let $M_V = \max_{\mathbf{x}, \mathbf{z} \in X}\{K(\mathbf{x}, 0, \mathbf{z}) - 2d(0, \mathbf{z}) + 2\Psi(\mathbf{z})\}$, $M = \max_{\mathbf{x}, \mathbf{z} \in X}\{|f(\mathbf{x})| + \frac{\mu}{2}\|\mathbf{x} - \mathbf{z}\|^2 + \frac{\rho}{2}\|A\mathbf{x} - \mathbf{b}\|^2\}$, where $K$ is defined in (19) and $M_\Psi$ is a uniform lower bound of $\Psi(\mathbf{z}_t)$, e.g., $\underline{f}$. According to Assumption 4.1, there exists $r > 0$ such that for any direction $\mathbf{d} \in \text{Range}(A)$, we can find $\mathbf{x} \in X$ satisfying $\|A\mathbf{x} - \mathbf{b}\| = r$ and $A\mathbf{x} - \mathbf{b}$ has the same direction as $\mathbf{d}$. Then, we choose $M_y$ as $M_y > \frac{M_V - M_\Psi + 2M}{r}$.

In this setting, we only state our theorem for near-stationarity. The $\varepsilon$-stationarity would follow in the same way as the previous section by a post-processing step.

**Theorem 4.3.** *Let Assumptions 1.1 and 4.1 hold and run Alg. 2 with parameters from (8). We have that $\mathbb{E}\|\nabla\Psi(\mathbf{z}_{t^*})\| \le \varepsilon$ where $t^*$ is randomly selected from $\{0, \ldots, T-1\}$ with $T = \Omega(\varepsilon^{-4})$. The stochastic oracle complexity is $O(\varepsilon^{-4})$.*

As mentioned earlier, the optimal sample complexity for nonconvex optimization with Lipschitz $\nabla f$ is $O(\varepsilon^{-4})$ (Arjevani et al., 2023). Our result matches this complexity while handling linear constraints with random sampling.

## 5. Extension with Variance Reduction

We now integrate the STORM variance reduction technique from (Cutkosky & Orabona, 2019) into our framework to solve (1) (See arXiv:2504.07607 for extension to stochastic constraints). We obtain Alg. 3, which improves the iteration and oracle complexity from $O(\varepsilon^{-4})$ to $O(\varepsilon^{-3})$ under a stronger assumption on the oracle, compared to Sec. 3. This not only leads to an improved rate, but also to a simpler analysis that does not rely on the Moreau envelope $\varphi_{1/\lambda}$.

---

**Algorithm 2** Stochastic smoothed and linearized ALM for stochastic constraints with dual safeguarding

**Input and Initialization:** $M_y > \frac{M_V - M_\Psi + 2M}{r}$ (check Remark 4.2), $\mathbf{x}_0 = \mathbf{z}_0 \in X$, $\mathbf{y}_0 \in \mathbb{R}^m$, $\rho \geq 0$.
**for** $t = 0$ **to** $T - 1$ **do**
    $\mathbf{y}_{t+1} = \mathbf{y}_t + \eta(A_{\zeta_t}\mathbf{x}_t - \mathbf{b}_{\zeta_t})$ where $\zeta_t \sim P$ is generated i.i.d.
    **if** $\|\mathbf{y}_{t+1}\| \geq M_y$ **then**
        $\mathbf{y}_{t+1} = 0$
    Sample $\xi_t \in \Xi$ i.i.d. and generate $\mathbb{E}_{\xi_t}[G(\mathbf{x}_t, \mathbf{y}_{t+1}, \mathbf{z}_t, \xi_t)] = \nabla_{\mathbf{x}}L_\rho(\mathbf{x}_t, \mathbf{y}_{t+1}) + \mu(\mathbf{x}_t - \mathbf{z}_t)$ as in (17)
    $\mathbf{x}_{t+1} = \text{proj}_X(\mathbf{x}_t - \tau G(\mathbf{x}_t, \mathbf{y}_{t+1}, \mathbf{z}_t, \xi_t))$
    $\mathbf{z}_{t+1} = \mathbf{z}_t + \beta(\mathbf{x}_t - \mathbf{z}_t)$

---

**Algorithm 3** Stochastic smoothed and linearized ALM with STORM

**Initialize:** $\mathbf{x}_0 = \mathbf{z}_0 \in X$, $\mathbf{y}_0 \in \mathbb{R}^m$, $\widehat{\nabla}f_0 = \frac{1}{N}\sum_{i=1}^{N}\nabla f(\mathbf{x}_0, \zeta_i)$, $N = T^{1/3}$ and $\rho \geq 0$
**for** $t = 0$ **to** $T - 1$ **do**
    $\mathbf{y}_{t+1} = \mathbf{y}_t + \eta(A\mathbf{x}_t - \mathbf{b})$
    $G(\mathbf{x}_t, \mathbf{y}_{t+1}, \mathbf{z}_t) = \widehat{\nabla}f_t + A^\top\mathbf{y}_{t+1} + \rho A^\top(A\mathbf{x}_t - \mathbf{b}) + \mu(\mathbf{x}_t - \mathbf{z}_t)$
    $\mathbf{x}_{t+1} = \text{proj}_X(\mathbf{x}_t - \tau G(\mathbf{x}_t, \mathbf{y}_{t+1}, \mathbf{z}_t))$
    $\mathbf{z}_{t+1} = \mathbf{z}_t + \beta(\mathbf{x}_t - \mathbf{z}_t)$
    Sample $\xi_{t+1} \sim \Xi$ i.i.d. and set $\widehat{\nabla}f_{t+1} = \nabla f(\mathbf{x}_{t+1}, \xi_{t+1}) + (1 - \alpha)(\widehat{\nabla}f_t - \nabla f(\mathbf{x}_t, \xi_{t+1}))$

---

Alg. 3 and Alg. 1 mainly differ in the update of stochastic gradient estimate $\widehat{\nabla}f_t$. If $\alpha = 0$, Alg. 3 trivially reduces to Alg. 1. We next see that a particular choice of $\alpha$ gives better complexity under Assumption 5.2 (which is stronger than the oracle access and smoothness in Assumption 1.1).

*Remark* 5.1. We only use a minibatch in the initialization, which does not affect the overall complexity. The minibatch size is $N = T^{1/3}$, which is small compared to the total number of iterations $T$. Iterations of our algorithm only require 2 stochastic gradients, $\nabla f(\mathbf{x}_t, \xi_{t+1})$ and $\nabla f(\mathbf{x}_{t+1}, \xi_{t+1})$.

For the analysis of Alg. 3, we introduce Assumption 5.2, used, e.g., in (Arjevani et al., 2023). In particular, Arjevani et al. (2023) showed that the oracle complexity $O(\varepsilon^{-3})$ is tight under Assumption 5.2 even with no constraints.

**Assumption 5.2.** We have access to a stochastic gradient of $f$ satisfying (3). For a given $\xi \sim \Xi$, we can query $\nabla f(\mathbf{x}, \xi)$ and $\nabla f(\mathbf{y}, \xi)$ for different points $\mathbf{x}, \mathbf{y}$. Moreover, we have $\mathbb{E}_{\xi \sim \Xi}\|\nabla f(\mathbf{x}, \xi) - \nabla f(\mathbf{y}, \xi)\|^2 \leq L_0^2\|\mathbf{x} - \mathbf{y}\|^2$.

We introduce the potential $\bar{V}_t$ differing from Sec. 3 and 4. This is similar to (Zhang & Luo, 2022), except the last term which controls the error from the variance. Define

$$\bar{V}_t = K(\mathbf{x}_t, \mathbf{y}_t, \mathbf{z}_t) - 2d(\mathbf{y}_t, \mathbf{z}_t) + 2\Psi(\mathbf{z}_t)$$
$$+ \frac{1}{48(L_0^2 + L_f^2)\tau}\|\widehat{\nabla}f_t - \nabla f(\mathbf{x}_t)\|^2, \quad (18)$$

where

$$K(\mathbf{x}, \mathbf{y}, \mathbf{z}) = L_\rho(\mathbf{x}, \mathbf{y}) + \frac{\mu}{2}\|\mathbf{x} - \mathbf{z}\|^2. \quad (19)$$

One-step evolution of $\hat{V}_t$ that we analyze next is a key step in the analysis. Compared to (Zhang & Luo, 2022), we have the extra error due to using $\widehat{\nabla}f_t$ instead of the full gradient.

**Lemma 5.3** (cf. Lemma C.4). *Under Assumptions 1.1 and 5.2, with parameters*

$$\mu = \max\{2, 4L_f\}, \quad \tau = T^{-3/2},$$
$$\eta = \Theta(\tau), \quad \beta = \Theta(\tau), \quad \alpha = \Theta(\tau^2), \quad (20)$$

*(for detailed parameters, see (82)) we have*

$$\mathbb{E}\bar{V}_t - \mathbb{E}\bar{V}_{t+1} \geq \frac{2\mu}{\beta}\mathbb{E}\|\mathbf{z}_t - \mathbf{z}_{t+1}\|^2 + \frac{1}{2\tau}\mathbb{E}\|\mathbf{x}_t - \mathbf{x}_{t+1}\|^2$$
$$+ 2\eta\mathbb{E}\|A\mathbf{x}^*(\mathbf{y}_{t+1}, \mathbf{z}_t) - b\|^2$$
$$+ \tau\mathbb{E}\|\widehat{\nabla}f_t - \nabla f(\mathbf{x}_t)\|^2 - O(\sigma^2\tau^3). \quad (21)$$

Note that, on a high level, the main difference between Lemma 5.3 and Lemma 3.6 is that the order of $\tau$ in the error term is different. In Lemma 5.3, the order of $\tau$ is $O(\tau^3)$, while in Lemma 3.6, the order of $\tau$ is $O(\tau^2)$, which contribute to a faster convergence rate in for Alg. 3.

**Theorem 5.4.** *Let Assumptions 1.1 and 5.2 hold. We have that $\mathbb{E}\|\nabla\Psi(\mathbf{z}_{t^*})\| \leq \varepsilon$, where $t^*$ is selected uniformly at random from $\{0, \ldots, T - 1\}$ with $T = \Omega(\varepsilon^{-3})$. The complexity of the whole procedure is $O(\varepsilon^{-3})$.*

## 6. Related Works

We now compare the complexity results for obtaining an $\varepsilon$-stationary point, in view of Section 1.2.

**Deterministic objective and deterministic constraints.** The setting when objective $f$ in (1) is deterministic is the most well-studied with many results in the classical literature (Bertsekas, 2014). Recent work characterized the global oracle complexity of Lagrangian-based methods or ALM.

With nonlinear and nonconvex constraints, many of the existing works analyzing AL-based algorithms rely on strong CQs and boundedness assumptions and use large penalty parameters to ensure feasibility (Li et al., 2021; Lin et al., 2022; Kong et al., 2019; Kong & Monteiro, 2023; Kong et al., 2023). The existing frameworks so far fail to capture the importance of dual variable updates, which are, in fact, the main reason behind the ability to use constant penalty parameters while ensuring convergence, see e.g., (Bertsekas, 2014, Sec. 2.2.5). Recent works mentioned above obtained the complexity bound $O(\varepsilon^{-3})$ for general nonlinear constraints with no specialization for linear constraints. When specialized to convex functional constraints, the best-known complexity for these methods is $O(\varepsilon^{-2.5})$ (Lin et al., 2022).

When the constraints are linear, such as (1) with $X = \mathbb{R}^n$, Hong (2016) analyzed ALM with constant penalty parameters and non-negligible dual updates to get optimal complexity $O(\varepsilon^{-2})$. The case of $X \neq \mathbb{R}^n$ turned out to be significantly more challenging with many works focusing on variants of ALM with large penalty parameters (depending on the inverse of the final accuracy) to ensure near-feasibility and *negligible* dual updates that do not help with feasibility (Kong & Monteiro, 2023; Kong et al., 2023) and obtained the suboptimal complexity $\widetilde{O}(\varepsilon^{-2.5})$. The exceptions are the works (Zhang & Luo, 2020; 2022) that showed, for the case $X$ polyhedral, near-optimal complexity $O(\varepsilon^{-2})$ with a constant penalty parameter and dual steps with constant step sizes, with no constraint qualification. The key step was the global error bound that our work also relied on.

**Stochastic objective and deterministic constraints.** One important step in generalizing the template to tasks arising in ML was to consider stochastic objectives where we access unbiased estimates. With general nonlinear constraints and Lipschitzness of $\nabla f$, the optimal sample complexity is $O(\varepsilon^{-4})$, obtained with double loop algorithms (Curtis et al., 2024; Boob et al., 2023; Ma et al., 2020). These works require strong assumptions on the boundedness of the primal domain as well as constraint qualifications, which are often not necessary with linear constraints.

Another set of results concerns stochastic optimization with deterministic nonlinear constraints with penalty-based algorithms. These works require large penalty parameters to ensure near-feasibility rather than dual updates (Lu et al., 2024; Alacaoglu & Wright, 2024). They assume expected Lipschitzness as Assumption 5.2, which is stronger than Lipschitzness of $\nabla f$. Since these works focus on nonlinear functional constraints, the analysis requires boundedness assumptions as well as constraint qualifications, unlike our results in Section 3 for deterministic linear constraints.

Alacaoglu & Wright (2024) considered ALM with a constant penalty parameter and non-negligible dual updates and obtained the complexity $O(\varepsilon^{-3})$ for linear *equality* con-

straints under Assumption 5.2. This work only covered the case $X = \mathbb{R}^n$ and left open the question of handling the case of general $X$, see (Alacaoglu & Wright, 2024, Sec. 5).

We resolve a special case of this question when $X$ is polyhedral (covering many applications), allowing our analysis to cover linear inequality constraints. Alacaoglu & Wright (2024) used variance reduction for $\nabla f$, which meant that they required Assumption 5.2, stronger than Assumption 1.1. In Sec. 5, we get the same complexity as this paper while allowing a polyhedral $X$ to cover linear inequality constraints, which cannot be handled by Alacaoglu & Wright (2024).

Moreover, we also get the complexity $O(\varepsilon^{-4})$ under Assumption 1.1. This is optimal under Assumption 1.1 and we refer to (Arjevani et al., 2023) for further details on the lower bounds. In contrast, the work in (Alacaoglu & Wright, 2024) does not have guarantees without Assumption 5.2.

In addition, though (Lu et al., 2024) considers the more general problem with nonconvex functional constraints, they make strong assumptions which are not easy to verify. It is not clear if their assumptions would hold with a general polyhedral constraint we have (see e.g., their Assumption 1(iv) and Eq. (7)). When the constraints are deterministic, we do not have any bounded domain assumption (our Sec. 3) whereas the assumptions of (Lu et al., 2024) are rather difficult to be satisfied without a bounded primal domain.

Lu et al. (2024) analyzes a QP-based method, whereas we analyze an ALM-variant. ALM is known to be more stable and desirable in practice, but significantly more difficult to analyze, which is because the penalty parameter is fixed in ALM and it increases to infinity for QP. Our ALM algorithm could be extended to stochastic constraints, while (Lu et al., 2024) only handles deterministic constraints. Alacaoglu & Wright (2024) highlights the importance of analyzing ALM compared to QP methods in their Sections 1 and 6.

**Stochastic objective and stochastic constraints.** This is the most general class, where the existing results come with many assumptions that are not always easy to interpret, similar to the case of stochastic objective and deterministic constraints described above (Li et al., 2024; Alacaoglu & Wright, 2024). The best-known complexity $O(\varepsilon^{-5})$ is obtained by using Assumption 5.2, with an inexact, double-loop, ALM in (Li et al., 2024) and by a single-loop QP algorithm in (Alacaoglu & Wright, 2024). These results concerning ALM need to use large penalty parameters, which renders them essentially as QP-methods since the dual updates do not contribute to the analysis for ensuring the feasibility. Other approaches for solving this sub-case also require double-loop algorithms and stronger assumptions since they focus on a generic nonconvex constraint (Boob et al., 2023; Ma et al., 2020), obtaining $O(\varepsilon^{-6})$ without expected Lipschitzness. Hence, in this sub-case, none of these results harness the structure of linear constraints.

## Acknowledgements

Jiawei Zhang is supported by the startup fund from the Department of Computer Sciences at the University of Wisconsin–Madison and the MIT Postdoctoral Fellowship for Engineering Excellence.

Ahmet Alacaoglu acknowledges the support of the Natural Sciences and Engineering Research Council of Canada (NSERC), [funding reference number RGPIN-2025-06634].

## Impact Statement

*This paper presents work whose goal is to advance the field of Machine Learning. There are many potential societal consequences of our work, none which we feel must be specifically highlighted here.*

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

### Notation.

Let us note that we define by $\mathbb{E}_t$ the expectation conditioned on all the randomness up to and including $\mathbf{x}_t$.

## A. Proofs for Section 3

In the proofs, let us recall

$$
\begin{aligned}
K(\mathbf{x}, \mathbf{y}, \mathbf{z}) &= L_\rho(\mathbf{x}, \mathbf{y}) + \frac{\mu}{2} \|\mathbf{x} - \mathbf{z}\|^2 \\
&= f(\mathbf{x}) + \langle A\mathbf{x} - \mathbf{b}, \mathbf{y} \rangle + \frac{\rho}{2} \|A\mathbf{x} - \mathbf{b}\|^2 + \frac{\mu}{2} \|\mathbf{x} - \mathbf{z}\|^2.
\end{aligned}
\tag{22}
$$

With this notation, we have the following, equivalent to (11):

$$
\begin{aligned}
\mathbf{u}^*(\mathbf{x}, \mathbf{y}, \mathbf{z}) &= \arg\min_{\mathbf{u} \in X} \left\{ K(\mathbf{u}, \mathbf{y}, \mathbf{z}) + \frac{\lambda}{2} \|\mathbf{u} - \mathbf{x}\|^2 \right\} \\
&= \arg\min_{\mathbf{u} \in X} \left\{ L_\rho(\mathbf{u}, \mathbf{y}, \mathbf{z}) + \frac{\mu}{2} \|\mathbf{u} - \mathbf{z}\|^2 + \frac{\lambda}{2} \|\mathbf{u} - \mathbf{x}\|^2 \right\}.
\end{aligned}
\tag{23}
$$

We also recall (10).

$$
\begin{aligned}
\varphi_{1/\lambda}(\mathbf{x}, \mathbf{y}, \mathbf{z}) &= \min_{\mathbf{u} \in X} \left\{ L_\rho(\mathbf{u}, \mathbf{y}) + \frac{\mu}{2} \|\mathbf{u} - \mathbf{z}\|^2 + \frac{\lambda}{2} \|\mathbf{u} - \mathbf{x}\|^2 \right\} \\
&= \min_{\mathbf{u} \in X} \left\{ K(\mathbf{u}, \mathbf{y}, \mathbf{z}) + \frac{\lambda}{2} \|\mathbf{u} - \mathbf{x}\|^2 \right\}.
\end{aligned}
\tag{24}
$$

We also introduce here some parameters that are used throughout, for convenience.

$$
\begin{aligned}
\mu &= \max\{2, 4L_f\}, \\
L_K &= L_f + \rho\|A\| + \mu, \\
\lambda &= 2L_K, \\
\sigma_4 &= \frac{\mu - L_f}{\mu}, \\
\tau &= \frac{1}{6\lambda^2\sqrt{T}}, \\
\eta &= \min\left\{ \frac{2\mu + \rho\|A\|}{4\|A\|^4}, \frac{\tau}{200\|A\|^2}, \frac{\tau(2\mu + \rho\|A\|^2)}{20\|A\|^2} \right\}, \\
\beta &= \min\left\{ \frac{\tau}{100}, \frac{1}{50\lambda}, \frac{\eta}{36\mu\bar{\sigma}^2} \right\}, \\
\gamma_s &= 2\mu + \rho\|A\|, \gamma = \frac{(\mu - L_f)\lambda}{\mu - L_f + \lambda}, \gamma_K = \mu - L_f.
\end{aligned}
\tag{25}
$$

We also mention the following basic facts that are used in the sequel.

**Fact A.1.** *For $\mathbf{x} \in X$, we have that $\mathbf{x} \mapsto K(\mathbf{x}, \mathbf{y}, \mathbf{z})$ is strongly convex with modulus $\gamma_K := \mu - L_f$, and $\mathbf{x} \mapsto \nabla_{\mathbf{x}} K(\mathbf{x}, \mathbf{y}, \mathbf{z})$ is $L_K := (L_f + \rho\|A\|^2 + \mu)$-Lipschitz continuous.*

*For $\mathbf{u} \in X$, $\mathbf{u} \mapsto K(\mathbf{u}, \mathbf{y}, \mathbf{z}) + \frac{\lambda}{2}\|\mathbf{x} - \mathbf{u}\|^2$ is strongly convex with modulus $\gamma_s = \mu - L_f + \lambda$, and $\mathbf{u}^*(\mathbf{x}, \mathbf{y}, \mathbf{z}) = \arg\min_{\mathbf{u} \in X} K(\mathbf{u}, \mathbf{y}, \mathbf{z}) + \frac{\lambda}{2}\|\mathbf{x} - \mathbf{u}\|^2$.*

**Lemma A.2.** *(Planiden & Wang, 2016, Lemma 2.19) Let $r > 0$. The function $f$ is $r$-strongly convex if and only if $f_1(\mathbf{x}) = \min_{\mathbf{u}} f(\mathbf{u}) + \frac{1}{2}\|\mathbf{x} - \mathbf{u}\|^2$ is $\frac{r}{r+1}$-strongly convex.*

**Lemma A.3.** *The function $\mathbf{x} \mapsto \varphi_{1/\lambda}(\mathbf{x}, \mathbf{y}, \mathbf{z})$ is $\gamma = \frac{(\mu - L_f)\lambda}{\mu - L_f + \lambda}$-strongly convex.*

*Proof.* By definition, we have

$$
\varphi_{1/\lambda}(\mathbf{x}, \mathbf{y}, \mathbf{z}) = \min_{\mathbf{u}} K(\mathbf{u}, \mathbf{y}, \mathbf{z}) + I_X(\mathbf{u}) + \frac{\lambda}{2}\|\mathbf{x} - \mathbf{u}\|^2 = \lambda \min_{\mathbf{u}} \frac{K(\mathbf{u}, \mathbf{y}, \mathbf{z}) + I_X(\mathbf{u})}{\lambda} + \frac{1}{2}\|\mathbf{x} - \mathbf{u}\|^2.
$$

Recall that $\gamma_K = \mu - L_f$. Then, since $K(\mathbf{x}, \mathbf{y}, \mathbf{z})/\lambda$ is $\frac{\gamma_K}{\lambda}$-strongly convex, we have $\min_{\mathbf{u}} \frac{K(\mathbf{u}, \mathbf{y}, \mathbf{z}) + I_X(\mathbf{u})}{\lambda} + \frac{1}{2}\|\mathbf{x} - \mathbf{u}\|^2$ is $\frac{\gamma_K/\lambda}{\gamma_K/\lambda + 1}$-strongly convex, by Lemma A.2. Hence, $\varphi_{1/\lambda}(\mathbf{x}, \mathbf{y}, \mathbf{z})$ is strongly convex with modulus $\frac{\gamma_K}{\gamma_K/\lambda + 1} = \frac{\lambda \gamma_K}{\lambda + \gamma_K} = \frac{(\mu - L_f)\lambda}{\mu - L_f + \lambda}$. ∎

## A.1. Proofs for Lemma 3.6

In the next lemma, the first part is using the idea of Davis & Drusvyatskiy (2019) to analyze the algorithm under the bounded variance assumption instead of the restrictive bounded stochastic gradient assumption. The second part of the lemma also follows a similar idea as this work, with the exception of the dependence on the changing center point $\mathbf{z}_t$. This introduces additional issues, since the stochastic gradient in the update of $\mathbf{x}_{t+1}$ depends on $\mathbf{z}_t$ whereas the proximal point $\mathbf{u}^*(\mathbf{x}_t, \mathbf{y}_{t+1}, \mathbf{z}_{t+1})$ (that characterizes the iteration below) depends on $\mathbf{z}_{t+1}$. Our analysis below estimates this additional error and shows it to be in the order of $\|\mathbf{z}_{t+1} - \mathbf{z}_t\|^2$, which will be handled later.

**Lemma A.4.** *Suppose that Assumption 1.1 holds, for the proximal point $\mathbf{u}^*(\mathbf{x}_t, \mathbf{y}_{t+1}, \mathbf{z}_{t+1})$, defined as (11) we have the characterization*

$$\mathbf{u}^*(\mathbf{x}_t, \mathbf{y}_{t+1}, \mathbf{z}_{t+1}) = \text{proj}_X(\tau\lambda\mathbf{x}_t + (1 - \tau\lambda)\mathbf{u}^*(\mathbf{x}_t, \mathbf{y}_{t+1}, \mathbf{z}_{t+1}) - \tau\nabla_{\mathbf{x}}K(\mathbf{u}^*(\mathbf{x}_t, \mathbf{y}_{t+1}, \mathbf{z}_{t+1}), \mathbf{y}_{t+1}, \mathbf{z}_{t+1})). \quad (26)$$

*Moreover, for the sequence $\mathbf{x}_{t+1}$ calculated as Algorithm 1, with $\lambda = 2L_K$ and $\tau \leq \frac{1}{6\lambda}$, where $L_K = L_f + \rho\|A\|^2 + \mu$, we have*

$$\mathbb{E}\|\mathbf{u}^*(\mathbf{x}_t, \mathbf{y}_{t+1}, \mathbf{z}_{t+1}) - \mathbf{x}_{t+1}\|^2 \leq \left(1 - \frac{\tau\lambda}{4}\right)\mathbb{E}\|\mathbf{u}^*(\mathbf{x}_t, \mathbf{y}_{t+1}, \mathbf{z}_{t+1}) - \mathbf{x}_t\|^2 + (\tau\mu + 2\tau^2\mu^2)\mathbb{E}\|\mathbf{z}_t - \mathbf{z}_{t+1}\|^2 + \tau^2\sigma^2.$$

*Proof.* From the definition of $\mathbf{u}^*(\mathbf{x}_t, \mathbf{y}_{t+1}, \mathbf{z}_{t+1})$ in (11) (see also (23)), we have

$$\lambda(\mathbf{x}_t - \mathbf{u}^*(\mathbf{x}_t, \mathbf{y}_{t+1}, \mathbf{z}_{t+1})) \in \nabla_{\mathbf{x}}K(\mathbf{u}^*(\mathbf{x}_t, \mathbf{y}_{t+1}, \mathbf{z}_{t+1}), \mathbf{y}_{t+1}, \mathbf{z}_{t+1}) + \partial I_X(\mathbf{u}^*(\mathbf{x}_t, \mathbf{y}_{t+1}, \mathbf{z}_{t+1})).$$

Multiplying both sides by the step size $\tau$, adding $\mathbf{u}^*(\mathbf{x}_t, \mathbf{y}_{t+1}, \mathbf{z}_{t+1})$ to both sides, and rearranging give

$$\tau\lambda\mathbf{x}_t - \tau\nabla_{\mathbf{x}}K(\mathbf{u}^*(\mathbf{x}_t, \mathbf{y}_{t+1}, \mathbf{z}_{t+1}), \mathbf{y}_{t+1}, \mathbf{z}_{t+1}) + (1 - \tau\lambda)\mathbf{u}^*(\mathbf{x}_t, \mathbf{y}_{t+1}, \mathbf{z}_{t+1})$$
$$\in \mathbf{u}^*(\mathbf{x}_t, \mathbf{y}_{t+1}, \mathbf{z}_{t+1}) + \tau\partial I_X(\mathbf{u}^*(\mathbf{x}_t, \mathbf{y}_{t+1}, \mathbf{z}_{t+1})).$$

Since $(I + \tau\partial I_X)^{-1} = \text{prox}_{I_X} = \text{proj}_X$ due to $\partial I_X$ being a cone and proximal operator of a normal cone being the projection to the set, we have the first assertion.

We next establish the second assertion. Using the just established identity (26), the update rule of $\mathbf{x}_{t+1}$ in Algorithm 1, and nonexpansiveness of the projection, we derive

$$\|\mathbf{u}^*(\mathbf{x}_t, \mathbf{y}_{t+1}, \mathbf{z}_{t+1}) - \mathbf{x}_{t+1}\|^2$$
$$\leq \|\tau\lambda\mathbf{x}_t + (1 - \tau\lambda)\mathbf{u}^*(\mathbf{x}_t, \mathbf{y}_{t+1}, \mathbf{z}_{t+1}) - \tau\nabla_{\mathbf{x}}K(\mathbf{u}^*(\mathbf{x}_t, \mathbf{y}_{t+1}, \mathbf{z}_{t+1}), \mathbf{y}_{t+1}, \mathbf{z}_{t+1}) - [\mathbf{x}_t - \tau G(\mathbf{x}_t, \mathbf{y}_{t+1}, \mathbf{z}_t, \xi_t)]\|^2.$$

We add and subtract $\nabla_{\mathbf{x}}K(\mathbf{x}_t, \mathbf{y}_{t+1}, \mathbf{z}_t)$ inside the squared norm on the right-hand side, expand and take conditional expectation to obtain

$$\mathbb{E}_t\|\mathbf{u}^*(\mathbf{x}_t, \mathbf{y}_{t+1}, \mathbf{z}_{t+1}) - \mathbf{x}_{t+1}\|^2$$
$$= \|(1 - \tau\lambda)(\mathbf{u}^*(\mathbf{x}_t, \mathbf{y}_{t+1}, \mathbf{z}_{t+1}) - \mathbf{x}_t) - \tau\nabla_{\mathbf{x}}K(\mathbf{u}^*(\mathbf{x}_t, \mathbf{y}_{t+1}, \mathbf{z}_{t+1}), \mathbf{y}_{t+1}, \mathbf{z}_{t+1}) + \tau\nabla_{\mathbf{x}}K(\mathbf{x}_t, \mathbf{y}_{t+1}, \mathbf{z}_t)\|^2$$
$$+ \tau^2\mathbb{E}_t\|G(\mathbf{x}_t, \mathbf{y}_{t+1}, \mathbf{z}_t, \xi_t) - \nabla_{\mathbf{x}}K(\mathbf{x}_t, \mathbf{y}_{t+1}, \mathbf{z}_t)\|^2, \quad (27)$$

where the cross term disappeared because

$$\mathbb{E}_t[G(\mathbf{x}_t, \mathbf{y}_{t+1}, \mathbf{z}_t, \xi_t)] = \nabla_{\mathbf{x}}K(\mathbf{x}_t, \mathbf{y}_{t+1}, \mathbf{z}_t)$$

and $\mathbf{x}_t, \mathbf{y}_{t+1}, \mathbf{z}_{t+1}, \mathbf{u}^*(\mathbf{x}_t, \mathbf{y}_{t+1}, \mathbf{z}_{t+1})$ are deterministic under the conditioning since $\mathbf{z}_{t+1}$ defined in Algorithm 1 only depends on $\mathbf{x}_t$ (that is, $\mathbf{z}_{t+1}$ is independent of $\xi_t$).

The second term on the right-hand side of (27) is trivially bounded by the oracle assumptions, that is,

$$\mathbb{E}_t\|G(\mathbf{x}_t, \mathbf{y}_{t+1}, \mathbf{z}_t, \xi_t) - \nabla_{\mathbf{x}}K(\mathbf{x}_t, \mathbf{y}_{t+1}, \mathbf{z}_t)\|^2 \leq \sigma^2. \quad (28)$$

For the first term on the right-hand side of (27), we further estimate as

$$
\begin{aligned}
&\|(1 - \tau\lambda)(\mathbf{u}^*(\mathbf{x}_t, \mathbf{y}_{t+1}, \mathbf{z}_{t+1}) - \mathbf{x}_t) - \tau\nabla_{\mathbf{x}}K(\mathbf{u}^*(\mathbf{x}_t, \mathbf{y}_{t+1}, \mathbf{z}_{t+1}), \mathbf{y}_{t+1}, \mathbf{z}_{t+1}) + \tau\nabla_{\mathbf{x}}K(\mathbf{x}_t, \mathbf{y}_{t+1}, \mathbf{z}_t)\|^2 \\
&= (1 - \tau\lambda)^2 \|\mathbf{u}^*(\mathbf{x}_t, \mathbf{y}_{t+1}, \mathbf{z}_{t+1}) - \mathbf{x}_t\|^2 \\
&\quad + 2\tau(1 - \tau\lambda)\langle \mathbf{u}^*(\mathbf{x}_t, \mathbf{y}_{t+1}, \mathbf{z}_{t+1}) - \mathbf{x}_t, \nabla_{\mathbf{x}}K(\mathbf{x}_t, \mathbf{y}_{t+1}, \mathbf{z}_t) - \nabla_{\mathbf{x}}K(\mathbf{u}^*(\mathbf{x}_t, \mathbf{y}_{t+1}, \mathbf{z}_{t+1}), \mathbf{y}_{t+1}, \mathbf{z}_{t+1})\rangle \\
&\quad + \tau^2 \|\nabla_{\mathbf{x}}K(\mathbf{x}_t, \mathbf{y}_{t+1}, \mathbf{z}_t) - \nabla_{\mathbf{x}}K(\mathbf{u}^*(\mathbf{x}_t, \mathbf{y}_{t+1}, \mathbf{z}_{t+1}), \mathbf{y}_{t+1}, \mathbf{z}_{t+1})\|^2.
\end{aligned} \tag{29}
$$

Next, we turn to estimating

$$
\begin{aligned}
&\|\nabla_{\mathbf{x}}K(\mathbf{x}_t, \mathbf{y}_{t+1}, \mathbf{z}_t) - \nabla_{\mathbf{x}}K(\mathbf{u}^*(\mathbf{x}_t, \mathbf{y}_{t+1}, \mathbf{z}_{t+1}), \mathbf{y}_{t+1}, \mathbf{z}_{t+1})\| \\
&\leq \|\nabla_{\mathbf{x}}K(\mathbf{x}_t, \mathbf{y}_{t+1}, \mathbf{z}_t) - \nabla_{\mathbf{x}}K(\mathbf{x}_t, \mathbf{y}_{t+1}, \mathbf{z}_{t+1})\| \\
&\quad + \|\nabla_{\mathbf{x}}K(\mathbf{x}_t, \mathbf{y}_{t+1}, \mathbf{z}_{t+1}) - \nabla_{\mathbf{x}}K(\mathbf{u}^*(\mathbf{x}_t, \mathbf{y}_{t+1}, \mathbf{z}_{t+1}), \mathbf{y}_{t+1}, \mathbf{z}_{t+1})\|.
\end{aligned} \tag{30}
$$

Note that, by definition, we have

$$
\nabla_{\mathbf{x}}K(\mathbf{x}_t, \mathbf{y}_{t+1}, \mathbf{z}_t) - \nabla_{\mathbf{x}}K(\mathbf{x}_t, \mathbf{y}_{t+1}, \mathbf{z}_{t+1}) = \mu(\mathbf{z}_{t+1} - \mathbf{z}_t).
$$

Using this and the $L_K$-Lipschitzness of $\nabla_{\mathbf{x}}K(\cdot, \mathbf{y}_{t+1}, \mathbf{z}_{t+1})$ as per Fact A.1, in (30), we obtain

$$
\|\nabla_{\mathbf{x}}K(\mathbf{x}_t, \mathbf{y}_{t+1}, \mathbf{z}_t) - \nabla_{\mathbf{x}}K(\mathbf{u}^*(\mathbf{x}_t, \mathbf{y}_{t+1}, \mathbf{z}_{t+1}), \mathbf{y}_{t+1}, \mathbf{z}_{t+1})\| \leq \mu\|\mathbf{z}_{t+1} - \mathbf{z}_t\| + L_K\|\mathbf{u}^*(\mathbf{x}_t, \mathbf{y}_{t+1}, \mathbf{z}_{t+1}) - \mathbf{x}_t\|.
$$

We plug this bound into the second term on the right-hand side of (29) after using Cauchy-Schwarz inequality, and then, we use Young's inequality to get

$$
\begin{aligned}
&2\tau(1 - \tau\lambda)\langle \mathbf{u}^*(\mathbf{x}_t, \mathbf{y}_{t+1}, \mathbf{z}_{t+1}) - \mathbf{x}_t, \nabla_{\mathbf{x}}K(\mathbf{x}_t, \mathbf{y}_{t+1}, \mathbf{z}_t) - \nabla_{\mathbf{x}}K(\mathbf{u}^*(\mathbf{x}_t, \mathbf{y}_{t+1}, \mathbf{z}_{t+1}), \mathbf{y}_{t+1}, \mathbf{z}_{t+1})\rangle \\
&\leq 2\tau(1 - \tau\lambda)\|\mathbf{u}^*(\mathbf{x}_t, \mathbf{y}_{t+1}, \mathbf{z}_{t+1}) - \mathbf{x}_t\|(\mu\|\mathbf{z}_{t+1} - \mathbf{z}_t\| + L_K\|\mathbf{u}^*(\mathbf{x}_t, \mathbf{y}_{t+1}, \mathbf{z}_{t+1}) - \mathbf{x}_t\|) \\
&\leq \tau(1 - \tau\lambda)(2L_K + \mu)\|\mathbf{u}^*(\mathbf{x}_t, \mathbf{y}_{t+1}, \mathbf{z}_{t+1}) - \mathbf{x}_t\|^2 + \tau(1 - \tau\lambda)\mu\|\mathbf{z}_{t+1} - \mathbf{z}_t\|^2.
\end{aligned}
$$

Using the last two inequalities in (29), along with Young's inequality, we obtain

$$
\begin{aligned}
&\|(1 - \tau\lambda)(\mathbf{u}^*(\mathbf{x}_t, \mathbf{y}_{t+1}, \mathbf{z}_{t+1}) - \mathbf{x}_t) - \tau\nabla_{\mathbf{x}}K(\mathbf{u}^*(\mathbf{x}_t, \mathbf{y}_{t+1}, \mathbf{z}_{t+1}), \mathbf{y}_{t+1}, \mathbf{z}_{t+1}) + \tau\nabla_{\mathbf{x}}K(\mathbf{x}_t, \mathbf{y}_{t+1}, \mathbf{z}_t)\|^2 \\
&\leq [(1 - \tau\lambda)^2 + \tau(1 - \tau\lambda)(2L_K + \mu) + 2\tau^2 L_K^2]\|\mathbf{u}^*(\mathbf{x}_t, \mathbf{y}_{t+1}, \mathbf{z}_{t+1}) - \mathbf{x}_t\|^2 \\
&\quad + (\tau(1 - \tau\lambda)\mu + 2\tau^2\mu^2)\|\mathbf{z}_{t+1} - \mathbf{z}_t\|^2.
\end{aligned} \tag{31}
$$

We estimate the coefficient of the first term. First, note that $1 - \tau\lambda \leq 1$. As a result, we have

$$
\begin{aligned}
(1 - \tau\lambda)^2 + \tau(1 - \tau\lambda)(2L_K + \mu) + 2\tau^2 L_K^2 &\leq 1 - 2\tau\lambda + \tau^2\lambda^2 + \tau(2L_K + \mu) + 2\tau^2 L_K^2 \\
&\leq 1 - 2\tau\lambda + \frac{1}{6}\tau\lambda + \tau\lambda + \frac{1}{2}\tau\lambda + \frac{1}{12}\tau\lambda \\
&= 1 - \frac{\tau\lambda}{4},
\end{aligned}
$$

where in second inequality, we use $\tau\lambda \leq \frac{1}{6}, L_K = \frac{1}{2}\lambda$ and $\tau\mu \leq \tau L_K = \frac{1}{2}\tau\lambda$.

Finally, since $\tau(1 - \tau\lambda)\mu + 2\tau^2\mu^2 \leq \tau\mu + 2\tau^2\mu^2$, the proof is completed after taking full expectation of (27) and plugging in (28) and (31). ∎

**Lemma A.5** (cf. Lemma 3.3). *Let Assumption 1.1 hold. Then, if $\lambda = 2L_K$ and $\tau \leq \frac{1}{6\lambda}$, we have for the iterates of Algorithm 1 that*

$$
\begin{aligned}
\mathbb{E}\varphi_{1/\lambda}(\mathbf{x}_{t+1}, \mathbf{y}_{t+1}, \mathbf{z}_{t+1}) &\leq \mathbb{E}\varphi_{1/\lambda}(\mathbf{x}_t, \mathbf{y}_{t+1}, \mathbf{z}_{t+1}) - \frac{\tau\lambda^2}{16}\mathbb{E}\|\mathbf{u}^*(\mathbf{x}_t, \mathbf{y}_{t+1}, \mathbf{z}_t) - \mathbf{x}_t\|^2 \\
&\quad + \left(\frac{\lambda\tau\mu}{2} + \lambda\tau^2\mu^2 + \frac{\tau\lambda^2\mu^2}{8\gamma_s^2}\right)\mathbb{E}\|\mathbf{z}_t - \mathbf{z}_{t+1}\|^2 + \frac{\lambda\tau^2\sigma^2}{2},
\end{aligned} \tag{32}
$$

*where $\gamma_s = 2\mu + \rho\|A\|$.*

*Proof.* By the definition of $\varphi_{1/\lambda}$ from (24) and $\mathbf{u}^*(\mathbf{x}, \mathbf{y}_{t+1}, \mathbf{z}_{t+1})$ from (23), we have

$$
\begin{aligned}
\mathbb{E}\varphi_{1/\lambda}(\mathbf{x}_{t+1}, \mathbf{y}_{t+1}, \mathbf{z}_{t+1}) &\leq \mathbb{E}K(\mathbf{u}^*(\mathbf{x}_t, \mathbf{y}_{t+1}, \mathbf{z}_{t+1}), \mathbf{y}_{t+1}, \mathbf{z}_{t+1}) + \frac{\lambda}{2}\mathbb{E}\|\mathbf{u}^*(\mathbf{x}_t, \mathbf{y}_{t+1}, \mathbf{z}_{t+1}) - \mathbf{x}_{t+1}\|^2 \\
&\leq \mathbb{E}K(\mathbf{u}^*(\mathbf{x}_t, \mathbf{y}_{t+1}, \mathbf{z}_{t+1}), \mathbf{y}_{t+1}, \mathbf{z}_{t+1}) + \left(\frac{\lambda}{2} - \frac{\tau\lambda^2}{8}\right)\mathbb{E}\|\mathbf{u}^*(\mathbf{x}_t, \mathbf{y}_{t+1}, \mathbf{z}_{t+1}) - \mathbf{x}_t\|^2 \\
&\quad + \left(\frac{\lambda\tau\mu}{2} + \lambda\tau^2\mu^2\right)\mathbb{E}\|\mathbf{z}_t - \mathbf{z}_{t+1}\|^2 + \frac{\lambda\tau^2\sigma^2}{2} \\
&= \mathbb{E}\varphi_{1/\lambda}(\mathbf{x}_t, \mathbf{y}_{t+1}, \mathbf{z}_{t+1}) - \frac{\tau\lambda^2}{8}\mathbb{E}\|\mathbf{u}^*(\mathbf{x}_t, \mathbf{y}_{t+1}, \mathbf{z}_{t+1}) - \mathbf{x}_t\|^2 \\
&\quad + \left(\frac{\lambda\tau\mu}{2} + \lambda\tau^2\mu^2\right)\mathbb{E}\|\mathbf{z}_t - \mathbf{z}_{t+1}\|^2 + \frac{\lambda\tau^2\sigma^2}{2}.
\end{aligned}
\tag{33}
$$

We next bound the second term on the right-hand side by using Young's inequality as

$$
\begin{aligned}
\|\mathbf{u}^*(\mathbf{x}_t, \mathbf{y}_{t+1}, \mathbf{z}_{t+1}) - \mathbf{x}_t\|^2 &\geq \frac{1}{2}\|\mathbf{u}^*(\mathbf{x}_t, \mathbf{y}_{t+1}, \mathbf{z}_t) - \mathbf{x}_t\|^2 - \|\mathbf{u}^*(\mathbf{x}_t, \mathbf{y}_{t+1}, \mathbf{z}_{t+1}) - \mathbf{u}^*(\mathbf{x}_t, \mathbf{y}_{t+1}, \mathbf{z}_t)\|^2 \\
&\geq \frac{1}{2}\|\mathbf{u}^*(\mathbf{x}_t, \mathbf{y}_{t+1}, \mathbf{z}_t) - \mathbf{x}_t\|^2 - \frac{\mu^2}{\gamma_s^2}\|\mathbf{z}_t - \mathbf{z}_{t+1}\|^2,
\end{aligned}
\tag{34}
$$

where the last line used (61).

We substitute the last inequality into (33) to conclude. ∎

Since the previous result only allowed us to connect $\varphi_{1/\lambda}(\mathbf{x}_{t+1}, \mathbf{y}_{t+1}, \mathbf{z}_{t+1})$ to $\varphi_{1/\lambda}(\mathbf{x}_t, \mathbf{y}_{t+1}, \mathbf{z}_{t+1})$, we now need to analyze the effect of changing $\mathbf{y}_{t+1}$ and $\mathbf{z}_{t+1}$ in $\varphi_{1/\lambda}$. The main idea of this lemma is similar to (Zhang & Luo, 2022), where the difference lies in the fact that our potential involves the Moreau envelope of $K(\mathbf{x}, \mathbf{y}, \mathbf{z})$ whereas the potential of (Zhang & Luo, 2022) involves $K(\mathbf{x}, \mathbf{y}, \mathbf{z})$. Hence this work considers the change of the arguments in the function $K$ instead of $\varphi_{1/\lambda}$. Therefore, our proof uses the properties of the Moreau envelope which was not needed in (Zhang & Luo, 2022).

**Lemma A.6.** *(cf. Lemma 3.4) Suppose that Assumption 1.1 holds, for $\varphi_{1/\lambda}$ defined in (10), we have for the iterates of Algorithm 1 that*

$$
\begin{aligned}
\varphi_{1/\lambda}(\mathbf{x}_t, \mathbf{y}_t, \mathbf{z}_t) - \varphi_{1/\lambda}(\mathbf{x}_t, \mathbf{y}_{t+1}, \mathbf{z}_t) &\geq \langle \mathbf{y}_t - \mathbf{y}_{t+1}, A\mathbf{u}^*(\mathbf{x}_t, \mathbf{y}_t, \mathbf{z}_t) - \mathbf{b}\rangle \\
&\quad + \frac{\gamma_s}{2}\|\mathbf{u}^*(\mathbf{x}_t, \mathbf{y}_t, \mathbf{z}_t) - \mathbf{u}^*(\mathbf{x}_t, \mathbf{y}_{t+1}, \mathbf{z}_t)\|^2, \\
\varphi_{1/\lambda}(\mathbf{x}_t, \mathbf{y}_{t+1}, \mathbf{z}_t) - \varphi_{1/\lambda}(\mathbf{x}_t, \mathbf{y}_{t+1}, \mathbf{z}_{t+1}) &\geq \frac{\mu}{2}\langle \mathbf{z}_{t+1} - \mathbf{z}_t, 2\mathbf{u}^*(\mathbf{x}_t, \mathbf{y}_{t+1}, \mathbf{z}_t) - \mathbf{z}_{t+1} - \mathbf{z}_t\rangle \\
&\quad + \frac{\gamma_s}{2}\|\mathbf{u}^*(\mathbf{x}_t, \mathbf{y}_{t+1}, \mathbf{z}_{t+1}) - \mathbf{u}^*(\mathbf{x}_t, \mathbf{y}_{t+1}, \mathbf{z}_t)\|^2,
\end{aligned}
$$

*where $\gamma_s = 2\mu + \rho\|A\|$.*

*Proof.* We first consider the change in $\mathbf{y}$ argument of $\varphi_{1/\lambda}$. By using the definition of $\varphi_{1/\lambda}$ in (24), we have

$$
\begin{aligned}
\varphi_{1/\lambda}(\mathbf{x}_t, \mathbf{y}_t, \mathbf{z}_t) - \varphi_{1/\lambda}(\mathbf{x}_t, \mathbf{y}_{t+1}, \mathbf{z}_t) &= K(\mathbf{u}^*(\mathbf{x}_t, \mathbf{y}_t, \mathbf{z}_t), \mathbf{y}_t, \mathbf{z}_t) + \frac{\lambda}{2}\|\mathbf{x}_t - \mathbf{u}^*(\mathbf{x}_t, \mathbf{y}_t, \mathbf{z}_t)\|^2 \\
&\quad - K(\mathbf{u}^*(\mathbf{x}_t, \mathbf{y}_{t+1}, \mathbf{z}_t), \mathbf{y}_{t+1}, \mathbf{z}_t) - \frac{\lambda}{2}\|\mathbf{x}_t - \mathbf{u}^*(\mathbf{x}_t, \mathbf{y}_{t+1}, \mathbf{z}_t)\|^2 \\
&= K(\mathbf{u}^*(\mathbf{x}_t, \mathbf{y}_t, \mathbf{z}_t), \mathbf{y}_t, \mathbf{z}_t) - K(\mathbf{u}^*(\mathbf{x}_t, \mathbf{y}_t, \mathbf{z}_t), \mathbf{y}_{t+1}, \mathbf{z}_t) \\
&\quad + K(\mathbf{u}^*(\mathbf{x}_t, \mathbf{y}_t, \mathbf{z}_t), \mathbf{y}_{t+1}, \mathbf{z}_t) + \frac{\lambda}{2}\|\mathbf{x}_t - \mathbf{u}^*(\mathbf{x}_t, \mathbf{y}_t, \mathbf{z}_t)\|^2 \\
&\quad - K(\mathbf{u}^*(\mathbf{x}_t, \mathbf{y}_{t+1}, \mathbf{z}_t), \mathbf{y}_{t+1}, \mathbf{z}_t) - \frac{\lambda}{2}\|\mathbf{x}_t - \mathbf{u}^*(\mathbf{x}_t, \mathbf{y}_{t+1}, \mathbf{z}_t)\|^2,
\end{aligned}
\tag{35}
$$

where the second equality adds and subtracts $K(\mathbf{u}^*(\mathbf{x}_t, \mathbf{y}_t, \mathbf{z}_t), \mathbf{y}_{t+1}, \mathbf{z}_t)$.

From the definition of $K$ in (22), it trivially follows that

$$K(\mathbf{u}^*(\mathbf{x}_t, \mathbf{y}_t, \mathbf{z}_t), \mathbf{y}_t, \mathbf{z}_t) - K(\mathbf{u}^*(\mathbf{x}_t, \mathbf{y}_t, \mathbf{z}_t), \mathbf{y}_{t+1}, \mathbf{z}_t) = \langle \mathbf{y}_t - \mathbf{y}_{t+1}, A\mathbf{u}^*(\mathbf{x}_t, \mathbf{y}_t, \mathbf{z}_t) - \mathbf{b} \rangle.$$

Next, we use the property that $K(\cdot, \mathbf{y}_{t+1}, \mathbf{z}_t) + \frac{\lambda}{2}\|\cdot -\mathbf{x}_t\|^2$ is $\gamma_s$-strongly convex with minimizer $\mathbf{u}^*(\mathbf{x}_t, \mathbf{y}_{t+1}, \mathbf{z}_t)$ (see Fact A.1 and (23)) to obtain

$$K(\mathbf{u}^*(\mathbf{x}_t, \mathbf{y}_t, \mathbf{z}_t), \mathbf{y}_{t+1}, \mathbf{z}_t) + \frac{\lambda}{2}\|\mathbf{x}_t - \mathbf{u}^*(\mathbf{x}_t, \mathbf{y}_t, \mathbf{z}_t)\|^2 - K(\mathbf{u}^*(\mathbf{x}_t, \mathbf{y}_{t+1}, \mathbf{z}_t), \mathbf{y}_{t+1}, \mathbf{z}_t) - \frac{\lambda}{2}\|\mathbf{x}_t - \mathbf{u}^*(\mathbf{x}_t, \mathbf{y}_{t+1}, \mathbf{z}_t)\|^2$$
$$\geq \frac{\gamma_s}{2}\|\mathbf{u}^*(\mathbf{x}_t, \mathbf{y}_t, \mathbf{z}_t) - \mathbf{u}^*(\mathbf{x}_t, \mathbf{y}_{t+1}, \mathbf{z}_t)\|^2.$$

Combining the last two estimates in (35) gives the first assertion.

Next, we analyze the effect of changing the $\mathbf{z}$ component in $\varphi_{1/\lambda}$. Similar to the proof of the first assertion, we start with the definition of $\varphi_{1/\lambda}$ and then add and subtract $K(\mathbf{u}^*(\mathbf{x}_t, \mathbf{y}_{t+1}, \mathbf{z}_{t+1})$ to obtain

$$\varphi_{1/\lambda}(\mathbf{x}_t, \mathbf{y}_{t+1}, \mathbf{z}_t) - \varphi_{1/\lambda}(\mathbf{x}_t, \mathbf{y}_{t+1}, \mathbf{z}_{t+1})$$
$$= K(\mathbf{u}^*(\mathbf{x}_t, \mathbf{y}_{t+1}, \mathbf{z}_t), \mathbf{y}_{t+1}, \mathbf{z}_t) + \frac{\lambda}{2}\|\mathbf{x}_t - \mathbf{u}^*(\mathbf{x}_t, \mathbf{y}_{t+1}, \mathbf{z}_t)\|^2$$
$$\quad - K(\mathbf{u}^*(\mathbf{x}_t, \mathbf{y}_{t+1}, \mathbf{z}_{t+1}), \mathbf{y}_{t+1}, \mathbf{z}_{t+1}) - \frac{\lambda}{2}\|\mathbf{x}_t - \mathbf{u}^*(\mathbf{x}_t, \mathbf{y}_{t+1}, \mathbf{z}_{t+1})\|^2$$
$$= K(\mathbf{u}^*(\mathbf{x}_t, \mathbf{y}_{t+1}, \mathbf{z}_t), \mathbf{y}_{t+1}, \mathbf{z}_t) - K(\mathbf{u}^*(\mathbf{x}_t, \mathbf{y}_{t+1}, \mathbf{z}_t), \mathbf{y}_{t+1}, \mathbf{z}_{t+1})$$
$$\quad + K(\mathbf{u}^*(\mathbf{x}_t, \mathbf{y}_{t+1}, \mathbf{z}_t), \mathbf{y}_{t+1}, \mathbf{z}_{t+1}) + \frac{\lambda}{2}\|\mathbf{x}_t - \mathbf{u}^*(\mathbf{x}_t, \mathbf{y}_{t+1}, \mathbf{z}_t)\|^2$$
$$\quad - K(\mathbf{u}^*(\mathbf{x}_t, \mathbf{y}_{t+1}, \mathbf{z}_{t+1}), \mathbf{y}_{t+1}, \mathbf{z}_{t+1}) - \frac{\lambda}{2}\|\mathbf{x}_t - \mathbf{u}^*(\mathbf{x}_t, \mathbf{y}_{t+1}, \mathbf{z}_{t+1})\|^2. \tag{36}$$

First, by definition, of $K$, it trivially follows that

$$K(\mathbf{u}^*(\mathbf{x}_t, \mathbf{y}_{t+1}, \mathbf{z}_t), \mathbf{y}_{t+1}, \mathbf{z}_t) - K(\mathbf{u}^*(\mathbf{x}_t, \mathbf{y}_{t+1}, \mathbf{z}_t), \mathbf{y}_{t+1}, \mathbf{z}_{t+1}) = \frac{\mu}{2}\|\mathbf{u}^*(\mathbf{x}_t, \mathbf{y}_{t+1}, \mathbf{z}_t) - \mathbf{z}_t\|^2$$
$$- \frac{\mu}{2}\|\mathbf{u}^*(\mathbf{x}_t, \mathbf{y}_{t+1}, \mathbf{z}_t) - \mathbf{z}_{t+1}\|^2.$$

For the remaining terms on the right-hand side, we again use that $K(\cdot, \mathbf{y}_{t+1}, \mathbf{z}_{t+1}) + \frac{\lambda}{2}\|\cdot -\mathbf{x}_t\|^2$ is $\gamma_s$-strongly convex with minimizer $\mathbf{u}^*(\mathbf{x}_t, \mathbf{y}_{t+1}, \mathbf{z}_{t+1})$ to deduce

$$K(\mathbf{u}^*(\mathbf{x}_t, \mathbf{y}_{t+1}, \mathbf{z}_t), \mathbf{y}_{t+1}, \mathbf{z}_{t+1}) + \frac{\lambda}{2}\|\mathbf{x}_t - \mathbf{u}^*(\mathbf{x}_t, \mathbf{y}_{t+1}, \mathbf{z}_t)\|^2$$
$$- K(\mathbf{u}^*(\mathbf{x}_t, \mathbf{y}_{t+1}, \mathbf{z}_{t+1}), \mathbf{y}_{t+1}, \mathbf{z}_{t+1}) - \frac{\lambda}{2}\|\mathbf{x}_t - \mathbf{u}^*(\mathbf{x}_t, \mathbf{y}_{t+1}, \mathbf{z}_{t+1})\|^2$$
$$\geq \frac{\gamma_s}{2}\|\mathbf{u}^*(\mathbf{x}_t, \mathbf{y}_{t+1}, \mathbf{z}_{t+1}) - \mathbf{u}^*(\mathbf{x}_t, \mathbf{y}_{t+1}, \mathbf{z}_t)\|^2.$$

Plugging in the last two estimates in (36) gives the second assertion. ∎

**Corollary A.7.** *Suppose that Assumption 1.1 holds, for $\varphi_{1/\lambda}$ defined in (10), if $\lambda = 2L_K$ and $\tau \leq \frac{1}{6\lambda}$, we have that*

$$\mathbb{E}\varphi_{1/\lambda}(\mathbf{x}_t, \mathbf{y}_t, \mathbf{z}_t) - \mathbb{E}\varphi_{1/\lambda}(\mathbf{x}_{t+1}, \mathbf{y}_{t+1}, \mathbf{z}_{t+1}) \geq \frac{\tau\lambda^2}{16}\mathbb{E}\|\mathbf{u}^*(\mathbf{x}_t, \mathbf{y}_{t+1}, \mathbf{z}_t) - \mathbf{x}_t\|^2$$
$$- \left(\frac{\lambda\tau\mu}{2} + \lambda\tau^2\mu^2 + \frac{\tau\lambda^2\mu^2}{8\gamma_s^2}\right)\mathbb{E}\|\mathbf{z}_t - \mathbf{z}_{t+1}\|^2 - \frac{\lambda\tau^2\sigma^2}{2}$$
$$- \eta\mathbb{E}\langle A\mathbf{x}_t - \mathbf{b}, A\mathbf{u}^*(\mathbf{x}_t, \mathbf{y}_t, \mathbf{z}_t) - b\rangle$$
$$+ \frac{\mu}{2}\mathbb{E}\langle \mathbf{z}_{t+1} - \mathbf{z}_t, 2\mathbf{u}^*(\mathbf{x}_t, \mathbf{y}_{t+1}, \mathbf{z}_t) - \mathbf{z}_{t+1} - \mathbf{z}_t\rangle,$$

*where $\gamma_s = 2\mu + \rho\|A\|$.*

*Proof.* We sum up the results in Lemma A.5 and Lemma A.6, plug in the definition of $\mathbf{y}_{t+1}$ and discard two nonnegative terms on the right-hand side to get the result. ∎

Next, we analyze the rest of the terms appearing in the potential function. This lemma is only using the definition of $d(\mathbf{y}, \mathbf{z})$ and $\Psi(\mathbf{z})$ and is equivalent to (Zhang & Luo, 2022) and hence we omit its proof. Notably, these bounds are agnostic to the algorithm used to generate the sequences. Note that the only difference is that in the result below, we do not use the definition of $\mathbf{y}_{t+1}$ whereas the proof in (Zhang & Luo, 2022) uses this definition. The rest of the estimations are precisely the same.

**Lemma A.8.** *(Zhang & Luo, 2020, Lemma 3.2, Lemma 3.3) For the functions $d(\mathbf{y}, \mathbf{z})$ and $\Psi(\mathbf{z})$ defined in (12) and (7),we have*

$$d(\mathbf{y}_{t+1}, \mathbf{z}_{t+1}) - d(\mathbf{y}_t, \mathbf{z}_t) \geq \eta\langle A\mathbf{x}_t - \mathbf{b}, A\mathbf{x}^*(\mathbf{y}_{t+1}, \mathbf{z}_t) - \mathbf{b}\rangle + \frac{\mu}{2}\langle \mathbf{z}_{t+1} - \mathbf{z}_t, \mathbf{z}_{t+1} + \mathbf{z}_t - 2\mathbf{x}^*(\mathbf{y}_{t+1}, \mathbf{z}_{t+1})\rangle,$$

$$\Psi(\mathbf{z}_{t+1}) - \Psi(\mathbf{z}_t) \leq \mu\langle \mathbf{z}_{t+1} - \mathbf{z}_t, \mathbf{z}_t - \bar{\mathbf{x}}^*(\mathbf{z}_t)\rangle + \frac{\mu}{2\sigma_4}\|\mathbf{z}_t - \mathbf{z}_{t+1}\|^2,$$

*where $\sigma_4$ is defined in* (25).

In the next lemma, we will join the previous lemmas and characterize the change in the potential function.

**Lemma A.9** (cf. Lemma 3.6)**.** *Let Assumption 1.1 hold. By using the parameters* (25) *in Algorithm 1, we obtain*

$$\mathbb{E}V_t - \mathbb{E}V_{t+1} \geq c_\beta\mathbb{E}\|\mathbf{z}_{t+1} - \mathbf{z}_t\|^2 + c_\tau\mathbb{E}\|\mathbf{u}^*(\mathbf{x}_t, \mathbf{y}_{t+1}, \mathbf{z}_t) - \mathbf{x}_t\|^2 + c_\eta\mathbb{E}\|A\mathbf{x}^*(\mathbf{y}_{t+1}, \mathbf{z}_t) - \mathbf{b}\|^2 - \frac{1}{2}\lambda\tau^2\sigma^2, \quad (37)$$

*where $c_\beta = \frac{\mu}{50\beta}$, $c_\tau = \frac{7\tau\lambda^2}{400}$, $c_\eta = \frac{\eta}{4}$.*

*Proof.* Combining Corollary A.7 and Lemma A.8, we obtain

$$\mathbb{E}[V_t - V_{t+1}] = \mathbb{E}\left[\varphi_{1/\lambda}(\mathbf{x}_t, \mathbf{y}_t, \mathbf{z}_t) - \varphi_{1/\lambda}(\mathbf{x}_{t+1}, \mathbf{y}_{t+1}, \mathbf{z}_{t+1}) + 2d(\mathbf{y}_{t+1}, \mathbf{z}_{t+1}) - 2d(\mathbf{y}_t, \mathbf{z}_t) + 2\Psi(\mathbf{z}_t) - 2\Psi(\mathbf{z}_{t+1})\right]$$

$$\geq \frac{\tau\lambda^2}{16}\mathbb{E}\|\mathbf{u}^*(\mathbf{x}_t, \mathbf{y}_{t+1}, \mathbf{z}_t) - \mathbf{x}_t\|^2 - \left(\frac{\lambda\tau\mu}{2} + \lambda\tau^2\mu^2 + \frac{\tau\lambda^2\mu^2}{8\gamma_s^2}\right)\mathbb{E}\|\mathbf{z}_t - \mathbf{z}_{t+1}\|^2 - \frac{\lambda\tau^2\sigma^2}{2}$$

$$- \eta\mathbb{E}\langle A\mathbf{x}_t - \mathbf{b}, A\mathbf{u}^*(\mathbf{x}_t, \mathbf{y}_t, \mathbf{z}_t) - \mathbf{b}\rangle + \frac{\mu}{2}\mathbb{E}\langle \mathbf{z}_{t+1} - \mathbf{z}_t, 2\mathbf{u}^*(\mathbf{x}_t, \mathbf{y}_{t+1}, \mathbf{z}_t) - \mathbf{z}_t - \mathbf{z}_{t+1}\rangle$$

$$+ 2\eta\mathbb{E}\langle A\mathbf{x}_t - \mathbf{b}, A\mathbf{x}^*(\mathbf{y}_{t+1}, \mathbf{z}_t) - \mathbf{b}\rangle + \mu\mathbb{E}\langle \mathbf{z}_{t+1} - \mathbf{z}_t, \mathbf{z}_{t+1} + \mathbf{z}_t - 2\mathbf{x}^*(\mathbf{y}_{t+1}, \mathbf{z}_{t+1})\rangle$$

$$- 2\mu\mathbb{E}\langle \mathbf{z}_{t+1} - \mathbf{z}_t, \mathbf{z}_t - \bar{\mathbf{x}}^*(\mathbf{z}_t)\rangle - \frac{\mu}{\sigma_4}\mathbb{E}\|\mathbf{z}_t - \mathbf{z}_{t+1}\|^2. \quad (38)$$

We next manipulate the terms on the right-hand side. First, by adding and subtracting $A\mathbf{x}_t$ on the second argument of the first inner product on the right-hand side, we get

$$-\eta\langle A\mathbf{x}_t - \mathbf{b}, A\mathbf{u}^*(\mathbf{x}_t, \mathbf{y}_t, \mathbf{z}_t) - \mathbf{b}\rangle = -\eta\|A\mathbf{x}_t - \mathbf{b}\|^2 - \eta\langle A\mathbf{x}_t - \mathbf{b}, A\mathbf{u}^*(\mathbf{x}_t, \mathbf{y}_t, \mathbf{z}_t) - A\mathbf{x}_t\rangle.$$

Consequently, we use this estimate and rewrite the third inner product on the right-hand side of (38) with quadratics to have

$$- \eta\langle A\mathbf{x}_t - \mathbf{b}, A\mathbf{u}^*(\mathbf{x}_t, \mathbf{y}_t, \mathbf{z}_t) - \mathbf{b}\rangle + 2\eta\langle A\mathbf{x}_t - \mathbf{b}, A\mathbf{x}^*(\mathbf{y}_{t+1}, \mathbf{z}_t) - \mathbf{b}\rangle$$

$$= -\eta\|A\mathbf{x}_t - A\mathbf{x}^*(\mathbf{y}_{t+1}, \mathbf{z}_t)\|^2 + \eta\|A\mathbf{x}^*(\mathbf{y}_{t+1}, \mathbf{z}_t) - \mathbf{b}\|^2 - \eta\langle A\mathbf{x}_t - \mathbf{b}, A\mathbf{u}^*(\mathbf{x}_t, \mathbf{y}_t, \mathbf{z}_t) - A\mathbf{x}_t\rangle.$$

Second, adding and subtracting $2\mathbf{x}_t$ in the second argument of the second inner product on the right-hand side of (38) gives

$$\frac{\mu}{2}\langle \mathbf{z}_{t+1} - \mathbf{z}_t, 2\mathbf{u}^*(\mathbf{x}_t, \mathbf{y}_{t+1}, \mathbf{z}_t) - \mathbf{z}_t - \mathbf{z}_{t+1}\rangle = \frac{\mu}{2}\langle \mathbf{z}_{t+1} - \mathbf{z}_t, 2\mathbf{u}^*(\mathbf{x}_t, \mathbf{y}_{t+1}, \mathbf{z}_t) - 2\mathbf{x}_t\rangle + \frac{\mu}{2}\langle \mathbf{z}_{t+1} - \mathbf{z}_t, 2\mathbf{x}_t - \mathbf{z}_t - \mathbf{z}_{t+1}\rangle.$$

For the right-hand side of this term, note that $\mathbf{z}_{t+1} = \mathbf{z}_t + \beta(\mathbf{x}_t - \mathbf{z}_t) \iff 2\mathbf{x}_t - 2\mathbf{z}_t = \frac{2}{\beta}(\mathbf{z}_{t+1} - \mathbf{z}_t)$ and hence

$$\frac{\mu}{2}\langle \mathbf{z}_{t+1} - \mathbf{z}_t, 2\mathbf{x}_t - \mathbf{z}_t - \mathbf{z}_{t+1}\rangle = \frac{\mu}{2}\langle \mathbf{z}_{t+1} - \mathbf{z}_t, 2\mathbf{x}_t - 2\mathbf{z}_t\rangle + \frac{\mu}{2}\langle \mathbf{z}_{t+1} - \mathbf{z}_t, \mathbf{z}_t - \mathbf{z}_{t+1}\rangle$$

$$= \frac{\mu}{2}\left(\frac{2}{\beta} - 1\right)\|\mathbf{z}_t - \mathbf{z}_{t+1}\|^2 \geq \frac{\mu}{2\beta}\|\mathbf{z}_t - \mathbf{z}_{t+1}\|^2,$$

where the last inequality is due to $\beta \leq 1$.

Next, for the remaining inner products in (38), we have

$$
\begin{aligned}
&\mu\langle \mathbf{z}_{t+1} - \mathbf{z}_t, \mathbf{z}_{t+1} + \mathbf{z}_t - 2\mathbf{x}^*(\mathbf{y}_{t+1}, \mathbf{z}_{t+1})\rangle - 2\mu\langle \mathbf{z}_{t+1} - \mathbf{z}_t, \mathbf{z}_t - \bar{\mathbf{x}}^*(\mathbf{z}_t)\rangle \\
&= \mu\|\mathbf{z}_{t+1} - \mathbf{z}_t\|^2 + 2\mu\langle \mathbf{z}_{t+1} - \mathbf{z}_t, \bar{\mathbf{x}}^*(\mathbf{z}_t) - \mathbf{x}^*(\mathbf{y}_{t+1}, \mathbf{z}_{t+1})\rangle.
\end{aligned}
\tag{39}
$$

We can use Cauchy-Schwarz, triangle and Young's inequalities on the second term here to get

$$
\begin{aligned}
\langle \mathbf{z}_{t+1} - \mathbf{z}_t, \bar{\mathbf{x}}^*(\mathbf{z}_t) - \mathbf{x}^*(\mathbf{y}_{t+1}, \mathbf{z}_{t+1})\rangle &\geq -\|\mathbf{z}_{t+1} - \mathbf{z}_t\| (\|\bar{\mathbf{x}}^*(\mathbf{z}_t) - \mathbf{x}^*(\mathbf{y}_{t+1}, \mathbf{z}_t)\| + \|\mathbf{x}^*(\mathbf{y}_{t+1}, \mathbf{z}_t) - \mathbf{x}^*(\mathbf{y}_{t+1}, \mathbf{z}_{t+1})\|) \\
&\geq -\left(\frac{1}{2\zeta} + \frac{1}{\sigma_4}\right)\|\mathbf{z}_{t+1} - \mathbf{z}_t\|^2 - \frac{\zeta}{2}\|\bar{\mathbf{x}}^*(\mathbf{z}_t) - \mathbf{x}^*(\mathbf{y}_{t+1}, \mathbf{z}_t)\|^2,
\end{aligned}
$$

where the last step also used (63). Consequently, plugging in this estimate to (39), we obtain

$$
\begin{aligned}
&\mu\langle \mathbf{z}_{t+1} - \mathbf{z}_t, \mathbf{z}_{t+1} + \mathbf{z}_t - 2\mathbf{x}^*(\mathbf{y}_{t+1}, \mathbf{z}_{t+1})\rangle - 2\mu\langle \mathbf{z}_{t+1} - \mathbf{z}_t, \mathbf{z}_t - \bar{\mathbf{x}}^*(\mathbf{z}_t)\rangle \\
&\geq \left(\mu - \frac{\mu}{\zeta} - \frac{2\mu}{\sigma_4}\right)\|\mathbf{z}_{t+1} - \mathbf{z}_t\|^2 - \mu\zeta\|\bar{\mathbf{x}}^*(\mathbf{z}_t) - \mathbf{x}^*(\mathbf{y}_{t+1}, \mathbf{z}_t)\|^2.
\end{aligned}
$$

After combining these estimates in (38), we get

$$
\begin{aligned}
&\mathbb{E}[V_t] - \mathbb{E}[V_{t+1}] \\
&\geq \frac{\tau\lambda^2}{16}\mathbb{E}\|\mathbf{u}^*(\mathbf{x}_t, \mathbf{y}_{t+1}, \mathbf{z}_t) - \mathbf{x}_t\|^2 - \left(\frac{1}{2}\lambda\tau\mu + \lambda\tau^2\mu^2 + \frac{\tau\lambda^2\mu^2}{8\gamma_s^2} + \frac{\mu}{\zeta} + \frac{3\mu}{\sigma_4} - \mu - \frac{\mu}{2\beta}\right)\mathbb{E}\|\mathbf{z}_t - \mathbf{z}_{t+1}\|^2 - \frac{1}{2}\lambda\tau^2\sigma^2 \\
&\quad - \eta\mathbb{E}\langle A\mathbf{x}_t - \mathbf{b}, A\mathbf{u}^*(\mathbf{x}_t, \mathbf{y}_t, \mathbf{z}_t) - A\mathbf{x}_t\rangle - \eta\mathbb{E}\|A\mathbf{x}_t - A\mathbf{x}^*(\mathbf{y}_{t+1}, \mathbf{z}_t)\|^2 + \eta\mathbb{E}\|A\mathbf{x}^*(\mathbf{y}_{t+1}, \mathbf{z}_t) - \mathbf{b}\|^2 \\
&\quad - \mu\zeta\mathbb{E}\|\bar{\mathbf{x}}^*(\mathbf{z}_t) - \mathbf{x}^*(\mathbf{y}_{t+1}, \mathbf{z}_t)\|^2 + \mu\mathbb{E}\langle \mathbf{z}_{t+1} - \mathbf{z}_t, \mathbf{u}^*(\mathbf{x}_t, \mathbf{y}_{t+1}, \mathbf{z}_t) - \mathbf{x}_t\rangle.
\end{aligned}
\tag{40}
$$

We will now operate on some of terms from the right-hand side of (40), by using Lemma A.11 and A.12. First, we have by Cauchy-Schwarz and Young's inequalities that

$$
\begin{aligned}
&- \eta\langle A\mathbf{x}_t - \mathbf{b}, A\mathbf{u}^*(\mathbf{x}_t, \mathbf{y}_t, \mathbf{z}_t) - A\mathbf{x}_t\rangle \\
&\geq -\frac{\eta}{4}\|A\mathbf{x}_t - \mathbf{b}\|^2 - \eta\|A\mathbf{u}^*(\mathbf{x}_t, \mathbf{y}_t, \mathbf{z}_t) - A\mathbf{x}_t\|^2 \\
&\geq -\frac{\eta}{4}\|A\mathbf{x}_t - \mathbf{b}\|^2 - 2\eta\|A\mathbf{u}^*(\mathbf{x}_t, \mathbf{y}_t, \mathbf{z}_t) - A\mathbf{u}^*(\mathbf{x}_t, \mathbf{y}_{t+1}, \mathbf{z}_t)\|^2 - 2\eta\|A\mathbf{u}^*(\mathbf{x}_t, \mathbf{y}_{t+1}, \mathbf{z}_t) - A\mathbf{x}_t\|^2.
\end{aligned}
$$

Next, by using the Lipschitzness of $\mathbf{u}^*(\mathbf{x}_t, \cdot, \mathbf{z}_t)$ from (60), we have

$$
\begin{aligned}
\|A\mathbf{u}^*(\mathbf{x}_t, \mathbf{y}_t, \mathbf{z}_t) - A\mathbf{u}^*(\mathbf{x}_t, \mathbf{y}_{t+1}, \mathbf{z}_t)\|^2 &\leq \|A\|^2\|\mathbf{u}^*(\mathbf{x}_t, \mathbf{y}_t, \mathbf{z}_t) - \mathbf{u}^*(\mathbf{x}_t, \mathbf{y}_{t+1}, \mathbf{z}_t)\|^2 \\
&\leq \frac{\|A\|^4}{\gamma_s^2}\|\mathbf{y}_t - \mathbf{y}_{t+1}\|^2 \\
&= \frac{\|A\|^4\eta^2}{\gamma_s^2}\|A\mathbf{x}_t - \mathbf{b}\|^2,
\end{aligned}
$$

where the last step also used the definition of $\mathbf{y}_{t+1}$. Using this estimation along with (66) gives

$$
\begin{aligned}
&- \eta\langle A\mathbf{x}_t - \mathbf{b}, A\mathbf{u}^*(\mathbf{x}_t, \mathbf{y}_t, \mathbf{z}_t) - A\mathbf{x}_t\rangle \\
&\geq -\left(\frac{\eta}{4} + \frac{2\|A\|^4\eta^3}{\gamma_s^2}\right)\|A\mathbf{x}_t - \mathbf{b}\|^2 - 2\eta\|A\|^2\|\mathbf{u}^*(\mathbf{x}_t, \mathbf{y}_{t+1}, \mathbf{z}_t) - \mathbf{x}_t\|^2 \\
&\geq -\left(\frac{\eta\|A\|^2\lambda^2}{2\gamma^2} + \frac{4\|A\|^6\eta^3\lambda^2}{\gamma^2\gamma_s^2} + 2\eta\|A\|^2\right)\|\mathbf{u}^*(\mathbf{x}_t, \mathbf{y}_{t+1}, \mathbf{z}_t) - \mathbf{x}_t\|^2 \\
&\quad - \left(\frac{\eta}{2} + \frac{4\|A\|^4\eta^3}{\gamma_s^2}\right)\|A\mathbf{x}^*(\mathbf{y}_{t+1}, \mathbf{z}_t) - \mathbf{b}\|^2.
\end{aligned}
$$

We next have by Young's inequality that for any $\theta > 0$:

$$\mu\langle \mathbf{z}_{t+1} - \mathbf{z}_t, \mathbf{u}^*(\mathbf{x}_t, \mathbf{y}_{t+1}, \mathbf{z}_t) - \mathbf{x}_t\rangle \geq -\frac{\mu}{4\theta}\|\mathbf{z}_{t+1} - \mathbf{z}_t\|^2 - \theta\mu\|\mathbf{u}^*(\mathbf{x}_t, \mathbf{y}_{t+1}, \mathbf{z}_t) - \mathbf{x}_t\|^2.$$

The inequality derived in (65) directly implies

$$-\eta\|A\mathbf{x}_t - A\mathbf{x}^*(\mathbf{y}_{t+1}, \mathbf{z}_t)\|^2 \geq -\frac{\eta\|A\|^2\lambda^2}{\gamma^2}\|\mathbf{x}_t - \mathbf{u}^*(\mathbf{x}_t, \mathbf{y}_{t+1}, \mathbf{z}_t)\|^2.$$

The key global error bound given in Lemma A.12 originally proved in (Zhang & Luo, 2022) results in

$$-6\mu\beta\|\mathbf{x}^*(\mathbf{y}_{t+1}, \mathbf{z}_t) - \bar{\mathbf{x}}^*(\mathbf{z}_t)\|^2 \geq -6\mu\beta\bar{\sigma}^2\|A\mathbf{x}^*(\mathbf{y}_{t+1}, \mathbf{z}_t) - \mathbf{b}\|^2.$$

Combining these estimates in (40) leads to

$$
\begin{aligned}
\mathbb{E}[V_t] - \mathbb{E}[V_{t+1}] \geq &- \left(\frac{1}{2}\lambda\tau\mu + \lambda\tau^2\mu^2 + \frac{\tau\lambda^2\mu^2}{8\gamma_s^2} + \frac{\mu}{\zeta} + \frac{3\mu}{\sigma_4} - \mu - \frac{\mu}{2\beta} + \frac{\mu}{4\theta}\right)\mathbb{E}\|\mathbf{z}_t - \mathbf{z}_{t+1}\|^2 - \frac{1}{2}\lambda\tau^2\sigma^2 \\
&+ \left(\frac{\tau\lambda^2}{16} - \frac{3\|A\|^2\lambda^2\eta}{2\gamma^2} - \frac{4\|A\|^6\eta^3\lambda^2}{\gamma_s^2\gamma^2} - 2\eta\|A\|^2 - \mu\theta\right)\mathbb{E}\|\mathbf{u}^*(\mathbf{x}_t, \mathbf{y}_{t+1}, \mathbf{z}_t) - \mathbf{x}_t\|^2 \\
&+ \left(\frac{\eta}{2} - \frac{4\|A\|^4\eta^3}{\gamma_s^2} - 6\mu\beta\bar{\sigma}^2\right)\mathbb{E}\|A\mathbf{x}^*(\mathbf{y}_{t+1}, \mathbf{z}_t) - \mathbf{b}\|^2.
\end{aligned}
\tag{41}
$$

We now estimate the coefficients inside the parantheses, with straightforward but tedious calculations which follow from the parameter settings.

First, we estimate the coefficient of $\mathbb{E}\|\mathbf{z}_t - \mathbf{z}_{t+1}\|^2$ in (41): Let $\mu \geq 4L_f$, we have $\sigma_4 \geq \frac{1}{2}$ because $\sigma_4 = \frac{\mu - L_f}{\mu}$. Then letting $\zeta = 6\beta, \beta < \frac{1}{30}$, we have

$$\mu - \frac{3\mu}{\sigma_4} \geq -5\mu \geq -\frac{\mu}{6\beta}, \quad \frac{\mu}{\zeta} = \frac{\mu}{6\beta}.$$

Therefore, we have that

$$\frac{\mu}{2\beta} + \mu - \frac{3\mu}{\sigma_4} - \frac{\mu}{\zeta} \geq \left(\frac{1}{2} - \frac{1}{6} - \frac{1}{6}\right)\frac{\mu}{\beta} \geq \frac{\mu}{6\beta}. \tag{42}$$

Hence, we estimate:

$$\text{coefficient of } \mathbb{E}\|\mathbf{z}_t - \mathbf{z}_{t+1}\| \geq -\frac{1}{2}\lambda\tau\mu - \lambda\tau^2\mu^2 - \frac{\tau\lambda^2\mu^2}{8\gamma_s^2} + \frac{\mu}{6\beta} - \frac{\mu}{8\beta}.$$

Let $\eta = \frac{\eta'}{2\|A\|^2}, \theta = 2\beta, \eta' \leq \frac{1}{40}$, and $\mu = \max\{2, 4L_f\}, \lambda = 2L_K = 2(L_f + \rho\|A\| + \mu), \tau \leq \frac{1}{10\lambda^2}$, and $\gamma_s = \mu - L_f + \gamma$ from Fact A.1. We have $-\lambda\tau\mu \geq -\frac{\mu}{10}$ and $-2\lambda\tau^2\mu^2 \geq -\frac{\mu}{100}$, then

$$\text{coefficient of } \mathbb{E}\|\mathbf{z}_t - \mathbf{z}_{t+1}\| \geq \frac{\mu}{24\beta} - \frac{\mu}{20} - \frac{\mu}{100} - \tau\lambda^2\frac{\mu^2}{(\mu - L_f + \lambda)^2}.$$

By $\beta \leq 1/30$, we have $\frac{1}{24\beta} - \frac{1}{20} - \frac{1}{100} \geq \frac{1}{30\beta}$. In addition, using $\tau\lambda^2\frac{\mu^2}{(\mu - L_f + \lambda)^2} \leq \tau\lambda^2 \leq \frac{1}{10}$, we fanally obtain:

$$\text{coefficient of } \mathbb{E}\|\mathbf{z}_t - \mathbf{z}_{t+1}\| \geq \frac{\mu}{30\beta} - \frac{1}{10} \overset{\mu \geq 2}{\geq} \frac{\mu}{50\beta}. \tag{43}$$

Then we estimate the coefficient of $\mathbb{E}\|\mathbf{u}^*(\mathbf{x}_t, \mathbf{y}_{t+1}, \mathbf{z}_t) - \mathbf{x}_t\|^2$ in (41).

From above assumptions, we can easily get $\gamma = \frac{(\mu - L_f)\lambda}{\mu - L_f + \lambda} \geq \frac{1}{2}$ because $\lambda \geq \mu \geq 2$. Moreover, we assume $\eta' \leq \frac{\tau}{40}, \frac{\eta'}{\mu - L_f + \lambda} \leq \frac{\tau}{10}, \beta \leq \frac{\tau}{40}$ First, by our new notations, we have

$$\text{coefficient of } \mathbb{E}\|\mathbf{u}^*(\mathbf{x}_t, \mathbf{y}_{t+1}, \mathbf{z}_t) - \mathbf{x}_t\|^2 = \frac{\tau\lambda^2}{16} - \frac{3\eta'\lambda^2}{4\gamma^2} - \frac{\eta'^3\lambda^2}{2\gamma^2\gamma_s^2} - \eta' - 2\mu\beta$$

By $\gamma \geq \frac{1}{2}$ and the definition of $\gamma_s$, we have $-\frac{3\eta'\lambda^2}{4\gamma^2} \geq -3\eta'\lambda^2$, $-\frac{\eta'^3\lambda^2}{2\gamma^2\gamma_s^2} \geq -\frac{\eta'^3\lambda^2}{(\mu-L_f+\lambda)^2}$, Then

$$\text{coefficient of } \mathbb{E}\|\mathbf{u}^*(\mathbf{x}_t, \mathbf{y}_{t+1}, \mathbf{z}_t) - \mathbf{x}_t\|^2 \geq \frac{\tau\lambda^2}{16} - 3\eta'\lambda^2 - \frac{2\eta'^3\lambda^2}{(\mu-L_f+\lambda)^2} - \eta' - 2\mu\beta.$$

With $2 \leq \mu \leq \lambda$, $\eta' \leq \frac{\tau}{100}$, $\frac{\eta'}{\mu-L_f+\lambda} \leq \frac{\tau}{10}$, $\beta \leq \frac{\tau}{200}$, we can obtain $-3\eta'\lambda^2 \geq -\frac{3\tau\lambda^2}{400}$, $-\frac{2\eta'^3\lambda^2}{(\mu-L_f+\lambda)^2} \geq -\frac{\lambda^2\tau^2}{400} \geq -\frac{\lambda^2\tau}{400}$, $-\eta' \geq -\frac{\tau}{100} \geq -\frac{\tau\lambda^2}{100}$, $-2\mu\beta \geq -\frac{\tau\mu}{50} \overset{\mu\leq\lambda}{\geq} -\frac{\tau\lambda}{50} \geq -\frac{\tau\lambda^2}{100}$. Hence,

$$\text{coefficient of } \mathbb{E}\|\mathbf{u}^*(\mathbf{x}_t, \mathbf{y}_{t+1}, \mathbf{z}_t) - \mathbf{x}_t\|^2 \geq \frac{\tau\lambda^2}{16} - \frac{3\tau\lambda^2}{100} - \frac{\tau\lambda^2}{400} - \frac{\tau\lambda^2}{400} - \frac{\tau\lambda^2}{100} = \frac{7\tau\lambda^2}{400}. \tag{44}$$

Last, we estimate the coefficient of $\mathbb{E}\|A\mathbf{x}^*(\mathbf{y}_{t+1}, \mathbf{z}_t) - \mathbf{b}\|^2$ in (41). By $6\mu\beta\bar{\sigma}^2 \leq \frac{\eta}{6}$ and the definition $\eta', \gamma_s$, we have

$-\frac{4\|A\|^2\eta^3}{\gamma_s^2} = -\frac{\eta'^2\eta}{(\mu-L_f+\lambda)^2} \overset{\frac{\eta'}{\mu-L_f+\lambda}\leq\frac{\tau}{10}}{\geq} -\frac{\eta\tau^2}{100} \geq -\frac{\eta}{100}$ and $-6\mu\beta\bar{\sigma}^2 \geq -\frac{\eta}{6}$. Hence, we have

$$\text{coefficient of } \mathbb{E}\|A\mathbf{x}^*(\mathbf{y}_{t+1}, \mathbf{z}_t) - \mathbf{b}\|^2 \geq \frac{\eta}{2} - \frac{\eta}{100} - \frac{\eta}{6} \geq \frac{\eta}{4}. \tag{45}$$

Plugging (43), (44) and (45) to (41), we finish the proof. ∎

### A.2. Proof of Theorem 3.1

*Proof of Theorem 3.1.* We start from the result in Lemma A.9. First, it follows from the definition of $\mathbf{z}_{t+1}$ that

$$\|\mathbf{z}_t - \mathbf{z}_{t+1}\| = \beta\|\mathbf{x}_t - \mathbf{z}_t\|.$$

So, we rewrite (37), as:

$$\mathbb{E}V_t - \mathbb{E}V_{t+1} \geq \beta^2 c_\beta \mathbb{E}\|\mathbf{x}_t - \mathbf{z}_t\|^2 + c_\tau \mathbb{E}\|\mathbf{u}^*(\mathbf{x}_t, \mathbf{y}_{t+1}, \mathbf{z}_t) - \mathbf{x}_t)\|^2 + c_\eta \mathbb{E}\|A\mathbf{x}^*(\mathbf{y}_{t+1}, \mathbf{z}_t) - \mathbf{b}\|^2 - \frac{1}{2}\lambda\tau^2\sigma^2. \tag{46}$$

For $t > 0$, we have $V_t \geq \underline{f}$, which is proven in Lemma A.13. It then follows that

$$\sum_{t=0}^{T-1}(\mathbb{E}V_t - \mathbb{E}V_{t+1}) = V_0 - \mathbb{E}V_T \leq V_0 - \underline{f}. \tag{47}$$

Then, summing up (46), using (47), and the fact that $c_\tau = \Theta(\tau)$, $c_\eta = \Theta(\tau)$, $\beta^2 c_\beta = \Theta(\tau)$ from (25), we have

$$V_0 - \underline{f} + \frac{1}{2}T\lambda\tau^2\sigma^2 \geq \sum_{t=1}^{T} C_0\tau \left[\mathbb{E}\|\mathbf{x}_t - \mathbf{z}_t\|^2 + \mathbb{E}\|\mathbf{u}^*(\mathbf{x}_t, \mathbf{y}_{t+1}, \mathbf{z}_t) - \mathbf{x}_t\|^2 + \mathbb{E}\|A\mathbf{x}^*(\mathbf{y}_{t+1}, \mathbf{z}_t) - \mathbf{b}\|^2\right],$$

for some explicit constant $C_0$.

Dividing both sides by $T$, rearranging and using the definition $\tau = \frac{1}{6\lambda^2\sqrt{T}}$ gives

$$\frac{1}{T}\sum_{t=0}^{T-1}\mathbb{E}\|\mathbf{x}_t - \mathbf{z}_t\|^2 + \mathbb{E}\|\mathbf{u}^*(\mathbf{x}_t, \mathbf{y}_{t+1}, \mathbf{z}_t) - \mathbf{x}_t\|^2 + \mathbb{E}\|A\mathbf{x}^*(\mathbf{y}_{t+1}, \mathbf{z}_t) - \mathbf{b}\|^2 \leq \frac{1}{C_0\sqrt{T}}\left(6\lambda(V_0 - \underline{f}) + \frac{\sigma^2}{12}\right). \tag{48}$$

Since we have

$$\nabla\Psi(\mathbf{z}_t) = \mu(\mathbf{z}_t - \bar{\mathbf{x}}^*(\mathbf{z}_t)),$$

by Danskin's theorem, we deduce for any $t$

$$
\begin{aligned}
\frac{1}{\mu^2}\|\nabla\Psi(\mathbf{z}_t)\| &= \|\mathbf{z}_t - \bar{\mathbf{x}}^*(\mathbf{z}_t)\| \\
&\leq \|\mathbf{z}_t - \mathbf{x}^*(\mathbf{y}_{t+1}, z_t)\| + \|\mathbf{x}^*(\mathbf{y}_{t+1}, \mathbf{z}_t) - \bar{\mathbf{x}}^*(\mathbf{z}_t)\| \\
&\leq \|\mathbf{z}_t - \mathbf{x}^*(\mathbf{y}_{t+1}, \mathbf{z}_t)\| + \bar{\sigma}\|A\mathbf{x}^*(\mathbf{y}_{t+1}, \mathbf{z}_s) - \mathbf{b}\| \\
&\leq \|\mathbf{z}_t - \mathbf{x}_t\| + \|\mathbf{x}_t - \mathbf{x}^*(\mathbf{y}_{t+1}, \mathbf{z}_t)\| + \bar{\sigma}\|A\mathbf{x}^*(\mathbf{y}_{t+1}, \mathbf{z}_t) - \mathbf{b}\| \\
&\leq \|\mathbf{z}_t - \mathbf{x}_t\| + \frac{\lambda}{\gamma}\|\mathbf{x}_t - \mathbf{u}^*(\mathbf{x}_t, \mathbf{y}_{t+1}, \mathbf{z}_t)\| + \bar{\sigma}\|A\mathbf{x}^*(\mathbf{y}_{t+1}, \mathbf{z}_t) - \mathbf{b}\|,
\end{aligned}
$$

where the first inequality is by triangle inequality, the second by (A.12), the third by triangle inequality and the fourth by (58).

Next, we take square of both sides, take expectation, use Young's inequality, sum for all $t = 0, 1, \ldots, T-1$, divide by $T$ and use (48) to derive

$$
\begin{aligned}
\frac{1}{\mu^2}\frac{1}{T}\sum_{t=0}^{T-1}\mathbb{E}\|\nabla\Psi(\mathbf{z}_t)\|^2 &\leq \frac{1}{T}\sum_{t=0}^{T-1}\mathbb{E}\left[3\|\mathbf{z}_t - \mathbf{x}_t\|^2 + \frac{3\lambda^2}{\gamma^2}\|\mathbf{x}_t - \mathbf{u}^*(\mathbf{x}_t, \mathbf{y}_{t+1}, \mathbf{z}_t)\|^2 + 3\bar{\sigma}^2\|A\mathbf{x}^*(\mathbf{y}_{t+1}, \mathbf{z}_t) - \mathbf{b}\|^2\right] \\
&= O\left(\frac{1}{\sqrt{T}}\right).
\end{aligned}
$$

The result then follows since $t^*$ is selected uniformly at random from $\{0, 1, 2, \ldots, T-1\}$. ∎

### A.3. Proof of Corollary 3.2

*Proof of Corollary 3.2.* From the definition of $\hat{\mathbf{x}}$, we have

$$
0 \in \hat{G}(\mathbf{x}_t, \mathbf{y}_{t+1}, \mathbf{z}_t) + \frac{2}{\tau}(\hat{\mathbf{x}} - \mathbf{x}_t) + \partial I_X(\hat{\mathbf{x}}).
$$

Let us set

$$
\mathbf{v} = \nabla_{\mathbf{x}}K(\hat{\mathbf{x}}, \mathbf{y}_{t+1}, \mathbf{z}_t) - \hat{G}(\mathbf{x}_t, \mathbf{y}_{t+1}, \mathbf{z}_t) - \frac{2}{\tau}(\hat{\mathbf{x}} - \mathbf{x}_t) - \rho A^T(A\hat{\mathbf{x}} - \mathbf{b}) - \mu(\hat{\mathbf{x}} - \mathbf{z}_t). \tag{49}
$$

Combining with the optimality condition, we have

$$
\begin{aligned}
\mathbf{v} &\in \nabla_{\mathbf{x}}K(\hat{\mathbf{x}}, \mathbf{y}_{t+1}, \mathbf{z}_t) - \rho A^T(A\hat{\mathbf{x}} - \mathbf{b}) - \mu(\hat{\mathbf{x}} - \mathbf{z}_t) + \partial I_X(\hat{\mathbf{x}}) \\
&= \nabla f(\hat{\mathbf{x}}) + A^T\mathbf{y}_{t+1} + \partial I_X(\hat{\mathbf{x}}).
\end{aligned}
$$

Hence, we need to estimate $\mathbb{E}\|A\hat{\mathbf{x}} - \mathbf{b}\|$ and $\mathbb{E}\|\mathbf{v}\|$.

For the mini-batch gradient in the post-processing step, we have

$$
\mathbb{E}\|\hat{G}(\mathbf{x}, \mathbf{y}, \mathbf{z}) - \nabla K(\mathbf{x}, \mathbf{y}, \mathbf{z})\|^2 \leq \frac{\sigma^2}{B}. \tag{50}
$$

which is a standard calculation, see for example, (Lan, 2020, Section 5.2.3). Since $B = \Theta(\varepsilon^{-2})$, this gives us

$$
\mathbb{E}\|\hat{G}(\mathbf{x}, \mathbf{y}, \mathbf{z}) - \nabla K(\mathbf{x}, \mathbf{y}, \mathbf{z})\|^2 \leq \varepsilon^2. \tag{51}
$$

First, let us note that the purpose of $\hat{\mathbf{x}}$ is to estimate $\mathbf{u}^*(\mathbf{x}_t, \mathbf{y}_{t+1}, \mathbf{z}_t)$, where

$$
\mathbf{u}^*(\mathbf{x}_t, \mathbf{y}_{t+1}, \mathbf{z}_t) = \arg\min_{\mathbf{u}\in X}\{l(\mathbf{u}) := K(\mathbf{u}, \mathbf{y}_{t+1}, \mathbf{z}_t) + \frac{\lambda}{2}\|\mathbf{x}_t - \mathbf{u}\|^2\}.
$$

Note that the gradient of this objective is

$$
\nabla l(\mathbf{u}) = \nabla_{\mathbf{x}}K(\mathbf{x}, \mathbf{y}_{t+1}, \mathbf{z}_t) + \lambda(\mathbf{x} - \mathbf{x}_t).
$$

As a result, we have $\nabla l(\mathbf{x}_t) = \nabla_{\mathbf{x}} K(\mathbf{x}_t, \mathbf{y}_{t+1}, \mathbf{z}_t)$.

Let us also denote

$$\mathbf{x}_t^* = \mathrm{proj}_X(\mathbf{x}_t - \tau \nabla l(\mathbf{x}_t)).$$

That is, $\mathbf{x}_t^*$ is the output of doing a full-gradient step on $\mathbf{x}_t$. Of course, this is not tractable in our setting, but we only use this as a theoretical tool.

Since this is a GD step on the objective $l$ which is $L_K$-smooth and convex with optimizer $\mathbf{u}^*(\mathbf{x}_t, \mathbf{y}_{t+1}, \mathbf{z}_t)$, the standard analysis for GD gives

$$\|\mathbf{x}_t^* - \mathbf{u}^*(\mathbf{x}_t, \mathbf{y}_{t+1}, \mathbf{z}_t)\|^2 \le \|\mathbf{x}_t - \mathbf{u}^*(\mathbf{x}_t, \mathbf{y}_{t+1}, \mathbf{z}_t)\|^2, \tag{52}$$

as long as $\tau \le \frac{1}{L_K}$.

Next, by the definitions of $\mathbf{x}_t^*$ and $\hat{\mathbf{x}}$, along with nonexpansiveness of the projection, we have

$$\begin{aligned}
\mathbb{E}\|\mathbf{x}_t^* - \hat{\mathbf{x}}\|^2 &\le \mathbb{E}\tau^2 \|\hat{G}(\mathbf{x}_t, \mathbf{y}_{t+1}, \mathbf{z}_{t+1}) - \nabla_{\mathbf{x}} K(\mathbf{x}_t, \mathbf{y}_{t+1}, \mathbf{z}_t)\|^2 \\
&\le \tau^2 \varepsilon^2,
\end{aligned} \tag{53}$$

where the second inequality used (51).

In view of (49), we estimate $\|\mathbf{v}\|$ as

$$\|\mathbf{v}\| \le \|\nabla_x K(\mathbf{x}_t, \mathbf{y}_{t+1}, \mathbf{z}_t) - \hat{G}(\mathbf{x}_t, \mathbf{y}_{t+1}, \mathbf{z}_t)\| + L_K \|\mathbf{x}_t - \hat{\mathbf{x}}\| + \frac{2}{\tau} \|\hat{\mathbf{x}} - \mathbf{x}_t\| + \rho \|A\| \|A\hat{\mathbf{x}} - \mathbf{b}\| + \mu \|\hat{\mathbf{x}} - \mathbf{z}_t\|.$$

On this, multiple applications of triangle inequality give

$$\begin{aligned}
\|\hat{\mathbf{x}} - \mathbf{x}_t\| &\le \|\hat{\mathbf{x}} - \mathbf{x}_t^*\| + \|\mathbf{x}_t^* - \mathbf{u}^*(\mathbf{x}_t, \mathbf{y}_{t+1}, \mathbf{z}_t)\| + \|\mathbf{u}^*(\mathbf{x}_t, \mathbf{y}_{t+1}, \mathbf{z}_t) - \mathbf{x}_t\| \\
&\le \|\hat{\mathbf{x}} - \mathbf{x}_t^*\| + 2\|\mathbf{u}^*(\mathbf{x}_t, \mathbf{y}_{t+1}, \mathbf{x}_t) - \mathbf{x}_t\|,
\end{aligned} \tag{54}$$

where the second line is due to (52).

For the feasibility, we have by triangle inequality that

$$\|\hat{\mathbf{x}} - \mathbf{z}_t\| \le \|\hat{\mathbf{x}} - \mathbf{x}_t\| + \|\mathbf{x}_t - \mathbf{z}_t\|. \tag{55}$$

As a result, we have that

$$\begin{aligned}
\|\mathbf{v}\| = O\big(&\|\hat{\mathbf{x}} - \mathbf{x}_t^*\| + \|\mathbf{x}_t - \mathbf{u}^*(\mathbf{x}_t, \mathbf{y}_{t+1}, \mathbf{z}_t)\| + \|A\hat{\mathbf{x}} - \mathbf{b}\| + \|\mathbf{x}_t - \mathbf{z}_t\| \\
&+ \|\nabla_{\mathbf{x}} K(\mathbf{x}_t, \mathbf{y}_{t+1}, \mathbf{z}_t) - \hat{G}(\mathbf{x}_t, \mathbf{y}_{t+1}, \mathbf{z}_t)\|\big).
\end{aligned} \tag{56}$$

For the feasibility, we have

$$\begin{aligned}
\|A\hat{\mathbf{x}} - \mathbf{b}\| &\le \|A\hat{\mathbf{x}} - A\mathbf{x}_t\| + \|A\mathbf{x}_t - \mathbf{b}\| \\
&\le \|A\| \|\hat{\mathbf{x}} - \mathbf{x}_t\| + \|A\mathbf{x}_t - \mathbf{b}\|.
\end{aligned}$$

Now, by invoking the above inequality for $t = t^*$, taking expectation, using Young's inequality, (54), (53) and (48) along with (66), we get that

$$\mathbb{E}\|A\hat{\mathbf{x}} - \mathbf{b}\|^2 \le \varepsilon^2, \tag{57}$$

since $T = \Omega(\varepsilon^{-4})$.

Finally, using $t = t^*$, taking square and then expectation of (56), using Young's inequality and then combining (57), (53), (51) and (48) gives the result since $T = \Omega(\varepsilon^{-4})$. $\blacksquare$

## A.4. Auxiliary Results

**Lemma A.10.** *Under Assumption 1.1, for any* $\mathbf{x}, \mathbf{z}, \mathbf{z}' \in X$, *we have*

$$\frac{\lambda}{\gamma}\|\mathbf{x} - \mathbf{u}^*(\mathbf{x}, \mathbf{y}, \mathbf{z})\| \geq \|\mathbf{x} - \mathbf{x}^*(\mathbf{y}, \mathbf{z})\|, \tag{58}$$

$$\|\mathbf{u}^*(\mathbf{x}, \mathbf{y}, \mathbf{z}) - \mathbf{x}\| \leq \|\mathbf{x} - \mathbf{x}^*(\mathbf{y}, \mathbf{z})\|, \tag{59}$$

$$\|\mathbf{u}^*(\mathbf{x}, \mathbf{y}, \mathbf{z}) - \mathbf{u}^*(\mathbf{x}, \mathbf{y}', \mathbf{z})\| \leq \frac{\|A\|}{\gamma_s}\|\mathbf{y} - \mathbf{y}'\|, \tag{60}$$

$$\|\mathbf{u}^*(\mathbf{x}, \mathbf{y}, \mathbf{z}) - \mathbf{u}^*(\mathbf{x}, \mathbf{y}, \mathbf{z}')\| \leq \frac{\mu}{\gamma_s}\|\mathbf{z} - \mathbf{z}'\|, \tag{61}$$

$$\|\mathbf{z}' - \mathbf{z}\| \geq \frac{\mu - L_f}{\mu}\|\mathbf{x}^*(\mathbf{y}, \mathbf{z}') - \mathbf{x}^*(\mathbf{y}, \mathbf{z})\|, \tag{62}$$

$$\|\mathbf{y}' - \mathbf{y}\| \geq \frac{\gamma_K}{\|A\|}\|\mathbf{x}^*(\mathbf{y}', \mathbf{z}) - \mathbf{x}^*(\mathbf{y}, \mathbf{z})\|, \tag{63}$$

$$\|\bar{\mathbf{x}}^*(\mathbf{z}) - \bar{\mathbf{x}}^*(\mathbf{z}')\| \leq \frac{\mu}{\mu - L_f}\|\mathbf{z} - \mathbf{z}'\|, \tag{64}$$

*where* $\gamma = \frac{(\mu - L_f)\lambda}{\mu - L_f + \lambda}, \gamma_s = \mu - L_f + \lambda, \gamma_K = \mu - L_f.$

*Proof.* The proofs for (62), (63), and (64) appear in (Zhang & Luo, 2022), so we omit these proofs.

We first prove (58). Let us note that $\mathbf{x}^*(\mathbf{y}, \mathbf{z})$ minimizes $\varphi_{1/\lambda}$, see for example (Hiriart-Urruty & Lemarechal, 1993, Theorem XV4.1.7). As a result, we have $\nabla_{\mathbf{x}}\varphi_{1/\lambda}(\mathbf{x}^*(\mathbf{y}, \mathbf{z}), \mathbf{y}, \mathbf{z}) = 0$. From Lemma A.3, we have that $\varphi_{1/\lambda}(\cdot, y, z)$ is $\gamma = \frac{(\mu - L_f)\lambda}{\mu - L_f + \lambda}$-strongly convex.

Then, by strong convexity, we have

$$\langle \nabla_{\mathbf{x}}\varphi_{1/\lambda}(\mathbf{x}^*(\mathbf{y}, \mathbf{z}), \mathbf{y}, \mathbf{z}) - \nabla_{\mathbf{x}}\varphi_{1/\lambda}(\mathbf{x}, \mathbf{y}, \mathbf{z}), \mathbf{x}^*(\mathbf{y}, \mathbf{z}) - \mathbf{x} \rangle \geq \gamma\|\mathbf{x} - \mathbf{x}^*(\mathbf{y}, \mathbf{z})\|^2$$
$$\iff \|\nabla_{\mathbf{x}}\varphi_{1/\lambda}(\mathbf{x}, \mathbf{y}, \mathbf{z})\| \geq \gamma\|\mathbf{x} - \mathbf{x}^*(\mathbf{y}, \mathbf{z})\|,$$

where the inclusion used $\nabla_{\mathbf{x}}\varphi_{1/\lambda}(\mathbf{x}^*(\mathbf{y}, \mathbf{z}), \mathbf{y}, \mathbf{z}) = 0$ established in the previous paragraph as well as Cauchy-Schwarz inequality. Then, using $\nabla_{\mathbf{x}}\varphi_{1/\lambda}(\mathbf{x}, \mathbf{y}, \mathbf{z}) = \lambda(\mathbf{x} - \mathbf{u}^*(\mathbf{x}, \mathbf{y}, \mathbf{z}))$, we obtain (58).

From definition of $\mathbf{u}^*(\mathbf{x}, \mathbf{y}, \mathbf{z})$, we have,

$$K(\mathbf{u}^*(\mathbf{x}, \mathbf{y}, \mathbf{z}), \mathbf{y}, \mathbf{z}) + \frac{\lambda}{2}\|\mathbf{x} - \mathbf{u}^*(\mathbf{x}, \mathbf{y}, \mathbf{z})\|^2 \leq K(\mathbf{x}^*(\mathbf{y}, \mathbf{z}), \mathbf{y}, \mathbf{z})\frac{\lambda}{2}\|\mathbf{x} - \mathbf{x}^*(\mathbf{y}, \mathbf{z})\|^2,$$

where we also remark that $\mathbf{x}^*(\mathbf{y}, \mathbf{z}) \in X$. Combining with $K(\mathbf{x} * (\mathbf{y}, \mathbf{z}), \mathbf{y}, \mathbf{z}) \leq K(\mathbf{u}^*(\mathbf{x}, \mathbf{y}, \mathbf{z}), \mathbf{y}, \mathbf{z})$, which follows from the definition of $\mathbf{x}^*(\mathbf{y}, \mathbf{z})$ we have (59).

The proofs of the other two assertions will use a similar idea to (Zhang & Luo, 2022), but there will be differences in the estimations since this previous work did not use the function $\varphi_{1/\lambda}$.

For (60), we proceed by using the definition of $\varphi_{1/\lambda}$ and adding and subtracting $K(\mathbf{u}^*(\mathbf{x}, \mathbf{y}', \mathbf{z}), \mathbf{y}, \mathbf{z})$ to get

$$K(\mathbf{u}^*(\mathbf{x}, \mathbf{y}, \mathbf{z}), \mathbf{y}, \mathbf{z}) + \frac{\lambda}{2}\|\mathbf{u}^*(\mathbf{x}, \mathbf{y}, \mathbf{z}) - \mathbf{x}\|^2 - K(\mathbf{u}^*(\mathbf{x}, \mathbf{y}', \mathbf{z}), \mathbf{y}', \mathbf{z}) - \frac{\lambda}{2}\|\mathbf{u}^*(\mathbf{x}, \mathbf{y}', \mathbf{z}) - \mathbf{x}\|^2$$

$$= K(\mathbf{u}^*(\mathbf{x}, \mathbf{y}, \mathbf{z}), \mathbf{y}, \mathbf{z}) + \frac{\lambda}{2}\|\mathbf{u}^*(\mathbf{x}, \mathbf{y}, \mathbf{z}) - \mathbf{x}\|^2$$

$$- K(\mathbf{u}^*(\mathbf{x}, \mathbf{y}', \mathbf{z}), \mathbf{y}, \mathbf{z}) - \frac{\lambda}{2}\|\mathbf{u}^*(\mathbf{x}, \mathbf{y}', \mathbf{z}) - \mathbf{x}\|^2$$

$$+ K(\mathbf{u}^*(\mathbf{x}, \mathbf{y}', \mathbf{z}), \mathbf{y}, \mathbf{z}) - K(\mathbf{u}^*(\mathbf{x}, \mathbf{y}', \mathbf{z}), \mathbf{y}', \mathbf{z})$$

$$\leq \frac{-\gamma_s}{2}\|\mathbf{u}^*(\mathbf{x}, \mathbf{y}, \mathbf{z}) - \mathbf{u}^*(\mathbf{x}, \mathbf{y}', \mathbf{z})\|^2 + \langle \mathbf{y} - \mathbf{y}', A\mathbf{u}^*(\mathbf{x}, \mathbf{y}', \mathbf{z}) - \mathbf{b} \rangle,$$

where last step uses $\mathbf{u} \mapsto K(\mathbf{u}, \mathbf{y}, \mathbf{z}) + \frac{\lambda}{2}\|\mathbf{u} - \mathbf{x}\|^2$ being $\gamma_s$-strongly convex (cf. Fact A.1) with minimizer $\mathbf{u}^*(\mathbf{x}, \mathbf{y}, \mathbf{z})$, as well as the definition of $K$.

We then argue similarly, this time adding and subtracting $K(\mathbf{u}^*(\mathbf{x}, \mathbf{y}, \mathbf{z}), \mathbf{y}', \mathbf{z})$:

$$K(\mathbf{u}^*(\mathbf{x}, \mathbf{y}, \mathbf{z}), \mathbf{y}, \mathbf{z}) + \frac{\lambda}{2}\|\mathbf{u}^*(\mathbf{x}, \mathbf{y}, \mathbf{z}) - \mathbf{x}\|^2 - K(\mathbf{u}^*(\mathbf{x}, \mathbf{y}', \mathbf{z}), \mathbf{y}', \mathbf{z}) - \frac{\lambda}{2}\|\mathbf{u}^*(\mathbf{x}, \mathbf{y}', \mathbf{z}) - \mathbf{x}\|^2$$

$$= K(\mathbf{u}^*(\mathbf{x}, \mathbf{y}, \mathbf{z}), \mathbf{y}', \mathbf{z}) + \frac{\lambda}{2}\|\mathbf{u}^*(\mathbf{x}, \mathbf{y}, \mathbf{z}) - \mathbf{x}\|^2$$

$$- K(\mathbf{u}^*(\mathbf{x}, \mathbf{y}', \mathbf{z}), \mathbf{y}', \mathbf{z}) - \frac{\lambda}{2}\|\mathbf{u}^*(\mathbf{x}, \mathbf{y}', \mathbf{z}) - \mathbf{x}\|^2$$

$$- K(\mathbf{u}^*(\mathbf{x}, \mathbf{y}, \mathbf{z}), \mathbf{y}', \mathbf{z}) + K(\mathbf{u}^*(\mathbf{x}, \mathbf{y}, \mathbf{z}), \mathbf{y}, \mathbf{z})$$

$$\geq \frac{\gamma_s}{2}\|\mathbf{u}^*(\mathbf{x}, \mathbf{y}, \mathbf{z}) - \mathbf{u}^*(\mathbf{x}, \mathbf{y}', \mathbf{z})\|^2 + \langle \mathbf{y} - \mathbf{y}', A\mathbf{u}^*(\mathbf{x}, \mathbf{y}, \mathbf{z}) - \mathbf{b} \rangle.$$

where last step uses that $\mathbf{u} \mapsto K(\mathbf{u}, \mathbf{y}', \mathbf{z}) + \frac{\lambda}{2}\|\mathbf{u} - \mathbf{x}\|^2$ is $\gamma_s$-strongly convex (cf. Fact A.1) with minimizer $\mathbf{u}^*(\mathbf{x}, \mathbf{y}', \mathbf{z})$ and the definition of $K$.

Combining the last two estimates give

$$\langle \mathbf{y} - \mathbf{y}', A\mathbf{u}^*(\mathbf{x}, \mathbf{y}', \mathbf{z}) - A\mathbf{u}^*(\mathbf{x}, \mathbf{y}, \mathbf{z}) \rangle \geq \gamma_s\|\mathbf{u}^*(\mathbf{x}, \mathbf{y}, \mathbf{z}) - \mathbf{u}^*(\mathbf{x}, \mathbf{y}', \mathbf{z})\|^2.$$

Using Cauchy-Schwarz inequality and the definition of operator norm gives (60).

The proof of (61) is similar to the proof of (60), just completed. In particular, by adding and subtracting $K(\mathbf{u}^*(\mathbf{x}, \mathbf{y}, \mathbf{z}), \mathbf{y}, \mathbf{z}')$, we have

$$K(\mathbf{u}^*(\mathbf{x}, \mathbf{y}, \mathbf{z}), \mathbf{y}, \mathbf{z}) + \frac{\lambda}{2}\|\mathbf{u}^*(\mathbf{x}, \mathbf{y}, \mathbf{z}) - \mathbf{x}\|^2 - K(\mathbf{u}^*(\mathbf{x}, \mathbf{y}, \mathbf{z}'), \mathbf{y}, \mathbf{z}') + \frac{\lambda}{2}\|\mathbf{u}^*(\mathbf{x}, \mathbf{y}, \mathbf{z}') - \mathbf{x}\|^2$$

$$= K(\mathbf{u}^*(\mathbf{x}, \mathbf{y}, \mathbf{z}), \mathbf{y}, \mathbf{z}) + \frac{\lambda}{2}\|\mathbf{u}^*(\mathbf{x}, \mathbf{y}, \mathbf{z}) - \mathbf{x}\|^2 - K(\mathbf{u}^*(\mathbf{x}, \mathbf{y}, \mathbf{z}'), \mathbf{y}, \mathbf{z}) - \frac{\lambda}{2}\|\mathbf{u}^*(\mathbf{x}, \mathbf{y}, \mathbf{z}') - \mathbf{x}\|^2$$

$$- K(\mathbf{u}^*(\mathbf{x}, \mathbf{y}, \mathbf{z}'), \mathbf{y}, \mathbf{z}') + K(\mathbf{u}^*(\mathbf{x}, \mathbf{y}, \mathbf{z}'), \mathbf{y}, \mathbf{z})$$

$$\leq -\frac{\gamma_s}{2}\|\mathbf{u}^*(\mathbf{x}, \mathbf{y}, \mathbf{z}) - \mathbf{u}^*(\mathbf{x}, \mathbf{y}, \mathbf{z}')\|^2 + \frac{\mu}{2}(\|\mathbf{u}^*(\mathbf{x}, \mathbf{y}, \mathbf{z}') - \mathbf{z}\|^2 - \|\mathbf{u}^*(\mathbf{x}, \mathbf{y}, \mathbf{z}') - \mathbf{z}'\|^2),$$

where we used that $\mathbf{u} \mapsto K(\mathbf{u}, \mathbf{y}, \mathbf{z}) + \frac{\lambda}{2}\|\mathbf{u} - \mathbf{x}\|^2$ is $\gamma_s$-strongly convex with minimizer $\mathbf{u}^*(\mathbf{x}, \mathbf{y}, \mathbf{z})$ and the definition of $K$.

Finally, we add and subtract $K(\mathbf{u}^*(\mathbf{x}, \mathbf{y}, \mathbf{z}'), \mathbf{y}, \mathbf{z})$ to get

$$K(\mathbf{u}^*(\mathbf{x}, \mathbf{y}, \mathbf{z}), \mathbf{y}, \mathbf{z}) + \frac{\lambda}{2}\|\mathbf{u}^*(\mathbf{x}, \mathbf{y}, \mathbf{z}) - \mathbf{x}\|^2 - K(\mathbf{u}^*(\mathbf{x}, \mathbf{y}, \mathbf{z}'), \mathbf{y}, \mathbf{z}') - \frac{\lambda}{2}\|\mathbf{u}^*(\mathbf{x}, \mathbf{y}, \mathbf{z}') - \mathbf{x}\|^2$$

$$= K(\mathbf{u}^*(\mathbf{x}, \mathbf{y}, \mathbf{z}), \mathbf{y}, \mathbf{z}') + \frac{\lambda}{2}\|\mathbf{u}^*(\mathbf{x}, \mathbf{y}, \mathbf{z}) - \mathbf{x}\|^2 - K(\mathbf{u}^*(\mathbf{x}, \mathbf{y}, \mathbf{z}'), \mathbf{y}, \mathbf{z}) - \frac{\lambda}{2}\|\mathbf{u}^*(\mathbf{x}, \mathbf{y}, \mathbf{z}') - \mathbf{x}\|^2$$

$$+ K(\mathbf{u}^*(\mathbf{x}, \mathbf{y}, \mathbf{z}), \mathbf{y}, \mathbf{z}) - K(\mathbf{u}^*(\mathbf{x}, \mathbf{y}, \mathbf{z}), \mathbf{y}, \mathbf{z}')$$

$$\geq \frac{\gamma_s}{2}\|\mathbf{u}^*(\mathbf{x}, \mathbf{y}, \mathbf{z}) - \mathbf{u}^*(\mathbf{x}, \mathbf{y}, \mathbf{z}')\|^2 + \frac{\mu}{2}(\|\mathbf{u}^*(\mathbf{x}, \mathbf{y}, \mathbf{z}) - \mathbf{z}\|^2 - \|\mathbf{u}^*(\mathbf{x}, \mathbf{y}, \mathbf{z}) - \mathbf{z}'\|^2),$$

where we used that $\mathbf{u} \mapsto K(\mathbf{u}, \mathbf{y}, \mathbf{z}') + \frac{\lambda}{2}\|\mathbf{u} - \mathbf{x}\|^2$ is $\gamma_s$-strongly convex with minimizer $\mathbf{u}^*(\mathbf{x}, \mathbf{y}, \mathbf{z}')$ and the definition of $K$.

Combining the last two inequalities give

$$\mu\langle \mathbf{u}^*(\mathbf{x}, \mathbf{y}, \mathbf{z}') - \mathbf{u}^*(\mathbf{x}, \mathbf{y}, \mathbf{z}), \mathbf{z}' - \mathbf{z} \rangle \geq \gamma_s\|\mathbf{u}^*(\mathbf{x}, \mathbf{y}, \mathbf{z}) - \mathbf{u}^*(\mathbf{x}, \mathbf{y}, \mathbf{z}')\|^2.$$

Using Cauchy-Schwarz inequality concludes the proof. ∎

**Lemma A.11.** *Under Assumption 1.1, for $\mathbf{x}_t, \mathbf{y}_{t+1}, \mathbf{z}_t$ generated by Algorithm 1, we have*

$$\|A\mathbf{x}_t - A\mathbf{x}^*(\mathbf{y}_{t+1}, \mathbf{z}_t)\|^2 \leq \frac{\|A\|^2\lambda^2}{\gamma^2}\|\mathbf{x}_t - \mathbf{u}^*(\mathbf{x}_t, \mathbf{y}_{t+1}, \mathbf{z}_t)\|^2, \tag{65}$$

$$\|A\mathbf{x}_t - \mathbf{b}\|^2 \leq \frac{2\|A\|^2\lambda^2}{\gamma^2}\|\mathbf{x}_t - \mathbf{u}^*(\mathbf{x}_t, \mathbf{y}_{t+1}, \mathbf{z}_t)\|^2 + 2\|A\mathbf{x}^*(\mathbf{y}_{t+1}, \mathbf{z}_t) - \mathbf{b}\|^2, \tag{66}$$

$$\|A\mathbf{u}^*(\mathbf{x}_t, \mathbf{y}_t, \mathbf{z}_t) - A\mathbf{x}_t\|^2 \leq \frac{2\|A\|^4}{\gamma_s^2}\|\mathbf{y}_t - \mathbf{y}_{t+1}\|^2 + 2\|A\|^2\|\mathbf{u}^*(\mathbf{x}_t, \mathbf{y}_{t+1}, \mathbf{z}_t) - \mathbf{x}_t\|^2, \tag{67}$$

*where $\gamma, \gamma_s$ are defined in (25).*

*Proof.* The assertion in (65) follows directly from (58) since

$$\|A\mathbf{x}_t - A\mathbf{x}^*(\mathbf{y}_{t+1}, \mathbf{z}_t)\|^2 \leq \|A\|^2\|\mathbf{x}_t - \mathbf{x}^*(\mathbf{y}_{t+1}, \mathbf{z}_t)\|^2 \leq \frac{\|A\|^2\lambda^2}{\gamma^2}\|\mathbf{x}_t - \mathbf{u}^*(\mathbf{x}_t, \mathbf{y}_{t+1}, \mathbf{z}_t)\|^2.$$

Combining the first assertion with Young's inequality gives the second assertion, since

$$
\begin{aligned}
\|A\mathbf{x}_t - \mathbf{b}\|^2 &\leq 2\|A\mathbf{x}_t - A\mathbf{x}^*(\mathbf{y}_{t+1}, \mathbf{z}_t)\|^2 + 2\|A\mathbf{x}^*(\mathbf{y}_{t+1}, \mathbf{z}_t) - \mathbf{b}\|^2 \\
&\leq \frac{2\|A\|^2\lambda^2}{\gamma^2}\|\mathbf{x}_t - \mathbf{u}^*(\mathbf{x}_t, \mathbf{y}_{t+1}, \mathbf{z}_t)\|^2 + 2\|A\mathbf{x}^*(\mathbf{y}_{t+1}, \mathbf{z}_t) - \mathbf{b}\|^2.
\end{aligned}
$$

Young's inequality and (60) gives the third assertion

$$
\begin{aligned}
\|A\mathbf{u}^*(\mathbf{x}_t, \mathbf{y}_t, \mathbf{z}_t) - A\mathbf{x}_t\|^2 &\leq 2\|A\mathbf{u}^*(\mathbf{x}_t, \mathbf{y}_t, \mathbf{z}_t) - A\mathbf{u}^*(\mathbf{x}_t, \mathbf{y}_{t+1}, \mathbf{z}_t)\|^2 + 2\|A\mathbf{u}^*(\mathbf{x}_t, \mathbf{y}_{t+1}, \mathbf{z}_t) - A\mathbf{x}_t\|^2 \\
&\leq \frac{2\|A\|^4}{\gamma_s^2}\|\mathbf{y}_t - \mathbf{y}_{t+1}\|^2 + 2\|A\|^2\|\mathbf{u}^*(\mathbf{x}_t, \mathbf{y}_{t+1}, \mathbf{z}_t) - \mathbf{x}_t\|^2.
\end{aligned}
$$

The proof is completed. ∎

The following important lemma is known as the global error bound in (Zhang & Luo, 2022). This global result holds in its entirety in our case, so we only state it here and refer to where it appeared originally for the precise definition of the constant $\bar{\sigma}$ which depends on Hoffman constant of certain linear systems.

**Lemma A.12.** *(Zhang & Luo, 2022, Lemma 3.2) If $\mu > L_f$, then we have*

$$\|\mathbf{x}^*(\mathbf{y}, \mathbf{z}) - \bar{\mathbf{x}}^*(\mathbf{z})\| \leq \bar{\sigma}\|A\mathbf{x}^*(\mathbf{y}, \mathbf{z}) - \mathbf{b}\| \quad \text{for any} \quad \mathbf{y}, \mathbf{z},$$

*where $\bar{\sigma} > 0$ depends only on the constants $C_1 = (L_f + \rho\|A\|^2 + \mu)$, $C_2 = -L_f + \mu$, and the matrices $A, H$ and is always finite.*

**Lemma A.13.** *If $(\mathbf{x}, \mathbf{z}) \in X \times X$, we have $\varphi_{1/\lambda}(\mathbf{x}, \mathbf{y}, \mathbf{z}) - 2d(\mathbf{y}, \mathbf{z}) + 2\Psi(\mathbf{z}) \geq \underline{f}$.*

*Proof.* Because $\mathbf{x}^*(\mathbf{y}, \mathbf{z})$ minimizes $\varphi_{1/\lambda}(\cdot, \mathbf{y}, \mathbf{z})$ (see for example (Hiriart-Urruty & Lemarechal, 1993, Theorem XV4.1.7)), we have

$$\varphi_{1/\lambda}(\mathbf{x}, \mathbf{y}, \mathbf{z}) \geq \varphi_{1/\lambda}(\mathbf{x}^*(\mathbf{y}, \mathbf{z}), \mathbf{y}, \mathbf{z}) = K(\mathbf{x}^*(\mathbf{y}, \mathbf{z}), \mathbf{y}, \mathbf{z}).$$

We can then deduce

$$
\begin{aligned}
\varphi_{1/\lambda}(\mathbf{x}, \mathbf{y}, \mathbf{z}) - 2d(\mathbf{y}, \mathbf{z}) + 2\Psi(\mathbf{z}) &\geq K(\mathbf{x}^*(\mathbf{y}, \mathbf{z}), \mathbf{y}, \mathbf{z}) - 2d(\mathbf{y}, \mathbf{z}) + 2\Psi(\mathbf{z}) \\
&= d(\mathbf{y}, \mathbf{z}) - 2d(\mathbf{y}, \mathbf{z}) + 2\Psi(\mathbf{z}) \\
&= \Psi(\mathbf{z}) + \Psi(\mathbf{z}) - d(\mathbf{y}, \mathbf{z}) \\
&\geq \Psi(\mathbf{z}) \\
&\geq \underline{f}
\end{aligned}
$$

The second inequality in the above chain comes from definition, that is, denoting $\mathbf{x}_\mu^* = \arg\min_{x \in X, A\mathbf{x}=\mathbf{b}}\{f(\mathbf{x}) + \frac{\mu}{2}\|\mathbf{x} - \mathbf{z}\|^2\}$ in view of (7), we have

$$d(\mathbf{y}, \mathbf{z}) = \min_{\mathbf{x} \in X} K(\mathbf{x}, \mathbf{y}, \mathbf{z}) \leq K(\mathbf{x}_\mu^*, \mathbf{y}, \mathbf{z}) = f(\mathbf{x}_\mu^*) + \frac{\mu}{2}\|\mathbf{x}_\mu^* - \mathbf{z}\|^2 = \Psi(\mathbf{z}),$$

where the first inequality also uses $\mathbf{x}_\mu^* \in X$, which is by definition. ∎

## B. Proofs for Section 4

**Notation.** In this section, we have $\|\nabla f(\mathbf{x}, \xi_t) - \nabla f(\mathbf{x}_t)\|^2 \leq \sigma^2$ and $\mathbb{E}\|A_{\zeta_t}\mathbf{x}_t - \mathbf{b}_{\zeta_t}\|^2 \leq L$, then we denote the boundedness of variance as $\mathbb{E}\|G(\mathbf{x}_t, \mathbf{y}_t, \mathbf{z}_t, \xi_t) - \nabla_x K(\mathbf{x}_t, \mathbf{y}_t, \mathbf{z}_t)\|^2 \leq \sigma_2^2$, where the boundedness is proved in B.2.

We start with some helper lemmas before proving Theorem 4.3.

**Lemma B.1.** *Let Assumption 4.1 hold. With the update rule of $\mathbf{y}_{t+1} = \mathbf{y}_t + \eta(A_{\zeta_t}\mathbf{x}_t - \mathbf{b}_{\zeta_t})$, where $\mathbb{E}_{\zeta_t}[A_{\zeta_t}\mathbf{x}_t - \mathbf{b}_{\zeta_t}] = A\mathbf{x}_t - \mathbf{b}$, we have*

$$\mathbb{E}d(\mathbf{y}_{t+1}, \mathbf{z}_{t+1}) - \mathbb{E}d(\mathbf{y}_t.\mathbf{z}_t) \geq \eta\mathbb{E}\langle(A\mathbf{x}_t - \mathbf{b}), A\mathbf{x}^*(\mathbf{y}_{t+1}, \mathbf{z}_t) - b\rangle - \frac{\eta^2}{32}\mathbb{E}\|A\mathbf{x}_t - \mathbf{b}\|^2 - \left(\frac{1}{2} + \frac{17\|A\|^4}{2\gamma_K}\right)\eta^2 L^2$$

$$+ \frac{\mu}{2}\mathbb{E}\langle\mathbf{z}_{t+1} - \mathbf{z}_t, \mathbf{z}_{t+1} + \mathbf{z}_t - 2\mathbf{x}^*(\mathbf{y}_{t+1}, \mathbf{z}_{t+1})\rangle, \tag{68}$$

$$\mathbb{E}\Psi(\mathbf{z}_{t+1}) - \mathbb{E}\Psi(\mathbf{z}_t) \leq \mu\mathbb{E}\langle\mathbf{z}_{t+1} - \mathbf{z}_t, \mathbf{z}_t - \bar{\mathbf{x}}^*(\mathbf{z}_t)\rangle + \frac{\mu}{2\sigma_4}\mathbb{E}\|\mathbf{z}_t - \mathbf{z}_{t+1}\|^2,$$

*where $\gamma_K, \sigma_4$ are introduceed in A.10, and by Assumption 4.1, we have $\mathbb{E}\|A_{\zeta_t}\mathbf{x}_t - \mathbf{b}_{\zeta_t}\|^2 \leq L$ for some finite $L$.*

*Proof.* It is easy to derive, for example as (Zhang & Luo, 2020, Lemma 3.2), that

$$d(\mathbf{y}_{t+1}, \mathbf{z}_{t+1}) - d(\mathbf{y}_t, \mathbf{z}_t) \geq \langle\mathbf{y}_{t+1} - \mathbf{y}_t, A\mathbf{x}^*(\mathbf{y}_{t+1}, \mathbf{z}_t) - \mathbf{b}\rangle + \frac{\mu}{2}\langle\mathbf{z}_{t+1} - \mathbf{z}_t, \mathbf{z}_{t+1} + \mathbf{z}_t - 2\mathbf{x}^*(\mathbf{y}_{t+1}, \mathbf{z}_{t+1})\rangle.$$

Hence, by using the update rule of $\mathbf{y}_{t+1}$, we get

$$d(\mathbf{y}_{t+1}, \mathbf{z}_{t+1}) - d(\mathbf{y}_t, \mathbf{z}_t) \geq \langle\mathbf{y}_{t+1} - \mathbf{y}_t, A\mathbf{x}^*(\mathbf{y}_t, \mathbf{z}_t) - \mathbf{b}\rangle + \langle\mathbf{y}_{t+1} - \mathbf{y}_t, A\mathbf{x}^*(\mathbf{y}_{t+1}, \mathbf{z}_t) - A\mathbf{x}^*(\mathbf{y}_t, \mathbf{z}_t)\rangle$$

$$+ \frac{\mu}{2}\langle\mathbf{z}_{t+1} - \mathbf{z}_t, \mathbf{z}_{t+1} + \mathbf{z}_t - 2\mathbf{x}^*(\mathbf{y}_{t+1}, \mathbf{z}_{t+1})\rangle$$

$$\geq \langle\mathbf{y}_{t+1} - \mathbf{y}_t, A\mathbf{x}^*(\mathbf{y}_t, \mathbf{z}_t) - \mathbf{b}\rangle - \frac{1}{2}\|\mathbf{y}_{t+1} - \mathbf{y}_t\|^2 - \frac{1}{2}\|A\mathbf{x}^*(\mathbf{y}_{t+1}, \mathbf{z}_t) - A\mathbf{x}^*(\mathbf{y}_t, \mathbf{z}_t)\|^2$$

$$+ \frac{\mu}{2}\langle\mathbf{z}_{t+1} - \mathbf{z}_t, \mathbf{z}_{t+1} + \mathbf{z}_t - 2\mathbf{x}^*(\mathbf{y}_{t+1}, \mathbf{z}_{t+1})\rangle$$

$$\geq \langle\eta(A_{\zeta_t}\mathbf{x}_t - \mathbf{b}_{\zeta_t}), A\mathbf{x}^*(\mathbf{y}_t, \mathbf{z}_t) - \mathbf{b}\rangle - \frac{1}{2}\eta^2 L^2 - \frac{\|A\|^4}{2\gamma_K^2}\|\mathbf{y}_{t+1} - \mathbf{y}_t\|^2$$

$$+ \frac{\mu}{2}\langle\mathbf{z}_{t+1} - \mathbf{z}_t, \mathbf{z}_{t+1} + \mathbf{z}_t - 2\mathbf{x}^*(\mathbf{y}_{t+1}, \mathbf{z}_{t+1})\rangle,$$

where we use Cauchy-Schwarz inequality in the second step, and the last inequality comes from the bound of $\mathbb{E}\|A_{\zeta_t}\mathbf{x}_t - \mathbf{b}_{\zeta_t}\|^2$ also (63).

After taking expectation and using tower property along with $\mathbf{y}_t, \mathbf{z}_t$ being deterministic under the conditioning, we have

$$\mathbb{E}d(\mathbf{y}_{t+1}, \mathbf{z}_{t+1}) - \mathbb{E}d(\mathbf{y}_t, \mathbf{z}_t) \geq \mathbb{E}\langle\eta(A\mathbf{x}_t - \mathbf{b}), A\mathbf{x}^*(\mathbf{y}_t, \mathbf{z}_t) - \mathbf{b}\rangle - \frac{1}{2}\eta^2 L^2 - \frac{\|A\|^4}{2\gamma_K^2}\mathbb{E}\|\mathbf{y}_{t+1} - \mathbf{y}_t\|^2$$

$$+ \frac{\mu}{2}\mathbb{E}\langle\mathbf{z}_{t+1} - \mathbf{z}_t, \mathbf{z}_{t+1} + \mathbf{z}_t - 2\mathbf{x}(\mathbf{y}_{t+1}, \mathbf{z}_{t+1})\rangle.$$

Then we estimate as

$$\mathbb{E}\langle\eta(A\mathbf{x}_t - \mathbf{b}), A\mathbf{x}^*(\mathbf{y}_t, \mathbf{z}_t) - \mathbf{b}\rangle - \frac{1}{2}\eta^2 L^2 - \frac{\|A\|^4}{2\gamma_K^2}\mathbb{E}\|\mathbf{y}_{t+1} - \mathbf{y}_t\|^2$$

$$\geq \eta\mathbb{E}\langle(A\mathbf{x}_t - \mathbf{b}), A\mathbf{x}^*(\mathbf{y}_t, \mathbf{z}_t) - \mathbf{b}\rangle - \left(\frac{1}{2} + \frac{\|A\|^4}{2\gamma_K^2}\right)\eta^2 L^2$$

$$= \eta\mathbb{E}[\langle(A\mathbf{x}_t - \mathbf{b}), A\mathbf{x}^*(\mathbf{y}_{t+1}, \mathbf{z}_t) - \mathbf{b}\rangle + \eta\langle(A\mathbf{x}_t - \mathbf{b}), -A\mathbf{x}^*(\mathbf{y}_{t+1}, \mathbf{z}_t) + A\mathbf{x}^*(\mathbf{y}_t, \mathbf{z}_t)\rangle] - \left(\frac{1}{2} + \frac{\|A\|^4}{2\gamma_K^2}\right)\eta^2 L^2$$

$$\geq \eta\mathbb{E}[\langle(A\mathbf{x}_t - \mathbf{b}), A\mathbf{x}^*(\mathbf{y}_{t+1}, \mathbf{z}_t) - \mathbf{b}\rangle - 8\|A\mathbf{x}^*(\mathbf{y}_{t+1}, \mathbf{z}_t) - A\mathbf{x}^*(\mathbf{y}_t, \mathbf{z}_t)\|^2] - \frac{\eta^2}{32}\|A\mathbf{x}_t - \mathbf{b}\|^2 - \left(\frac{1}{2} + \frac{\|A\|^4}{2\gamma_K^2}\right)\eta^2 L^2$$

$$\geq \eta\mathbb{E}\langle(A\mathbf{x}_t - \mathbf{b}), A\mathbf{x}^*(\mathbf{y}_{t+1}, \mathbf{z}_t) - \mathbf{b}\rangle - \frac{\eta^2}{32}\mathbb{E}\|A\mathbf{x}_t - \mathbf{b}\|^2 - \frac{1}{2}\eta^2 L^2 - \frac{17\|A\|^4}{2\gamma_K^2}\eta^2 L^2,$$

where the first inequality comes from $\mathbb{E}[\|A_{\zeta_t}\mathbf{x}_t - \mathbf{b}_{\zeta_t}\|^2] \leq L$ and the second inequality comes from $\langle a, b\rangle \leq \frac{1}{32}\|a\|^2 + 8\|b\|^2 (\forall a, b)$. And in last inequality we use (63) again.

The estimation of $\mathbb{E}\Psi(\mathbf{z}_{t+1}) - \mathbb{E}\Psi(\mathbf{z}_t)$ is the same as Lemma A.8. Because the randomness of $\zeta_t$ in the stochastic dual update does not change the recursion in $\mathbb{E}\Psi(\mathbf{z}_{t+1}) - \mathbb{E}\Psi(\mathbf{z}_t)$, where $\mathbf{z}_t, \mathbf{z}_{t+1}$ only depend on the randomness before $\zeta_t$. Hence we omit the proof here.

This completes the proof. ∎

**Lemma B.2.** *Let Assumption 1.1 and 4.1 hold. By using the parameters (25) in Algorithm 2, then in the iteration $t + 1$, if the dual update runs as $\mathbf{y}_{t+1} = \mathbf{y}_t + \eta(A_{\zeta_t}\mathbf{x}_t - \mathbf{b}_{\zeta_t})$, we obtain*

$$\mathbb{E}V_t - \mathbb{E}V_{t+1} \geq \widetilde{c}_\beta\mathbb{E}\|\mathbf{z}_{t+1} - \mathbf{z}_t\|^2 + \widetilde{c}_\tau\mathbb{E}\|\mathbf{u}^*(\mathbf{x}_t, \mathbf{y}_{t+1}, \mathbf{z}_t) - \mathbf{x}_t\|^2 + \widetilde{c}_\eta\mathbb{E}\|A\mathbf{x}^*(\mathbf{y}_{t+1}, \mathbf{z}_t) - \mathbf{b}\|^2$$
$$- \lambda\tau^2\sigma_2^2 - \left(1 + \frac{17\|A\|^4}{\gamma_K^2}\right)\eta^2 L^2, \tag{69}$$

*where $\widetilde{c}_\beta = \frac{\mu}{50\beta}$, $\widetilde{c}_\tau = \frac{6\tau\lambda^2}{400}$, $\widetilde{c}_\eta = \frac{\eta}{8}$ and $\mathbb{E}\|G(\mathbf{x}_t, \mathbf{y}_t, \mathbf{z}_t, \xi_t) - \nabla_x K(\mathbf{x}_t, \mathbf{y}_t, \mathbf{z}_t)\|^2 \leq \sigma_2^2$.*

*Proof.* First, we show $\mathbb{E}\|G(\mathbf{x}_t, \mathbf{y}_t, \mathbf{z}_t, \xi_t) - \nabla_x K(\mathbf{x}_t, \mathbf{y}_t, \mathbf{z}_t)\|^2$ is bounded.

Recall that in Equation (17) we have

$$G(\mathbf{x}, \mathbf{y}, \mathbf{z}, \xi) = \nabla f(\mathbf{x}, \xi) + A_{\zeta^1}^\top\mathbf{y} + \rho A_{\zeta^1}^\top(A_{\zeta^2}\mathbf{x} - \mathbf{b}_{\zeta^2}) + \mu(\mathbf{x} - \mathbf{z}). \tag{70}$$

We estimate by using Young's inequalities

$$\mathbb{E}\|G(\mathbf{x}_t, \mathbf{y}_t, \mathbf{z}_t, \xi_t) - \nabla_\mathbf{x}K(\mathbf{x}_t, \mathbf{y}_t, \mathbf{z}_t)\|^2$$
$$\leq 2\mathbb{E}\|G(\mathbf{x}_t, \mathbf{y}_t, \mathbf{z}_t, \xi_t) - G(\mathbf{x}_t, 0, \mathbf{z}_t, \xi_t)\|^2 + 2\mathbb{E}\|G(\mathbf{x}_t, 0, \mathbf{z}_t, \xi_t) - \nabla_\mathbf{x}K(\mathbf{x}_t, \mathbf{y}_t, \mathbf{z}_t)\|^2$$
$$\leq 2\mathbb{E}L_G\|\mathbf{y}_t\|^2 + 2\mathbb{E}\|G(\mathbf{x}_t, 0, \mathbf{z}_t, \xi_t) - \nabla_\mathbf{x}K(\mathbf{x}_t, \mathbf{y}_t, \mathbf{z}_t)\|^2$$
$$\leq 2\mathbb{E}L_G\|\mathbf{y}_t\|^2 + 4\mathbb{E}\|G(\mathbf{x}_t, 0, \mathbf{z}_t, \xi_t) - \nabla_\mathbf{x}K(\mathbf{x}_t, 0, \mathbf{z}_t)\|^2 + 4\mathbb{E}\|\nabla_\mathbf{x}K(\mathbf{x}_t, 0, \mathbf{z}_t) - \nabla_\mathbf{x}K(\mathbf{x}_t, \mathbf{y}_t, \mathbf{z}_t)\|^2$$
$$\leq 2L_G M_y^2 + 4\mathbb{E}\|G(\mathbf{x}_t, 0, \mathbf{z}_t, \xi_t) - \nabla_\mathbf{x}K(\mathbf{x}_t, 0, \mathbf{z}_t)\|^2 + 4\|A\|^2\|\mathbf{y}_t\|^2,$$

where in second inequality we use $L_G$ is the Lipschitz constant of $G$ with respect to variable $\mathbf{y}$, then in third inequality we use $M_y$ as the upper bound of $\|\mathbf{y}_t\|$.

Because $\mathbf{x}_t, \mathbf{y}_t, \mathbf{z}_t$ are all bounded, $\mathbb{E}\|G(\mathbf{x}_t, \mathbf{y}_t, \mathbf{z}_t, \xi_t) - \nabla_\mathbf{x}K(\mathbf{x}_t, \mathbf{y}_t, \mathbf{z}_t)\|^2$ is bounded, we denote the upper bound as $\sigma_2^2$.

Note that Corollary A.7 still holds for $\mathbf{x}_t, \mathbf{y}_t, \mathbf{z}_t, \mathbf{x}_{t+1}, \mathbf{y}_{t+1}, \mathbf{z}_{t+1}$, but the variance $\sigma$ is changed to $\sigma_2$ (since this corollary

and the lemmas used in its proof do not use the particular form of $\mathbf{y}_{t+1}$). Then combining with Lemma B.1, we have

$$
\begin{aligned}
\mathbb{E}[V_t - V_{t+1}] = {}& \mathbb{E}\left[\varphi_{1/\lambda}(\mathbf{x}_t, \mathbf{y}_t, \mathbf{z}_t) - \varphi_{1/\lambda}(\mathbf{x}_{t+1}, \mathbf{y}_{t+1}, \mathbf{z}_{t+1}) + 2d(\mathbf{y}_{t+1}, \mathbf{z}_{t+1}) - 2d(\mathbf{y}_t, \mathbf{z}_t) + 2\Psi(\mathbf{z}_t) - 2\Psi(\mathbf{z}_{t+1})\right] \\
\geq {}& \frac{\tau\lambda^2}{16}\mathbb{E}\|\mathbf{u}^*(\mathbf{x}_t, \mathbf{y}_{t+1}, \mathbf{z}_t) - \mathbf{x}_t\|^2 - \left(\frac{\lambda\tau\mu}{2} + \lambda\tau^2\mu^2 + \frac{\tau\lambda^2\mu^2}{8\gamma_s^2}\right)\mathbb{E}\|\mathbf{z}_t - \mathbf{z}_{t+1}\|^2 - \frac{\lambda\tau^2\sigma_2^2}{2} \\
& - \eta\mathbb{E}\langle A\mathbf{x}_t - \mathbf{b}, A\mathbf{u}^*(\mathbf{x}_t, \mathbf{y}_{t+1}, \mathbf{z}_t) - \mathbf{b}\rangle + \frac{\mu}{2}\mathbb{E}\langle \mathbf{z}_{t+1} - \mathbf{z}_t, 2\mathbf{u}^*(\mathbf{x}_t, \mathbf{y}_{t+1}, \mathbf{z}_t) - \mathbf{z}_t - \mathbf{z}_{t+1}\rangle \\
& + 2\eta\mathbb{E}\langle A\mathbf{x}_t - \mathbf{b}, A\mathbf{x}^*(\mathbf{y}_{t+1}, \mathbf{z}_t) - \mathbf{b}\rangle + \mu\mathbb{E}\langle \mathbf{z}_{t+1} - \mathbf{z}_t, \mathbf{z}_{t+1} + \mathbf{z}_t - 2\mathbf{x}^*(\mathbf{y}_{t+1}, \mathbf{z}_{t+1})\rangle \\
& - 2\mu\mathbb{E}\langle \mathbf{z}_{t+1} - \mathbf{z}_t, \mathbf{z}_t - \bar{\mathbf{x}}^*(\mathbf{z}_t)\rangle - \frac{\mu}{\sigma_4}\mathbb{E}\|\mathbf{z}_t - \mathbf{z}_{t+1}\|^2 - \frac{\eta^2}{32}\mathbb{E}\|A\mathbf{x}_t - \mathbf{b}\|^2 - \left(\frac{1}{2} + \frac{17\|A\|^4}{2\gamma_K^2}\right)\eta^2 L^2,
\end{aligned}
$$

where $-\frac{\eta^2}{32}\mathbb{E}\|A\mathbf{x}_t - \mathbf{b}\|^2 - \left(\frac{1}{2} + \frac{17\|A\|^4}{2\gamma_K^2}\right)\eta^2 L^2$ is the difference comparing to the deterministic linear constraints result in Lemma A.9. We then estimate like Lemma A.9 to have

$$
\begin{aligned}
\mathbb{E}V_t - \mathbb{E}V_{t+1} \geq {}& c_\beta \mathbb{E}\|\mathbf{z}_{t+1} - \mathbf{z}_t\|^2 + c_\tau \mathbb{E}\|\mathbf{u}^*(\mathbf{x}_t, \mathbf{y}_{t+1}, \mathbf{z}_t) - \mathbf{x}_t\|^2 + c_\eta \mathbb{E}\|A\mathbf{x}^*(\mathbf{y}_{t+1}, \mathbf{z}_t) - \mathbf{b}\|^2 - \frac{1}{2}\lambda\tau^2\sigma_2^2 \\
& - \frac{\eta^2}{16}\mathbb{E}\|A\mathbf{x}_t - \mathbf{b}\|^2 - \left(1 + \frac{17\|A\|^4}{\gamma_K^2}\right)\eta^2 L^2,
\end{aligned} \tag{71}
$$

where $c_\beta = \frac{\mu}{50\beta}$, $c_\tau = \frac{7\tau\lambda^2}{400}$, $c_\eta = \frac{\eta}{4}$.

We also have by Young's inequality and Lemma A.11 that

$$
-\frac{\eta^2}{16}\mathbb{E}\|A\mathbf{x}_t - \mathbf{b}\|^2 \geq -\frac{\|A\|^2\lambda^2\eta^2}{8\gamma^2}\mathbb{E}\|\mathbf{x}_t - \mathbf{u}^*(\mathbf{x}_t, \mathbf{y}_{t+1}, \mathbf{z}_t)\|^2 - \frac{\eta^2}{8}\mathbb{E}\|A\mathbf{x}(\mathbf{y}_{t+1}, \mathbf{z}_t) - \mathbf{b}\|^2.
$$

By the parameter choices, we have $\frac{7\tau\lambda^2}{400} - \frac{\|A\|^2\lambda^2\eta^2}{8\gamma^2} \geq \frac{6\tau\lambda^2}{400}$ and $\frac{\eta}{4} - \frac{\eta^2}{8} \geq \frac{\eta}{8}$. Using these estimations in (71) gives the proof. ∎

**Proposition B.3.** *Under Assumption 4.1, $\|\mathbf{y}_t\| \leq \frac{\Psi(\mathbf{z}_t) - d(\mathbf{y}_t, \mathbf{z}_t) + 2M}{r}$, where $M = \max_{\mathbf{x}, \mathbf{z} \in X}\{|f(\mathbf{x})| + \frac{\mu}{2}\|\mathbf{x} - \mathbf{z}\|^2 + \frac{\rho}{2}\|A\mathbf{x} - \mathbf{b}\|^2\}$ and $r > 0$ is defined as $\|A\hat{\mathbf{x}} - \mathbf{b}\| = r$ where $\hat{\mathbf{x}}$ is in the relative interior of the constraints. The existence of this is guaranteed by our assumption.*

*Proof.* Given $\widetilde{\mathbf{x}} \in X$, we have

$$
\begin{aligned}
\Psi(\mathbf{z}_t) - d(\mathbf{y}_t, \mathbf{z}_t) \geq {}& f(\bar{\mathbf{x}}^*(\mathbf{z}_t)) + \frac{\mu}{2}\|\bar{\mathbf{x}}^*(\mathbf{z}_t) - \mathbf{z}_t\|^2 - K(\widetilde{\mathbf{x}}, \mathbf{y}_t, \mathbf{z}_t) \\
\geq {}& f(\bar{\mathbf{x}}^*(\mathbf{z}_t)) + \frac{\mu}{2}\|\bar{\mathbf{x}}^*(\mathbf{z}_t) - \mathbf{z}_t\|^2 - \left[f(\widetilde{\mathbf{x}}) + \langle \mathbf{y}_t, A\widetilde{\mathbf{x}}\rangle + \frac{\rho}{2}\|A\widetilde{\mathbf{x}} - \mathbf{b}\|^2 + \frac{\mu}{2}\|\widetilde{\mathbf{x}} - \mathbf{z}_t\|^2\right] \\
= {}& \left[f(\bar{\mathbf{x}}^*(\mathbf{z}_t)) + \frac{\mu}{2}\|\bar{\mathbf{x}}^*(\mathbf{z}_t) - \mathbf{z}_t\|^2 - f(\widetilde{\mathbf{x}}) - \frac{\mu}{2}\|\widetilde{\mathbf{x}} - \mathbf{z}_t\|^2\right] - \langle \mathbf{y}_t, A\widetilde{\mathbf{x}} - \mathbf{b}\rangle - \frac{\rho}{2}\|A\widetilde{\mathbf{x}} - \mathbf{b}\|^2 \\
= {}& \left[f(\bar{\mathbf{x}}^*(\mathbf{z}_t)) + \frac{\mu}{2}\|\bar{\mathbf{x}}^*(\mathbf{z}_t) - \mathbf{z}_t\|^2 - f(\widetilde{\mathbf{x}}) - \frac{\mu}{2}\|\widetilde{\mathbf{x}} - \mathbf{z}_t\|^2 - \frac{\rho}{2}\|A\widetilde{\mathbf{x}} - \mathbf{b}\|^2\right] - \langle \mathbf{y}_t, A\widetilde{\mathbf{x}} - \mathbf{b}\rangle \\
\geq {}& -2M - \langle \mathbf{y}_t, A\widetilde{\mathbf{x}} - \mathbf{b}\rangle,
\end{aligned}
$$

where the first inequality comes from the definition of $\Psi(\mathbf{z}_t)$ and

$$
d(\mathbf{y}_t, \mathbf{z}_t) = \min_{\mathbf{x} \in X} K(\mathbf{x}, \mathbf{y}, \mathbf{z}).
$$

Finally, in the last inequality, we let

$$
M = \max_{(\mathbf{x}, \mathbf{z}) \in X \times X}\{|f(\mathbf{x})| + \frac{\mu}{2}\|\mathbf{x} - \mathbf{z}\|^2 + \frac{\rho}{2}\|A\mathbf{x} - \mathbf{b}\|^2\}.
$$

As a result, we have the last inequality.

According to Assumption 4.1(2), there exists a positive $r > 0$ such that for any direction $\mathbf{d} \in \text{Range}(A)$, we can find a $\mathbf{x} \in X$ satisfying $\|A\mathbf{x} - \mathbf{b}\| = r$ and $A\mathbf{x} - \mathbf{b}$ has the same direction as $\mathbf{d}$. Because $\mathbf{y}_t \in \text{Range}(A)$ (by assumption 4.1(3), $\text{Range}(A) = \mathbb{R}^m$) we can choose $\widetilde{\mathbf{x}}$ such that $A\widetilde{\mathbf{x}} - \mathbf{b}$ is of the same direction as $-\mathbf{y}_t$ and $\|A\widetilde{\mathbf{x}} - \mathbf{b}\| = r$. Then we obtain

$$\Psi(\mathbf{z}_t) - d(\mathbf{y}_t, \mathbf{z}_t) \geq -2M + r\|\mathbf{y}_t\| \implies \|\mathbf{y}_t\| \leq \frac{\Psi(\mathbf{z}_t) - d(\mathbf{y}_t, \mathbf{z}_t) + 2M}{r}, \forall t \in \{0, 1, ..., T\}.$$

This concludes the proof. ∎

## Proof of Theorem 4.3

*Proof of Theorem 4.3.* First, let $M_V = \max_{\mathbf{x},\mathbf{z} \in X}\{K(\mathbf{x}, 0, \mathbf{z}) - 2d(0, \mathbf{z}) + 2\Psi(\mathbf{z})\}$ and $M_y > \frac{M_V - M_\Psi + 2M}{r}$ where $M_\Psi$ is a uniform lower bound of $\Psi(\mathbf{z}_t)$, for example, $\underline{f}$.

Here, We denote the $\mathbf{x}, \mathbf{y}, \mathbf{z}$ generated by Algorithm 2 at iteration $t$ as $\mathbf{x}_t, \mathbf{y}_t, \mathbf{z}_t$ and the output of iteration $t + 1$ as $\mathbf{x}_{t+1}, \mathbf{y}_{t+1}, \mathbf{z}_{t+1}$.

If $\|\mathbf{y}_t + \eta(A_{\zeta_t}\mathbf{x}_t - \mathbf{b}_{\zeta_t})\| \leq M_y$, then

$$\begin{aligned}
&\mathbb{E}\left[V(\mathbf{x}_t, \mathbf{y}_t, \mathbf{z}_t) - V(\mathbf{x}_{t+1}, \mathbf{y}_{t+1}, \mathbf{z}_{t+1})\right] \\
&\geq \widetilde{c}_\beta\mathbb{E}\|\mathbf{z}_{t+1} - \mathbf{z}_t\|^2 + \widetilde{c}_\tau\mathbb{E}\|\mathbf{u}^*(\mathbf{x}_t, \mathbf{y}_{t+1}, \mathbf{z}_t) - \mathbf{x}_t\|^2 + \widetilde{c}_\eta\mathbb{E}\|A\mathbf{x}^*(\mathbf{y}_{t+1}, \mathbf{z}_t) - \mathbf{b}\|^2 \\
&\quad - \frac{1}{2}\lambda\tau^2\sigma_2^2 - \left(1 + \frac{17\|A\|^4}{\gamma_K^2}\right)\eta^2 L^2 \\
&= \widetilde{c}_\beta\mathbb{E}\|\mathbf{z}_{t+1} - \mathbf{z}_t\|^2 + \widetilde{c}_\tau\mathbb{E}\|\mathbf{u}^*(\mathbf{x}_t, \mathbf{y}_t + \eta(A_{\zeta_t}\mathbf{x}_t - \mathbf{b}_{\zeta_t}), \mathbf{z}_t) - \mathbf{x}_t\|^2 \\
&\quad + \widetilde{c}_\eta\mathbb{E}\|A\mathbf{x}^*(\mathbf{y}_t + \eta(A_{\zeta_t}\mathbf{x}_t - \mathbf{b}_{\zeta_t}), \mathbf{z}_t) - \mathbf{b}\|^2 - \frac{1}{2}\lambda\tau^2\sigma_2^2 - \left(1 + \frac{17\|A\|^4}{\gamma_K^2}\right)\eta^2 L^2,
\end{aligned} \tag{72}$$

where the first inequality use Lemma B.2 and the equality comes from the update of $\mathbf{y}_{t+1}$ when $\|\mathbf{y}_t + \eta(A_{\zeta_t}\mathbf{x}_t - \mathbf{b}_{\zeta_t})\| \leq M_y$.

If $\|\mathbf{y}_t + \eta(A_{\zeta_t}\mathbf{x}_t - \mathbf{b}_{\zeta_t})\| > M_y$, we have $\mathbf{y}_{t+1} = 0$. Let us use $\hat{\mathbf{y}}_{t+1}, \hat{\mathbf{x}}_{t+1}, \hat{\mathbf{z}}_{t+1}$ denote the iteration generated with $\hat{\mathbf{y}}_{t+1} = \mathbf{y}_t + \eta(A_{\zeta_t}\mathbf{x}_t - \mathbf{b}_{\zeta_t})$. Then

$$\begin{aligned}
K(\hat{\mathbf{x}}_{t+1}, \hat{\mathbf{y}}_{t+1}, \hat{\mathbf{z}}_{t+1}) - 2d(\hat{\mathbf{y}}_{t+1}, \hat{\mathbf{z}}_{t+1}) + 2\Psi(\hat{\mathbf{z}}_{t+1}) &\geq \Psi(\hat{\mathbf{z}}_{t+1}) - d(\hat{\mathbf{y}}_{t+1}, \hat{\mathbf{z}}_{t+1}) + \Psi(\hat{\mathbf{z}}_{t+1}) \\
&\geq r\|\hat{\mathbf{y}}_{t+1}\| - 2M + M_\Psi \\
&\geq rM_y - 2M + M_\Psi \\
&\geq M_V \\
&= \max_{\mathbf{x},\mathbf{z} \in X}\{K(\mathbf{x}, 0, \mathbf{z}) - 2d(0, \mathbf{z}) + 2\Psi(\mathbf{z})\} \\
&\geq K(\mathbf{x}_{t+1}, 0, \mathbf{z}_{t+1}) - 2d(0, \mathbf{z}_{t+1}) + 2\Psi(\mathbf{z}_{t+1}) \\
&= K(\mathbf{x}_{t+1}, \mathbf{y}_{t+1}, \mathbf{z}_{t+1}) - 2d(\mathbf{y}_{t+1}, \mathbf{z}_{t+1}) + 2\Psi(\mathbf{z}_{t+1}),
\end{aligned}$$

where the first step used $d(\hat{\mathbf{y}}_{t+1}, \hat{\mathbf{z}}_{t+1}) \leq K(\hat{\mathbf{x}}_{t+1}, \hat{\mathbf{y}}_{t+1}, \hat{\mathbf{z}}_{t+1})$ and the second line uses Prop. B.3 and $\Psi(\hat{\mathbf{z}}_{t+1}) \geq M_\Psi$.

Hence we have

$$\begin{aligned}
&\mathbb{E}V(\mathbf{x}_t, \mathbf{y}_t, \mathbf{z}_t) - \mathbb{E}V(\mathbf{x}_{t+1}, \mathbf{y}_{t+1}, \mathbf{z}_{t+1}) \\
&= \mathbb{E}\left[K(\mathbf{x}_t, \mathbf{y}_t, \mathbf{z}_t) - 2d(\mathbf{y}_t, \mathbf{z}_t) + 2\Psi(\mathbf{z}_t)\right] - \mathbb{E}\left[K(\mathbf{x}_{t+1}, \mathbf{y}_{t+1}, \mathbf{z}_{t+1}) - 2d(\mathbf{y}_{t+1}, \mathbf{z}_{t+1}) + 2\Psi(\mathbf{z}_{t+1})\right] \\
&\geq \mathbb{E}\left[K(\mathbf{x}_t, \mathbf{y}_t, \mathbf{z}_t) - 2d(\mathbf{y}_t, \mathbf{z}_t) + 2\Psi(\mathbf{z}_t)\right] - \mathbb{E}\left[K(\hat{\mathbf{x}}_{t+1}, \hat{\mathbf{y}}_{t+1}, \hat{\mathbf{z}}_{t+1}) - 2d(\hat{\mathbf{y}}_{t+1}, \hat{\mathbf{z}}_{t+1}) + 2\Psi(\hat{\mathbf{z}}_{t+1})\right] \\
&\geq \widetilde{c}_\beta\mathbb{E}\|\hat{\mathbf{z}}_{t+1} - \mathbf{z}_t\|^2 + \widetilde{c}_\tau\mathbb{E}\|\mathbf{u}^*(\mathbf{x}_t, \hat{\mathbf{y}}_{t+1}, \mathbf{z}_t) - \mathbf{x}_t\|^2 + \widetilde{c}_\eta\mathbb{E}\|A\mathbf{x}^*(\hat{\mathbf{y}}_{t+1}, \mathbf{z}_t) - \mathbf{b}\|^2 \\
&\quad - \frac{1}{2}\lambda\tau^2\sigma_2^2 - \left(1 + \frac{17\|A\|^4}{\gamma_K^2}\right)\eta^2 L^2 \\
&= \widetilde{c}_\beta\mathbb{E}\|\mathbf{z}_{t+1} - \mathbf{z}_t\|^2 + \widetilde{c}_\tau\mathbb{E}\|\mathbf{u}^*(\mathbf{x}_t, \mathbf{y}_t + \eta(A_{\zeta_t}\mathbf{x}_t - \mathbf{b}_{\zeta_t}), \mathbf{z}_t) - \mathbf{x}_t\|^2 + \widetilde{c}_\eta\mathbb{E}\|A\mathbf{x}^*(\mathbf{y}_t + \eta(A_{\zeta_t}\mathbf{x}_t - \mathbf{b}_{\zeta_t}), \mathbf{z}_t) - \mathbf{b}\|^2 \\
&\quad - \frac{1}{2}\lambda\tau^2\sigma_2^2 - \left(1 + \frac{17\|A\|^4}{\gamma_K^2}\right)\eta^2 L^2,
\end{aligned} \tag{73}$$

where in last inequality, we use Lemma B.2, and in the last equality we use the fact that $\hat{\mathbf{z}}_{t+1} = \mathbf{z}_t + \beta(\mathbf{x}_t - \mathbf{z}_t) = \mathbf{z}_{t+1}$, $\hat{\mathbf{y}}_{t+1} = \mathbf{y}_t + \eta(A_{\zeta_t}\mathbf{x}_t - \mathbf{b}_{\zeta_t})$.

Combining (72) and (73), we have that

$$
\begin{aligned}
&\mathbb{E}\left[V_t - V_{t+1}\right] \\
&\geq \widetilde{c}_\beta \mathbb{E}\|\mathbf{z}_{t+1} - \mathbf{z}_t\|^2 + \widetilde{c}_\tau \mathbb{E}\|\mathbf{u}^*(\mathbf{x}_t, \mathbf{y}_t + \eta(A_{\zeta_t}\mathbf{x}_t - \mathbf{b}_{\zeta_t}), \mathbf{z}_t) - \mathbf{x}_t\|^2 + \widetilde{c}_\eta \mathbb{E}\|A\mathbf{x}^*(\mathbf{y}_t + \eta(A_{\zeta_t}\mathbf{x}_t - \mathbf{b}_{\zeta_t}), \mathbf{z}_t) - \mathbf{b}\|^2 \\
&\quad - \frac{1}{2}\lambda\tau^2\sigma_2^2 - \left(1 + \frac{17\|A\|^4}{\gamma_K^2}\right)\eta^2 L^2,
\end{aligned}
$$

holds for both $\|\mathbf{y}_t + \eta(A_{\zeta_t}\mathbf{x}_t - \mathbf{b}_{\zeta_t})\| \leq M_y$ and $\|\mathbf{y}_t + \eta(A_{\zeta_t}\mathbf{x}_t - \mathbf{b}_{\zeta_t})\| > M_y$, which means it holds for $\mathbf{x}_{t+1}, \mathbf{y}_{t+1}, \mathbf{z}_{t+1}$ generated by Algorithm 2. Then we can telescope as before and the convergence result follows.

We also now sketch the argument for the complexity. We have for the gradient of the Moreau envelope that

$$
\begin{aligned}
\frac{1}{\mu^2}\|\nabla\Psi(\mathbf{z}_t)\| &= \|\mathbf{z}_t - \bar{\mathbf{x}}^*(\mathbf{z}_t)\| \\
&\leq \|\mathbf{z}_t - \mathbf{x}^*(\mathbf{y}_t + \eta(A_{\zeta_t}\mathbf{x}_t - \mathbf{b}_{\zeta_t}), z_t)\| + \|\mathbf{x}^*(\mathbf{y}_t + \eta(A_{\zeta_t}\mathbf{x}_t - \mathbf{b}_{\zeta_t}), \mathbf{z}_t) - \bar{\mathbf{x}}^*(\mathbf{z}_t)\| \\
&\leq \|\mathbf{z}_t - \mathbf{x}^*(\mathbf{y}_t + \eta(A_{\zeta_t}\mathbf{x}_t - \mathbf{b}_{\zeta_t}), \mathbf{z}_t)\| + \bar{\sigma}\|A\mathbf{x}^*(\mathbf{y}_t + \eta(A_{\zeta_t}\mathbf{x}_t - \mathbf{b}_{\zeta_t}), \mathbf{z}_s) - \mathbf{b}\| \\
&\leq \|\mathbf{z}_t - \mathbf{x}_t\| + \|\mathbf{x}_t - \mathbf{x}^*(\mathbf{y}_t + \eta(A_{\zeta_t}\mathbf{x}_t - \mathbf{b}_{\zeta_t}), \mathbf{z}_t)\| + \bar{\sigma}\|A\mathbf{x}^*(\mathbf{y}_t + \eta(A_{\zeta_t}\mathbf{x}_t - \mathbf{b}_{\zeta_t}), \mathbf{z}_t) - \mathbf{b}\| \\
&\leq \|\mathbf{z}_t - \mathbf{x}_t\| + \frac{\lambda}{\gamma}\|\mathbf{x}_t - \mathbf{u}^*(\mathbf{x}_t, \mathbf{y}_t + \eta(A_{\zeta_t}\mathbf{x}_t - \mathbf{b}_{\zeta_t}), \mathbf{z}_t)\| + \bar{\sigma}\|A\mathbf{x}^*(\mathbf{y}_t + \eta(A_{\zeta_t}\mathbf{x}_t - \mathbf{b}_{\zeta_t}), \mathbf{z}_t) - \mathbf{b}\|,
\end{aligned}
$$

where the second line is by triangle inequality, the second inequality is by Lemma A.12, and the fourth line is by triangle inequality and the last estimation is by (58).

The rest of the proof for the complexity result proceeds the same as Appendix A.2 up to simple changes in the constants, and hence is omitted. ∎

## C. Proofs for Section 5

**Notation.** Let us note that we define by $\mathbb{E}_{\xi_t}$ the expectation conditioned on all the randomness before $\xi_t$.

### C.1. Proofs for Theorem 5.3

First, with the idea of the STORM estimator of Cutkosky & Orabona (2019), we have the following lemma to control the variance of the stochastic gradient.

**Lemma C.1.** *(from (Cutkosky & Orabona, 2019)) Let Assumption 5.2 hold. We have the estimation of the variance as:*

$$
\mathbb{E}\|\widehat{\nabla}f_{t+1} - \nabla f(\mathbf{x}_{t+1})\|^2 \leq (1-\alpha)^2\mathbb{E}\|\widehat{\nabla}f_t - \nabla f(\mathbf{x}_t)\|^2 + 3(L_0^2 + L_f^2)\mathbb{E}\|\mathbf{x}_{t+1} - \mathbf{x}_t\|^2 + 3\alpha^2\sigma^2.
$$

*Proof.* By the definition of $\widehat{\nabla}f_{t+1}$ in Alg. 3, we have

$$
\begin{aligned}
&\widehat{\nabla}f_{t+1} - \nabla f(\mathbf{x}_{t+1}) \\
&= \nabla f(\mathbf{x}_{t+1}, \xi_{t+1}) + (1-\alpha)(\widehat{\nabla}f_t - \nabla f(\mathbf{x}_t, \xi_{t+1})) - \nabla f(\mathbf{x}_{t+1}) \\
&= \nabla f(\mathbf{x}_{t+1}, \xi_{t+1}) + (1-\alpha)(\widehat{\nabla}f_t - \nabla f(\mathbf{x}_t)) + (1-\alpha)(\nabla f(\mathbf{x}_t) - \nabla f(\mathbf{x}_t, \xi_{t+1})) - \nabla f(\mathbf{x}_{t+1}) \\
&= (1-\alpha)(\widehat{\nabla}f_t - \nabla f(\mathbf{x}_t)) + (1-\alpha)(\nabla f(\mathbf{x}_t) - \nabla f(\mathbf{x}_t, \xi_{t+1})) + \nabla f(\mathbf{x}_{t+1}, \xi_{t+1}) - \nabla f(\mathbf{x}_{t+1}), \quad (74)
\end{aligned}
$$

where in the second equality, we added and subtracted $(1-\alpha)\nabla f(\mathbf{x}_t)$.

Then, we compute the squared norm of (74) and expand to get

$$
\begin{aligned}
&\|\widehat{\nabla}f_{t+1} - \nabla f(\mathbf{x}_{t+1})\|^2 \\
&= (1-\alpha)^2\|\widehat{\nabla}f_t - \nabla f(\mathbf{x}_t)\|^2 + \|(1-\alpha)(\nabla f(\mathbf{x}_t) - \nabla f(\mathbf{x}_t, \xi_{t+1})) + \nabla f(\mathbf{x}_{t+1}, \xi_{t+1}) - \nabla f(\mathbf{x}_{t+1})\|^2 \\
&\quad + 2(1-\alpha)\langle\widehat{\nabla}f_t - \nabla f(\mathbf{x}_t), (1-\alpha)(\nabla f(\mathbf{x}_t) - \nabla f(\mathbf{x}_t, \xi_{t+1})) + \nabla f(\mathbf{x}_{t+1}, \xi_{t+1}) - \nabla f(\mathbf{x}_{t+1})\rangle.
\end{aligned}
$$

Next, we take expectation with respect to the randomness of $\xi_{t+1}$ to obtain

$$\mathbb{E}_{\xi_{t+1}}\|\widehat{\nabla}f_{t+1} - \nabla f(\mathbf{x}_{t+1})\|^2 = (1-\alpha)^2 \mathbb{E}_{\xi_{t+1}}\|\widehat{\nabla}f_t - \nabla f(\mathbf{x}_t)\|^2$$
$$+ \mathbb{E}_{\xi_{t+1}}\|(1-\alpha)(\nabla f(\mathbf{x}_t) - \nabla f(\mathbf{x}_t, \xi_{t+1})) + \nabla f(\mathbf{x}_{t+1}, \xi_{t+1}) - \nabla f(\mathbf{x}_{t+1})\|^2, \quad (75)$$

which is due to $\widehat{\nabla}f_t - \nabla f(\mathbf{x}_t)$ being independent of $\xi_{t+1}$, as well as

$$\mathbb{E}_{\xi_{t+1}}[\nabla f(\mathbf{x}_t) - \nabla f(\mathbf{x}_t, \xi_{t+1})] = 0, \quad \mathbb{E}_{\xi_{t+1}}[\nabla f(\mathbf{x}_{t+1}, \xi_{t+1}) - \nabla f(\mathbf{x}_{t+1})] = 0.$$

Finally, we estimate the last term on the right-hand side of (75):

$$\mathbb{E}_{\xi_{t+1}}\|(1-\alpha)(\nabla f(\mathbf{x}_t) - \nabla f(\mathbf{x}_t, \xi_{t+1})) + \nabla f(\mathbf{x}_{t+1}, \xi_{t+1}) - \nabla f(\mathbf{x}_{t+1})\|^2$$
$$= \mathbb{E}_{\xi_{t+1}}\|\nabla f(\mathbf{x}_{t+1}, \xi_{t+1}) - \nabla f(\mathbf{x}_t, \xi_{t+1}) + \nabla f(\mathbf{x}_t) - \nabla f(\mathbf{x}_{t+1}) + \alpha(f(\mathbf{x}_t, \xi_{t+1}) - \nabla f(\mathbf{x}_t))\|^2$$
$$\leq 3\mathbb{E}_{\xi_{t+1}}\left[\|\nabla f(\mathbf{x}_{t+1}, \xi_{t+1}) - \nabla f(\mathbf{x}_t, \xi_{t+1})\|^2 + \|\nabla f(\mathbf{x}_t) - \nabla f(\mathbf{x}_{t+1})\|^2 + \|\alpha(\nabla f(\mathbf{x}_t, \xi_{t+1}) - \nabla f(\mathbf{x}_t))\|^2\right]$$
$$\leq 3L_0^2\|\mathbf{x}_{t+1} - \mathbf{x}_t\|^2 + 3L_f^2\|\mathbf{x}_t - \mathbf{x}_{t+1}\|^2 + 3\alpha^2\sigma^2,$$

where in the first equality, we rearrange the terms, and in the first inequality, we use Young's inequality. In the second inequality, we use Assumption 5.2, $L_f$-smoothness of $f(\mathbf{x})$ and $\mathbb{E}_\xi\|\nabla f(\mathbf{x}, \xi) - \nabla f(\mathbf{x})\|^2 \leq \sigma^2$. We use this estimation in (75) and take total expectation to get the result. ∎

Let us recall from (18) that

$$\bar{V}_t = K(\mathbf{x}_t, \mathbf{y}_t, \mathbf{z}_t) - 2d(\mathbf{y}_t, \mathbf{z}_t) + 2\Psi(\mathbf{z}_t) + \frac{1}{48(L_0^2 + L_f^2)\tau}\|\widehat{\nabla}f_t - \nabla f(\mathbf{x}_t)\|^2, \quad (76)$$

where (as (22))

$$K(\mathbf{x}, \mathbf{y}, \mathbf{z}) = L_\rho(\mathbf{x}, \mathbf{y}) + \frac{\mu}{2}\|\mathbf{x} - \mathbf{z}\|^2 \quad (77)$$

and $\mathbf{x} \mapsto \nabla K(\mathbf{x}, \mathbf{y}, \mathbf{z})$ is $L_K$-Lipschitz with $L_K = L_f + \rho\|A\| + \mu$ (see also Fact A.1).

We already have the descent-type lemma of $d(\mathbf{y}_t, \mathbf{z}_t)$ and $\Psi(\mathbf{z}_t)$ in Lemma A.8, and only need to show the descent-type lemma of $K(\mathbf{x}_t, \mathbf{y}_t, \mathbf{z}_t)$. We write $K(\mathbf{x}_t, \mathbf{y}_t, \mathbf{z}_t) - K(\mathbf{x}_{t+1}, \mathbf{y}_{t+1}, \mathbf{z}_{t+1})$ as:

$$[K(\mathbf{x}_t, \mathbf{y}_{t+1}, \mathbf{z}_t) - K(\mathbf{x}_{t+1}, \mathbf{y}_{t+1}, \mathbf{z}_t)] + [K(\mathbf{x}_t, \mathbf{y}_t, \mathbf{z}_t) - K(\mathbf{x}_t, \mathbf{y}_{t+1}, \mathbf{z}_t)] + [K(\mathbf{x}_{t+1}, \mathbf{y}_{t+1}, \mathbf{z}_t) - K(\mathbf{x}_{t+1}, \mathbf{y}_{t+1}, \mathbf{z}_{t+1})]$$

and lower bound each term separately in the following lemmas.

**Lemma C.2.** *Let Assumption 1.1 hold. For the iterates generated by Algorithm 3, we have*

$$K(\mathbf{x}_{t+1}, \mathbf{y}_{t+1}, \mathbf{z}_t) - K(\mathbf{x}_t, \mathbf{y}_{t+1}, \mathbf{z}_t) \leq \frac{\tau}{2}\|\nabla f(\mathbf{x}_t) - \widehat{\nabla}f_t\|^2 - \left(\frac{1}{2\tau} - \frac{L_K}{2}\right)\|\mathbf{x}_{t+1} - \mathbf{x}_t\|^2.$$

*Proof.* We have, by smoothness of $K(\cdot, \mathbf{y}_{t+1}, \mathbf{z}_t)$:

$$K(\mathbf{x}_{t+1}, \mathbf{y}_{t+1}, \mathbf{z}_t) \leq K(\mathbf{x}_t, \mathbf{y}_{t+1}, \mathbf{z}_t) + \langle \nabla_\mathbf{x} K(\mathbf{x}_t, \mathbf{y}_{t+1}, \mathbf{z}_t), \mathbf{x}_{t+1} - \mathbf{x}_t\rangle + \frac{L_K}{2}\|\mathbf{x}_{t+1} - \mathbf{x}_t\|^2. \quad (78)$$

We estimate the inner product here as

$$\langle \nabla_\mathbf{x} K(\mathbf{x}_t, \mathbf{y}_{t+1}, \mathbf{z}_t), \mathbf{x}_{t+1} - \mathbf{x}_t\rangle = \langle G(\mathbf{x}_t, \mathbf{y}_{t+1}, \mathbf{z}_t), \mathbf{x}_{t+1} - \mathbf{x}_t\rangle$$
$$+ \langle \nabla_\mathbf{x} K(\mathbf{x}_t, \mathbf{y}_{t+1}, \mathbf{z}_t) - G(\mathbf{x}_t, \mathbf{y}_{t+1}, \mathbf{z}_t), \mathbf{x}_{t+1} - \mathbf{x}_t\rangle. \quad (79)$$

We first have, in view of Alg. 3 that

$$\nabla_\mathbf{x} K(\mathbf{x}_t, \mathbf{y}_{t+1}, \mathbf{z}_t) - G(\mathbf{x}_t, \mathbf{y}_{t+1}, \mathbf{z}_t) = \nabla f(\mathbf{x}_t) - \widehat{\nabla}f_t.$$

The definition of $\mathbf{x}_{t+1}$ in Alg. 3 gives

$$\langle \mathbf{x}_{t+1} - \mathbf{x}_t + \tau G(\mathbf{x}_t, \mathbf{y}_{t+1}, \mathbf{z}_t), \mathbf{x}_t - \mathbf{x}_{t+1}\rangle \geq 0 \iff \langle G(\mathbf{x}_t, \mathbf{y}_{t+1}, \mathbf{z}_t), \mathbf{x}_{t+1} - \mathbf{x}_t\rangle \leq -\frac{1}{\tau}\|\mathbf{x}_{t+1} - \mathbf{x}_t\|^2. \quad (80)$$

Using $\langle \nabla_{\mathbf{x}} K(\mathbf{x}_t, \mathbf{y}_{t+1}, \mathbf{z}_t) - G(\mathbf{x}_t, \mathbf{y}_{t+1}, \mathbf{z}_t), \mathbf{x}_{t+1} - \mathbf{x}_t \rangle \leq \frac{\tau}{2} \|\nabla f(\mathbf{x}_t) - \widehat{\nabla} f_t\|^2 + \frac{1}{2\tau} \|\mathbf{x}_{t+1} - \mathbf{x}_t\|^2$ along with (80) in (79), we have

$$\langle \nabla_{\mathbf{x}} K(\mathbf{x}_t, \mathbf{y}_{t+1}, \mathbf{z}_t), \mathbf{x}_{t+1} - \mathbf{x}_t \rangle \leq \frac{\tau}{2} \|\nabla f(\mathbf{x}_t) - \widehat{\nabla} f_t\|^2 - \frac{1}{2\tau} \|\mathbf{x}_{t+1} - \mathbf{x}_t\|^2.$$

Then the result follows after substituting the last estimate in (78). ∎

**Lemma C.3.** *Let Assumption 1.1 hold. For the iterates generated by Algorithm 3, we have*

$$K(\mathbf{x}_t, \mathbf{y}_t, \mathbf{z}_t) - K(\mathbf{x}_{t+1}, \mathbf{y}_{t+1}, \mathbf{z}_{t+1}) \geq -\eta \|A\mathbf{x}_t - \mathbf{b}\|^2 + \left( \frac{\mu}{\beta} - \frac{3\mu}{4} \right) \|\mathbf{z}_{t+1} - \mathbf{z}_t\|^2 \tag{81}$$
$$- \frac{\tau}{2} \|\nabla f(\mathbf{x}_t) - \widehat{\nabla} f_t\|^2 + \left( \frac{1}{2\tau} - \frac{L_K}{2} - \mu \right) \|\mathbf{x}_{t+1} - \mathbf{x}_t\|^2.$$

*Proof.* First, from the definition of $K$ in (22), we have

$$K(\mathbf{x}_t, \mathbf{y}_t, \mathbf{z}_t) - K(\mathbf{x}_t, \mathbf{y}_{t+1}, \mathbf{z}_t) = -\eta \|A\mathbf{x}_t - \mathbf{b}\|^2.$$

Moreover, it follows that

$$
\begin{aligned}
& K(\mathbf{x}_{t+1}, \mathbf{y}_{t+1}, \mathbf{z}_t) - K(\mathbf{x}_{t+1}, \mathbf{y}_{t+1}, \mathbf{z}_{t+1}) \\
& = \frac{\mu}{2} (\|\mathbf{x}_{t+1} - \mathbf{z}_t\|^2 - \|\mathbf{x}_{t+1} - \mathbf{z}_{t+1}\|^2) \\
& = \frac{\mu}{2} \langle \mathbf{z}_{t+1} - \mathbf{z}_t, 2\mathbf{x}_{t+1} - \mathbf{z}_t - \mathbf{z}_{t+1} \rangle \\
& = \frac{\mu}{2} \langle \mathbf{z}_{t+1} - \mathbf{z}_t, 2\mathbf{x}_{t+1} - 2\mathbf{x}_t + 2\mathbf{x}_t - 2\mathbf{z}_t + \mathbf{z}_t - \mathbf{z}_{t+1} \rangle \\
& = \frac{\mu}{2} \langle \mathbf{z}_{t+1} - \mathbf{z}_t, 2\mathbf{x}_{t+1} - 2\mathbf{x}_t \rangle + \frac{\mu}{2} \langle \mathbf{z}_{t+1} - \mathbf{z}_t, 2\mathbf{x}_t - 2\mathbf{z}_t \rangle - \frac{\mu}{2} \|\mathbf{z}_{t+1} - \mathbf{z}_t\|^2 \\
& \geq -\frac{\mu}{4} \|\mathbf{z}_{t+1} - \mathbf{z}_t\|^2 - \mu \|\mathbf{x}_{t+1} - \mathbf{x}_t\|^2 + \frac{\mu}{\beta} \|\mathbf{z}_t - \mathbf{z}_{t+1}\|^2 - \frac{\mu}{2} \|\mathbf{z}_{t+1} - \mathbf{z}_t\|^2,
\end{aligned}
$$

where the first equality comes from the definition of $K$. In the last inequality, we use $\langle \mathbf{a}, \mathbf{b} \rangle \geq -\frac{1}{4} \|\mathbf{a}\|^2 - \|\mathbf{b}\|^2$ and $\mathbf{x}_t - \mathbf{z}_t = \frac{\mathbf{z}_{t+1} - \mathbf{z}_t}{\beta}$ by the definition of $\mathbf{z}_{t+1}$ in Algorithm 3.

Fanally combining the above two results with Lemma C.2 and combining like-terms yields the claim. ∎

We next follow with a detailed restatement of Lemma 5.3 and its proof.

**Lemma C.4** (cf. Lemma 5.3). *Under Assumption 1.1 and Assumption 5.2, with the parameters chosen as:*

$$\mu = \max\{2, 4L_f\}, \quad \tau \leq \min \left\{ \frac{1}{8L_K + 16\mu}, \frac{1}{\sqrt{48(L_0^2 + L_f^2)}} \right\}$$

$$\eta = \min \left\{ \frac{(\mu - L_f)^2 \tau}{8\|A\|^2}, \frac{2\mu + \rho\|A\|}{4\|A\|^4}, \frac{\tau}{200\|A\|^2}, \frac{\tau(2\mu + \rho\|A\|^2)}{20\|A\|^2} \right\}, \tag{82}$$

$$\beta = \min \left\{ \frac{\tau}{100}, \frac{1}{50}, \frac{\eta}{36\mu\bar{\sigma}^2} \right\},$$

$$\alpha = 48(L_0^2 + L_f^2)\tau^2,$$

*where $L_K = L_f + \rho\|A\| + \mu$, $\bar{\sigma}$ is defined in Lemma A.12, we have*

$$\mathbb{E}\bar{V}_t - \mathbb{E}\bar{V}_{t+1} \geq \frac{\mu}{2\beta} \mathbb{E}\|\mathbf{z}_t - \mathbf{z}_{t+1}\|^2 + \frac{1}{8\tau} \mathbb{E}\|\mathbf{x}_t - \mathbf{x}_{t+1}\|^2 + \frac{\eta}{2} \mathbb{E}\|A\mathbf{x}^*(\mathbf{y}_{t+1}, \mathbf{z}_t) - b\|^2 + \frac{\tau}{4} \mathbb{E}\|\widehat{\nabla} f_t - \nabla f(\mathbf{x}_t)\|^2 \tag{83}$$
$$- 144(L_0^2 + L_f^2)\sigma^2 \tau^3.$$

*Proof.* We denote

$$V_t = K(\mathbf{x}_t, \mathbf{y}_t, \mathbf{z}_t) - 2d(\mathbf{y}_t, \mathbf{z}_t) + 2\Psi(\mathbf{z}_t). \tag{84}$$

Joining (81) with Lemma A.8 (since this lemma only uses the update rules of $\mathbf{y}_{t+1}, \mathbf{z}_{t+1}$ that is common in Alg. 1 and Alg. 3), we have

$$
\begin{aligned}
\mathbb{E}V_t - \mathbb{E}V_{t+1} \geq{} & -\eta\mathbb{E}\|A\mathbf{x}_t - \mathbf{b}\|^2 + \left(\frac{\mu}{\beta} - \frac{3\mu}{4}\right)\mathbb{E}\|\mathbf{z}_{t+1} - \mathbf{z}_t\|^2 \\
& -\frac{\tau}{2}\mathbb{E}\|\nabla f(\mathbf{x}_t) - \widehat{\nabla} f_t\|^2 + \left(\frac{1}{2\tau} - \frac{L_K}{2} - \mu\right)\mathbb{E}\|\mathbf{x}_{t+1} - \mathbf{x}_t\|^2 \\
& + 2\eta\mathbb{E}\langle A\mathbf{x}_t - \mathbf{b}, A\mathbf{x}^*(\mathbf{y}_{t+1}, \mathbf{z}_t) - \mathbf{b}\rangle + \mu\mathbb{E}\langle \mathbf{z}_{t+1} - \mathbf{z}_t, \mathbf{z}_{t+1} + \mathbf{z}_t - 2\mathbf{x}^*(\mathbf{y}_{t+1}, \mathbf{z}_{t+1})\rangle \\
& - 2\mu\mathbb{E}\langle \mathbf{z}_{t+1} - \mathbf{z}_t, \mathbf{z}_t - \bar{\mathbf{x}}^*(\mathbf{z}_t)\rangle - \frac{\mu}{\sigma_4}\mathbb{E}\|\mathbf{z}_t - \mathbf{z}_{t+1}\|^2. 
\end{aligned}
\tag{85}
$$

First, let us combine the first and fifth terms on the right-hand side to obtain

$$-\eta\|A\mathbf{x}_t - \mathbf{b}\|^2 + 2\eta\langle A\mathbf{x}_t - \mathbf{b}, A\mathbf{x}^*(\mathbf{y}_{t+1}, \mathbf{z}_t) - \mathbf{b}\rangle = -\eta\|A\mathbf{x}_t - A\mathbf{x}^*(\mathbf{y}_{t+1}, \mathbf{z}_t)\|^2 + \eta\|A\mathbf{x}^*(\mathbf{y}_{t+1}, \mathbf{z}_t) - \mathbf{b}\|^2. \tag{86}$$

Next, we combine the sixth and seventh terms on the right-hand side of (85) to get

$$
\begin{aligned}
& \mu\langle \mathbf{z}_{t+1} - \mathbf{z}_t, \mathbf{z}_{t+1} + \mathbf{z}_t - 2\mathbf{x}^*(\mathbf{y}_{t+1}, \mathbf{z}_{t+1})\rangle - 2\mu\langle \mathbf{z}_{t+1} - \mathbf{z}_t, \mathbf{z}_t - \bar{\mathbf{x}}^*(\mathbf{z}_t)\rangle \\
& = \mu\langle \mathbf{z}_{t+1} - \mathbf{z}_t, \mathbf{z}_{t+1} - \mathbf{z}_t - 2\mathbf{x}^*(\mathbf{y}_{t+1}, \mathbf{z}_{t+1}) + 2\bar{\mathbf{x}}^*(\mathbf{z}_t)\rangle \\
& = \mu\|\mathbf{z}_{t+1} - \mathbf{z}_t\|^2 + 2\mu\langle \mathbf{z}_{t+1} - \mathbf{z}_t, -\mathbf{x}^*(\mathbf{y}_{t+1}, \mathbf{z}_{t+1}) + \bar{\mathbf{x}}^*(\mathbf{z}_t)\rangle. 
\end{aligned}
\tag{87}
$$

We now single out the inner product in the last equality and estimate it by adding and subtracting $\mathbf{x}^*(\mathbf{y}_{t+1}, \mathbf{z}_t)$ in the second argument of the inner product:

$$
\begin{aligned}
& 2\mu\langle \mathbf{z}_{t+1} - \mathbf{z}_t, -\mathbf{x}^*(\mathbf{y}_{t+1}, \mathbf{z}_{t+1}) + \bar{\mathbf{x}}^*(\mathbf{z}_t)\rangle \\
& = 2\mu\langle \mathbf{z}_{t+1} - \mathbf{z}_t, -\mathbf{x}^*(\mathbf{y}_{t+1}, \mathbf{z}_{t+1}) + \mathbf{x}^*(\mathbf{y}_{t+1}, \mathbf{z}_t)\rangle + 2\mu\langle \mathbf{z}_{t+1} - \mathbf{z}_t, -\mathbf{x}^*(\mathbf{y}_{t+1}, \mathbf{z}_t) + \bar{\mathbf{x}}^*(\mathbf{z}_t)\rangle \\
& \geq -\mu\|\mathbf{z}_{t+1} - \mathbf{z}_t\|^2 - \mu\|\mathbf{x}^*(\mathbf{y}_{t+1}, \mathbf{z}_t) - \mathbf{x}^*(\mathbf{y}_{t+1}, \mathbf{z}_{t+1})\|^2 - \frac{\mu}{\zeta}\|\mathbf{z}_{t+1} - \mathbf{z}_t\|^2 - \mu\zeta\|\bar{\mathbf{x}}^*(\mathbf{z}_t) - \mathbf{x}^*(\mathbf{y}_{t+1}, \mathbf{z}_t)\|, 
\end{aligned}
\tag{88}
$$

for any $\zeta$, where we used Young's inequality twice. Then, we plug this into (87) to obtain

$$
\begin{aligned}
& \mu\langle \mathbf{z}_{t+1} - \mathbf{z}_t, \mathbf{z}_{t+1} + \mathbf{z}_t - 2\mathbf{x}^*(\mathbf{y}_{t+1}, \mathbf{z}_{t+1})\rangle - 2\mu\langle \mathbf{z}_{t+1} - \mathbf{z}_t, \mathbf{z}_t - \bar{\mathbf{x}}^*(\mathbf{z}_t)\rangle \\
& \geq -\frac{\mu}{\sigma_4^2}\|\mathbf{z}_{t+1} - \mathbf{z}_t\|^2 - \frac{\mu}{\zeta}\|\mathbf{z}_{t+1} - \mathbf{z}_t\|^2 - \mu\zeta\|\bar{\mathbf{x}}^*(\mathbf{z}_t) - \mathbf{x}^*(\mathbf{y}_{t+1}, \mathbf{z}_t)\|^2, 
\end{aligned}
\tag{89}
$$

where we use (62) to bound the second term on the right-hand side of (88), with $\sigma_4$ being as (25).

Then we use (86) and (89) in (85) to obtain

$$
\begin{aligned}
\mathbb{E}V_t - \mathbb{E}V_{t+1} \geq{} & \left(\frac{\mu}{\beta} - \frac{3\mu}{4}\right)\mathbb{E}\|\mathbf{z}_{t+1} - \mathbf{z}_t\|^2 - \frac{\tau}{2}\mathbb{E}\|\nabla f(\mathbf{x}_t) - \widehat{\nabla} f_t\|^2 + \left(\frac{1}{2\tau} - \frac{L_K}{2} - \mu\right)\mathbb{E}\|\mathbf{x}_{t+1} - \mathbf{x}_t\|^2 \\
& -\eta\mathbb{E}\|A\mathbf{x}_t - A\mathbf{x}^*(\mathbf{y}_{t+1}, \mathbf{z}_t)\|^2 + \eta\mathbb{E}\|A\mathbf{x}^*(\mathbf{y}_{t+1}, \mathbf{z}_t) - \mathbf{b}\|^2 \\
& -\frac{\mu}{\sigma_4^2}\mathbb{E}\|\mathbf{z}_{t+1} - \mathbf{z}_t\|^2 - \frac{\mu}{\zeta}\mathbb{E}\|\mathbf{z}_{t+1} - \mathbf{z}_t\|^2 - \mu\zeta\|\bar{\mathbf{x}}^*(\mathbf{z}_t) - \mathbf{x}^*(\mathbf{y}_{t+1}, \mathbf{z}_t)\|^2 - \frac{\mu}{\sigma_4}\mathbb{E}\|\mathbf{z}_{t+1} - \mathbf{z}_t\|^2 \\
\geq{} & \left(\frac{\mu}{\beta} - \frac{3\mu}{4} - \frac{\mu}{\sigma_4^2} - \frac{\mu}{\zeta} - \frac{\mu}{\sigma_4}\right)\mathbb{E}\|\mathbf{z}_{t+1} - \mathbf{z}_t\|^2 \\
& -\frac{\tau}{2}\mathbb{E}\|\nabla f(\mathbf{x}_t) - \widehat{\nabla} f_t\|^2 - \frac{2\eta\|A\|^2}{(\mu - L_f)^2}\mathbb{E}\|\nabla f(\mathbf{x}_t) - \widehat{\nabla} f_t\|^2 \\
& + \left(\frac{1}{2\tau} - \frac{L_K}{2} - \mu - \eta\|A\|^2\frac{2}{\tau^2(\mu - L_f)^2}\right)\mathbb{E}\|\mathbf{x}_{t+1} - \mathbf{x}_t\|^2 \\
& + \left(\eta - \mu\zeta\bar{\sigma}^2\right)\mathbb{E}\|A\mathbf{x}^*(\mathbf{y}_{t+1}, \mathbf{z}_t) - \mathbf{b}\|^2, 
\end{aligned}
\tag{90}
$$

where in the last inequality, we use Lem. C.6 and Lem. A.12, then combine the like-terms.

Then we need to estimate the coefficients of each terms in the above inequality. Let us recall from (25) that $\sigma_4 = \frac{\mu - L_f}{\mu} > \frac{1}{2}$ and let $\zeta = 6\beta$.

We now estimate the coefficient of $\mathbb{E}\|\mathbf{z}_t - \mathbf{z}_{t+1}\|^2$ in (90). First, by $\sigma_4 > \frac{1}{2}$, we have $\frac{\mu}{\sigma_4^2} \leq 4\mu$ and $\frac{\mu}{\sigma_4} \leq 2\mu$. By also using $\zeta = 6\beta$, we have:

$$\text{The coefficient of } \mathbb{E}\|\mathbf{z}_t - \mathbf{z}_{t+1}\|^2 \geq \frac{\mu}{\beta} - \frac{3\mu}{4} - 4\mu - \frac{\mu}{6\beta} - 2\mu.$$

Using $\beta \leq 1/50$, we obtain $(\frac{3}{4} + 4 + 2)\mu \leq \frac{\mu}{5\beta}$, then we estimate:

$$\text{The coefficient of } \mathbb{E}\|\mathbf{z}_t - \mathbf{z}_{t+1}\|^2 \geq \frac{\mu}{\beta} - \frac{\mu}{5\beta} - \frac{\mu}{6\beta} \geq \frac{\mu}{2\beta}.$$

We move on to estimating the coefficient of $\mathbb{E}\|\mathbf{x}_t - \mathbf{x}_{t+1}\|^2$ in (90). With $\eta \leq \frac{(\mu - L_f)^2 \tau}{8\|A\|^2}$, we have $2\eta\|A\|^2 \frac{1}{\tau^2(\mu - L_f)^2} \leq \frac{1}{4\tau}$, we have:

$$\text{The coefficient of } \mathbb{E}\|\mathbf{x}_t - \mathbf{x}_{t+1}\|^2 \geq \frac{1}{4\tau} - \frac{L_K}{2} - \mu.$$

Last, we work on the coefficient of $\mathbb{E}\|A\mathbf{x}^*(\mathbf{y}_{t+1}, \mathbf{z}_t) - \mathbf{b}\|^2$ in (90). Because $\zeta = 6\beta$, it follows that $\eta - \mu\zeta\bar{\sigma}^2 = \eta - 6\mu\beta\bar{\sigma}^2$. With $\beta \leq \frac{\eta}{36\mu\bar{\sigma}^2}$, we have $6\mu\beta\bar{\sigma}^2 \leq \frac{\eta}{6}$, then we estimate:

$$\text{the coefficient of } \mathbb{E}\|A\mathbf{x}^*(\mathbf{y}_{t+1}, \mathbf{z}_t) - \mathbf{b}\|^2 \geq \eta - \frac{\eta}{6} \geq \frac{\eta}{2}.$$

Next, we estimate the coefficient of $\mathbb{E}\|\nabla f(\mathbf{x}_t) - \widehat{\nabla} f_t\|^2$. With $\eta \leq \frac{(\mu - L_f)^2 \tau}{8\|A\|^2}$, we have $-\frac{\tau}{2} - \frac{2\eta\|A\|^2}{(\mu - L_f)^2} \geq -\frac{3}{4}\tau$. Finally, we have

$$\mathbb{E}V_t - \mathbb{E}V_{t+1}$$

$$\geq \frac{\mu}{2\beta}\mathbb{E}\|\mathbf{z}_t - \mathbf{z}_{t+1}\|^2 + \left(\frac{1}{4\tau} - \frac{L_K}{2} - \mu\right)\mathbb{E}\|\mathbf{x}_t - \mathbf{x}_{t+1}\|^2 + \frac{\eta}{2}\mathbb{E}\|A\mathbf{x}^*(\mathbf{y}_{t+1}, \mathbf{z}_t) - \mathbf{b}\|^2 - \frac{3\tau}{4}\mathbb{E}\|\nabla f(\mathbf{x}_t) - \widehat{\nabla} f_t\|^2$$

$$= \frac{\mu}{2\beta}\mathbb{E}\|\mathbf{z}_t - \mathbf{z}_{t+1}\|^2 + \left(\frac{1}{4\tau} - \frac{L_K}{2} - \mu\right)\mathbb{E}\|\mathbf{x}_t - \mathbf{x}_{t+1}\|^2 + \frac{\eta}{2}\mathbb{E}\|A\mathbf{x}^*(\mathbf{y}_{t+1}, \mathbf{z}_t) - \mathbf{b}\|^2$$

$$+ \frac{\tau}{4}\mathbb{E}\|\nabla f(\mathbf{x}_t) - \widehat{\nabla} f_t\|^2 - \tau\mathbb{E}\|\nabla f(\mathbf{x}_t) - \widehat{\nabla} f_t\|^2.$$

Then recalling Lemma C.1 and assuming $0 < \alpha \leq 1$, we have

$$\mathbb{E}\|\widehat{\nabla} f_{t+1} - \nabla f(\mathbf{x}_{t+1})\|^2 \leq (1 - \alpha)\mathbb{E}\|\widehat{\nabla} f_t - \nabla f(\mathbf{x}_t)\|^2 + 3(L_0^2 + L_f^2)\mathbb{E}\|\mathbf{x}_{t+1} - \mathbf{x}_t\|^2 + 3\alpha^2\sigma^2. \tag{91}$$

We multiply (91) by $\frac{\tau}{\alpha}$, rearrange, and plug into (91), to get

$$\mathbb{E}V_t - \mathbb{E}V_{t+1} \geq \frac{\mu}{2\beta}\mathbb{E}\|\mathbf{z}_t - \mathbf{z}_{t+1}\|^2 + \left(\frac{1}{4\tau} - \frac{L_K}{2} - \mu\right)\mathbb{E}\|\mathbf{x}_t - \mathbf{x}_{t+1}\|^2$$

$$+ \frac{\eta}{2}\mathbb{E}\|A\mathbf{x}^*(\mathbf{y}_{t+1}, \mathbf{z}_t) - \mathbf{b}\|^2 + \frac{\tau}{4}\mathbb{E}\|\nabla f(\mathbf{x}_t) - \widehat{\nabla} f_t\|^2$$

$$+ \frac{\tau}{\alpha}\mathbb{E}\|\widehat{\nabla} f_{t+1} - \nabla f(\mathbf{x}_{t+1})\|^2 - \frac{\tau}{\alpha}\mathbb{E}\|\widehat{\nabla} f_t - \nabla f(\mathbf{x}_t)\|^2$$

$$- \frac{3(L_0^2 + L_f^2)\tau}{\alpha}\mathbb{E}\|\mathbf{x}_t - \mathbf{x}_{t+1}\|^2 - 3\alpha\sigma^2\tau. \tag{92}$$

Because $\alpha = 48(L_0^2 + L_f^2)\tau^2$ and $\tau \leq \min\left\{\frac{1}{8L_K + 16\mu}, \frac{1}{\sqrt{48(L_0^2 + L_f^2)}}\right\}$, we obtain

$$\frac{L_K}{2} + \mu \leq \frac{1}{16\tau}, \qquad \frac{3(L_0^2 + L_f^2)\tau}{\alpha} = \frac{1}{16\tau}.$$

Hence, we have

$$\mathbb{E}V_t - \mathbb{E}V_{t+1} \geq \frac{\mu}{2\beta}\mathbb{E}\|\mathbf{z}_t - \mathbf{z}_{t+1}\|^2 + \frac{1}{8\tau}\mathbb{E}\|\mathbf{x}_t - \mathbf{x}_{t+1}\|^2 + \frac{\eta}{2}\mathbb{E}\|A\mathbf{x}^*(\mathbf{y}_{t+1}, \mathbf{z}_t) - \mathbf{b}\|^2 + \frac{\tau}{4}\mathbb{E}\|\nabla f(\mathbf{x}_t) - \widehat{\nabla} f_t\|^2$$

$$+ \frac{1}{48(L_0^2 + L_f^2)\tau}\mathbb{E}\|\widehat{\nabla} f_{t+1} - \nabla f(\mathbf{x}_{t+1})\|^2 - \frac{1}{48(L_0^2 + L_f^2)\tau}\mathbb{E}\|\widehat{\nabla} f_t - \nabla f(\mathbf{x}_t)\|^2 - 144(L_0^2 + L_f^2)\sigma^2\tau^3.$$

Finally, we move $\frac{1}{48(L_0^2+L_f^2)\tau}\mathbb{E}\|\widehat{\nabla} f_{t+1} - \nabla f(\mathbf{x}_{t+1})\|^2 - \frac{1}{48(L_0^2+L_f^2)\tau}\mathbb{E}\|\widehat{\nabla} f_t - \nabla f(\mathbf{x}_t)\|^2$ to the left-hand side of the above inequality and use the definition of $\bar{V}_t$ in (18) to get the desired result. ∎

## C.2. Proofs for Theorem 5.4

First, we need two lemmas for the error bound that helps us analyze the sample complexity that we include for being self-contained.

**Lemma C.5.** *(Zhang & Luo, 2020, Lemma 3.10) Under Assumption 1.1, we have*

$$\|\mathbf{x} - \mathrm{proj}_X(\mathbf{x} - \tau\nabla K(\mathbf{x}, \mathbf{y}, \mathbf{z}))\| \geq \frac{\tau(\mu - L_f)}{2}\|\mathbf{x} - \mathbf{x}^*(\mathbf{y}, \mathbf{z})\|,$$

*where $K(\mathbf{x}, \mathbf{y}, \mathbf{z}) = L_\rho(\mathbf{x}, \mathbf{y}) + \frac{\mu}{2}\|\mathbf{x} - \mathbf{z}\|^2$, and $\mathbf{x}^*(\mathbf{y}, \mathbf{z}) = \arg\min_{\mathbf{x} \in X} K(\mathbf{x}, \mathbf{y}, \mathbf{z})$.*

*Proof.* First, we denote that $\hat{\mathbf{x}} = \mathbf{x} - \mathrm{proj}_X(\mathbf{x} - \tau\nabla K(\mathbf{x}, \mathbf{y}, \mathbf{z}))$, then by the definition of $\mathbf{x}^*(\mathbf{y}, \mathbf{z})$, we have

$$\langle \mathbf{x} - \hat{\mathbf{x}} - \mathbf{x}^*(\mathbf{y}, \mathbf{z}), \tau\nabla K(\mathbf{x}^*(\mathbf{y}, \mathbf{z}), \mathbf{y}, \mathbf{z})\rangle \geq 0,$$

where we use the fact that $\mathbf{x} - \hat{\mathbf{x}} \in X$.

Then by the definition of projection (that is, $\bar{\mathbf{z}} = \mathrm{proj}_X(\mathbf{z}) \iff \langle \bar{\mathbf{z}} - \mathbf{z}, \mathbf{t} - \bar{\mathbf{z}}\rangle \geq 0 \; \forall \mathbf{t} \in X$), the definition of $\hat{\mathbf{x}}$, and $\mathbf{x}^*(\mathbf{y}, \mathbf{z}) \in X$, we have

$$\langle \mathbf{x}^*(\mathbf{y}, \mathbf{z}) - \mathrm{proj}_X(\mathbf{x} - \tau\nabla K(\mathbf{x}, \mathbf{y}, \mathbf{z})), \mathbf{x} - \tau\nabla K(\mathbf{x}, \mathbf{y}, \mathbf{z}) - \mathrm{proj}_X(\mathbf{x} - \tau\nabla K(\mathbf{x}, \mathbf{y}, \mathbf{z}))\rangle$$
$$= \langle \mathbf{x}^*(\mathbf{y}, \mathbf{z}) - (\mathbf{x} - \hat{\mathbf{x}}), -\tau\nabla K(\mathbf{x}, \mathbf{y}, \mathbf{z}) + \hat{\mathbf{x}}\rangle \leq 0.$$

Combining above two inequalities and rearranging terms, we have

$$\langle \mathbf{x} - \mathbf{x}^*(\mathbf{y}, \mathbf{z}), \tau\nabla K(\mathbf{x}, \mathbf{y}, \mathbf{z}) - \tau\nabla K(\mathbf{x}^*(\mathbf{y}, \mathbf{z}), \mathbf{y}, \mathbf{z})\rangle$$
$$\leq \langle \hat{\mathbf{x}}, \tau\nabla K(\mathbf{x}, \mathbf{y}, \mathbf{z}) - \tau\nabla K(\mathbf{x}^*(\mathbf{y}, \mathbf{z}), \mathbf{y}, \mathbf{z}) + \mathbf{x} - \mathbf{x}^*(\mathbf{y}, \mathbf{z})\rangle - \|\hat{\mathbf{x}}\|^2$$
$$\leq \|\hat{\mathbf{x}}\|\|\tau\nabla K(\mathbf{x}, \mathbf{y}, \mathbf{z}) - \tau\nabla K(\mathbf{x}^*(\mathbf{y}, \mathbf{z}), \mathbf{y}, \mathbf{z}) + \mathbf{x} - \mathbf{x}^*(\mathbf{y}, \mathbf{z})\|$$
$$\leq \|\hat{\mathbf{x}}\|(\tau L_K + 1)\|\mathbf{x} - \mathbf{x}^*(\mathbf{y}, \mathbf{z})\|$$
$$\leq 2\|\hat{\mathbf{x}}\|\|\mathbf{x} - \mathbf{x}^*(\mathbf{y}, \mathbf{z})\|,$$

where in the second inequality we use the Cauchy-Schwarz inequality and in the last inequality we use the Lipschitz continuity of $\nabla K$ with respect to $\mathbf{x}$.

By $K(\mathbf{x}, \mathbf{y}, \mathbf{z})$ being $(\mu - L_f)$-strongly convex with respect to $\mathbf{x}$ (see Fact A.1), we have

$$\langle \mathbf{x} - \mathbf{x}^*(\mathbf{y}, \mathbf{z}), \tau\nabla K(\mathbf{x}, \mathbf{y}, \mathbf{z}) - \tau\nabla K(\mathbf{x}^*(\mathbf{y}, \mathbf{z}), \mathbf{y}, \mathbf{z})\rangle \geq \tau(\mu - L_f)\|\mathbf{x} - \mathbf{x}^*(\mathbf{y}, \mathbf{z})\|^2.$$

Then, the desired result follows by combining the above two inequalities and using the definition of $\hat{\mathbf{x}} = \mathbf{x} - \mathrm{proj}_X(\mathbf{x} - \tau\nabla K(\mathbf{x}, \mathbf{y}, \mathbf{z}))$. ∎

With the next lemma, we proceed to prove that $\|\mathbf{x}_t - \mathbf{x}^*(\mathbf{y}_{t+1}, \mathbf{z}_t)\|$ is bounded by a combination of $\|\mathbf{x}_t - \mathbf{x}_{t+1}\|$ and $\|\widehat{\nabla} f_t - \nabla f(\mathbf{x}_t)\|$.

**Lemma C.6.** *Under Assumption 1.1, for the iterates generated by Algorithm 3 we have*

$$\|\mathbf{x}_t - \mathbf{x}^*(\mathbf{y}_{t+1}, \mathbf{z}_t)\| \leq \frac{2}{\tau(\mu - L_f)}\|\mathbf{x}_t - \mathbf{x}_{t+1}\| + \frac{2}{(\mu - L_f)}\|\widehat{\nabla} f_t - \nabla f(\mathbf{x}_t)\|.$$

*Proof.* Taking $\mathbf{x}, \mathbf{y}, \mathbf{z}$ as $\mathbf{x}_t, \mathbf{y}_{t+1}, \mathbf{z}_t$ in Lemma C.5, we have

$$
\begin{aligned}
\|\mathbf{x}_t - \mathbf{x}^*(\mathbf{y}_{t+1}, \mathbf{z}_t)\| &\leq \frac{2}{\tau(\mu - L_f)}\|\mathbf{x}_t - \mathrm{proj}_X(\mathbf{x}_t - \tau\nabla K(\mathbf{x}, \mathbf{y}_{t+1}, \mathbf{z}_t))\| \\
&\leq \frac{2}{\tau(\mu - L_f)}\|\mathbf{x}_t - \mathrm{proj}_X(\mathbf{x}_t - \tau G(\mathbf{x}_t, \mathbf{y}_{t+1}, \mathbf{z}_t))\| \\
&\quad + \frac{2}{\tau(\mu - L_f)}\|\mathrm{proj}_X(\mathbf{x}_t - \tau\nabla K(\mathbf{x}_t, \mathbf{y}_{t+1}, \mathbf{z}_t)) - \mathrm{proj}_X(\mathbf{x}_t - \tau G(\mathbf{x}_t, \mathbf{y}_{t+1}, \mathbf{z}_t))\| \\
&\leq \frac{2}{\tau(\mu - L_f)}\|\mathbf{x}_t - \mathbf{x}_{t+1}\| + \frac{2}{(\mu - L_f)}\|\widehat{\nabla}f_t - \nabla f(\mathbf{x}_t)\|,
\end{aligned}
$$

where the second inequality comes form triangle inequality and the last inequality comes from the fact that $\mathrm{proj}_X$ is nonexpansive and $\nabla K(\mathbf{x}_t, \mathbf{y}_{t+1}, \mathbf{z}_t) - G(\mathbf{x}_t, \mathbf{y}_{t+1}, \mathbf{z}_t) = \nabla f(\mathbf{x}_t) - \widehat{\nabla}f_t$. ∎

We now continue with the proof of Theorem 5.4.

*Proof of Theorem 5.4.* Because $\mathbf{z}_{t+1} - \mathbf{z}_t = \beta(\mathbf{x}_t - \mathbf{z}_t)$, $\frac{\mu\beta}{2} = \Theta(\tau)$ and $\frac{\eta}{2} = \Theta(\tau)$ in view of Lemma C.4, hence there exists a constant $C$ such that we get from (83):

$$
\begin{aligned}
\mathbb{E}\bar{V}_t - \mathbb{E}\bar{V}_{t+1} &\geq C\tau\{\mathbb{E}\|\mathbf{x}_t - \mathbf{z}_t\|^2 + \mathbb{E}\|\tau^{-1}(\mathbf{x}_t - \mathbf{x}_{t+1})\|^2 + \mathbb{E}\|A\mathbf{x}^*(\mathbf{y}_{t+1}, \mathbf{z}_t) - \mathbf{b}\|^2 + \mathbb{E}\|\nabla f(\mathbf{x}_t) - \widehat{\nabla}f_t\|^2\} \\
&\quad - 144\left(L_0^2 + L_f^2\right)\sigma^2\tau^3.
\end{aligned} \tag{93}
$$

Then, summing up (93) over $t = 0, 1, \ldots, T-1$, we have

$$
\begin{aligned}
\bar{V}_0 - \mathbb{E}\bar{V}_T &\geq \sum_{t=0}^{T-1} C\tau\{\mathbb{E}\|\mathbf{x}_t - \mathbf{z}_t\|^2 + \mathbb{E}\|\tau^{-1}(\mathbf{x}_t - \mathbf{x}_{t+1})\|^2 + \mathbb{E}\|A\mathbf{x}^*(\mathbf{y}_{t+1}, \mathbf{z}_t) - \mathbf{b}\|^2 + \mathbb{E}\|\nabla f(\mathbf{x}_t) - \widehat{\nabla}f_t\|^2\} \\
&\quad - 144\left(L_0^2 + L_f^2\right)\sigma^2\tau^3 T.
\end{aligned} \tag{94}
$$

From the definition, we have $K(\mathbf{x}, \mathbf{y}, \mathbf{z}) \geq d(\mathbf{y}, \mathbf{z})$ (since $d(\mathbf{y}, \mathbf{z}) = \min_{\mathbf{x}\in X} K(\mathbf{x}, \mathbf{y}, \mathbf{z})$) and $\Psi(\mathbf{z}) \geq d(\mathbf{y}, \mathbf{z})$ (see also Lemma A.13), then

$$
V_t = K(\mathbf{x}_t, \mathbf{y}_t, \mathbf{z}_t) - 2d(\mathbf{y}_t, \mathbf{z}_t) + 2\Psi(\mathbf{z}_t) \geq \Psi(\mathbf{z}_t) \geq \underline{f}.
$$

Consequently, we have

$$
\bar{V}_t = K(\mathbf{x}_t, \mathbf{y}_t, \mathbf{z}_t) - 2d(\mathbf{y}_t, \mathbf{z}_t) + 2\Psi(\mathbf{z}_t) + \frac{1}{48(L_0^2 + L_f^2)\tau}\mathbb{E}\|\widehat{\nabla}f_t - \nabla f(\mathbf{x}_t)\|^2 \geq \underline{f}. \tag{95}
$$

Let $\tau = T^{-1/3}$ and use mini-batch in the initial step where we will have $\mathbb{E}\|\widehat{\nabla}f_0 - \nabla f(\mathbf{x}_0)\|^2 \leq T^{-1/3}\sigma^2$ (by the definition of $\widehat{\nabla}f_0$ and a standard computation), then

$$
\begin{aligned}
\bar{V}_0 &= K(\mathbf{x}_0, \mathbf{y}_0, \mathbf{z}_0) - 2d(\mathbf{y}_0, \mathbf{z}_0) + 2\Psi(\mathbf{z}_0) + \frac{1}{48(L_0^2 + L_f^2)\tau}\mathbb{E}\|\widehat{\nabla}f_0 - \nabla f(\mathbf{x}_0)\|^2 \\
&\leq K(\mathbf{x}_0, \mathbf{y}_0, \mathbf{z}_0) - 2d(\mathbf{y}_0, \mathbf{z}_0) + 2\Psi(\mathbf{z}_0) + \frac{\sigma^2}{48(L_0^2 + L_f^2)},
\end{aligned} \tag{96}
$$

where the right-hand is proportional to a constant independent of $T$, we denote it as $C_0$.

Combining (94) with (95) and (96), we have

$$
\begin{aligned}
&\frac{1}{T}\sum_{t=0}^{T-1} C\{\mathbb{E}\|\mathbf{x}_t - \mathbf{z}_t\|^2 + \mathbb{E}\|\tau^{-1}(\mathbf{x}_t - \mathbf{x}_{t+1})\|^2 + \mathbb{E}\|A\mathbf{x}^*(\mathbf{y}_{t+1}, \mathbf{z}_t) - \mathbf{b}\|^2 + \mathbb{E}\|\nabla f(\mathbf{x}_t) - \widehat{\nabla}f_t\|^2\} \\
&\leq T^{-2/3}\left(C_0 - \underline{f} + 144(L_0^2 + L_f^2)\sigma^2\right).
\end{aligned} \tag{97}
$$

Then, for index $s$ selected uniformly at random from $\{0, 1, ..., T-1\}$, we have

$$
\begin{aligned}
\mathbb{E}\|\mathbf{x}_s - \mathbf{z}_s\|^2 = O(T^{-2/3}), \quad & \mathbb{E}\|\tau^{-1}(\mathbf{x}_s - \mathbf{x}_{s+1})\|^2 = O(T^{-2/3}), \\
\mathbb{E}\|A\mathbf{x}^*(\mathbf{y}_{s+1}, \mathbf{z}_s) - \mathbf{b}\|^2 = O(T^{-2/3}), \quad & \mathbb{E}\|\nabla f(\mathbf{x}_t) - \widehat{\nabla} f_t\|^2 = O(T^{-2/3}).
\end{aligned}
\tag{98}
$$

According to Algorithm 3, we have

$$
\mathbf{x}_{s+1} = \arg\min_{\mathbf{x}} \left\{ \langle G(\mathbf{x}_s, \mathbf{y}_{s+1}, \mathbf{z}_s), \mathbf{x} - \mathbf{x}^s \rangle + \frac{1}{\tau}\|\mathbf{x} - \mathbf{x}_s\|^2 + \partial I_X(\mathbf{x}) \right\}.
$$

By the definition of $\mathbf{x}_{s+1}$, we have

$$
0 \in G(\mathbf{x}_s, \mathbf{y}_{s+1}, \mathbf{z}_s) + \frac{2}{\tau}(\mathbf{x}_{s+1} - \mathbf{x}_s) + \partial I_X(\mathbf{x}_{s+1}).
\tag{99}
$$

We now set

$$
\mathbf{v} = \nabla_{\mathbf{x}} K(\mathbf{x}_{s+1}, \mathbf{y}_{s+1}, \mathbf{z}_s) - G(\mathbf{x}_s, \mathbf{y}_{s+1}, \mathbf{z}_s) - \frac{2}{\tau}(\mathbf{x}_{s+1} - \mathbf{x}_s) - \rho A^\top(A\mathbf{x}_{s+1} - \mathbf{b}) - \mu(\mathbf{x}_{s+1} - \mathbf{z}_s).
$$

Now, by using the definition of $K(\mathbf{x}, \mathbf{y}, \mathbf{z})$ from (19) and (99), we obtain (cf. (5))

$$
\mathbf{v} \in \nabla f(\mathbf{x}_{s+1}) + A^\top \mathbf{y}_{s+1} + \partial I_X(\mathbf{x}_{s+1})
$$

We now derive the guarantees on the feasibility and the norm of $\mathbf{v}$. First, by triangle inequality, we have

$$
\begin{aligned}
\|A\mathbf{x}_{s+1} - \mathbf{b}\| &\leq \|A\mathbf{x}^*(\mathbf{y}_{s+1}, \mathbf{z}_s) - \mathbf{b}\| + \|A\mathbf{x}_{s+1} - A\mathbf{x}_s\| + \|A(\mathbf{x}_s - \mathbf{x}^*(\mathbf{y}_{s+1}, \mathbf{z}_s))\| \\
&\leq \|A\mathbf{x}^*(\mathbf{y}_{s+1}, \mathbf{z}_s) - \mathbf{b}\| + \|A\|\|\mathbf{x}_{s+1} - \mathbf{x}_s\| + \frac{2\|A\|}{\tau(\mu - L_f)}\|\mathbf{x}_s - \mathbf{x}_{s+1}\| + \frac{2\|A\|}{\mu - L_f}\|\widehat{\nabla} f_s - \nabla f(\mathbf{x}_s)\| \\
&= O(T^{-1/3}),
\end{aligned}
\tag{100}
$$

where in the second inequality, we use Lemma C.6 and the last estimate uses (98).

Then, we have by triangle inequality that

$$
\begin{aligned}
\|\mathbf{v}\| &\leq \|\nabla_{\mathbf{x}} K(\mathbf{x}_{s+1}, \mathbf{y}_{s+1}, \mathbf{z}_s) - \nabla_{\mathbf{x}} K(\mathbf{x}_s, \mathbf{y}_{s+1}, \mathbf{z}_s)\| + \|\nabla_{\mathbf{x}} K(\mathbf{x}_s, \mathbf{y}_{s+1}, \mathbf{z}_s) - G(\mathbf{x}_s, \mathbf{y}_{s+1}, \mathbf{z}_s)\| \\
&\quad + \frac{2}{\tau}\|\mathbf{x}_{s+1} - \mathbf{x}_s\| + \rho\|A\|\|A\mathbf{x}_{s+1} - \mathbf{b}\| + \mu\|\mathbf{x}_{s+1} - \mathbf{z}_s\| \\
&\leq \left(L_K + \frac{2}{\tau}\right)\|\mathbf{x}_{s+1} - \mathbf{x}_s\| + \|\nabla f(\mathbf{x}_s) - \widehat{\nabla} f_s\| + \rho\|A\|\|A\mathbf{x}_{s+1} - \mathbf{b}\| + \mu\left(\|\mathbf{x}_s - \mathbf{z}_s\| + \|\mathbf{x}_{s+1} - \mathbf{x}_s\|\right) \\
&= O(T^{-1/3}),
\end{aligned}
$$

where in first inequality, we introduce a term $\nabla_{\mathbf{x}} K(\mathbf{x}_s, \mathbf{y}_{s+1}, \mathbf{z}_s)$ and then use triangle inequality. The second inequality used Lipschitzness of $K$, the definition of $G$, and the triangle inequality. The last step uses (98) and (100) and $\rho = O(1)$ since it is chosen arbitrarily in Alg. 3. ∎

