# OpenReview forum: "Stochastic Smoothed Primal-Dual Algorithms for Nonconvex Optimization with Linear Inequality Constraints"
_ICML.cc/2025/Conference — ICML 2025 spotlightposter_

### Official Review · Reviewer_1NyX · 2025-02-23

**Overall Recommendation:** 2

**Summary:**

This paper investigates a single-loop ALM-type algorithm for solving linearly constrained nonconvex optimization problems. The framework accommodates both stochastic objective functions and stochastic constraints. Theoretical analysis demonstrates that the proposed algorithm achieves competitive complexity.

**Claims And Evidence:**

Yes, it is clear.

**Essential References Not Discussed:**

The literature is well discussed.

**Experimental Designs Or Analyses:**

No experiments.

**Methods And Evaluation Criteria:**

The complexity measure studied is standand.

**Other Comments Or Suggestions:**

N/A

**Other Strengths And Weaknesses:**

Although this is a theoretical paper, validating the proposed algorithm with numerical examples would strengthen its contributions. Additionally, the differences from previous work should be highlighted, as this paper heavily relies on the smoothed technique, which has been extensively studied in analysis.

**Questions For Authors:**

1. Are inequality constraints handled solely through projection? From the problem formulation, it appears that they can only be incorporated into the set $X$. Could you clarify this?

2. The paper states that the key difference from Alacaoglu \& Wright (2024) is the inclusion of additional constraints in $X$. However, the complexity result in Alacaoglu \& Wright (2024) is $O(\epsilon^{-3})$, which is better than the result presented in this paper. Could you provide further explanation on this? Is there potential to improve the complexity in this paper, and what are the underlying reasons?

3. I notice that in Theorem 3.1, the stepsizes $\tau$, $\eta$, and $\beta$ decrease as the number of iterations $T$ increases. Would this lead to impractically small stepsizes in real applications? How should these parameters be set in practical implementations?

4. How should the threshold $M_y$ be chosen in Algorithm 2? If it is set too small, $y$ may always remain at zero, potentially causing issues with the feasibility update. Could you provide guidance on selecting an appropriate value for $M_y$?

**Relation To Broader Scientific Literature:**

This paper builds upon the approach of Zhang & Luo (2020, 2022) and extends it to the stochastic setting.

**Theoretical Claims:**

Yes, I think the main theorem makes sense.

---

> ### Author Rebuttal · Authors · 2025-04-01
>
> We thank the reviewer for their careful review and their detailed questions. We really appreciate their effort in evaluating our work.
>
> > Numerical examples
>
> We ran a preliminary experiment to validate our theory. Please see our response to Reviewer WK98.
>
> > Additionally, the differences from previous work should be highlighted, as this paper heavily relies on the smoothed technique, which has been extensively studied in analysis.
>
> We want to highlight that, to our knowledge, this is the first algorithm achieving the optimal $\mathcal{O}(\epsilon^{-4})$ sample complexity for stochastic nonconvex optimization problems with stochastic constraints using only one sample per iteration.
>
> First, in view of [1], which works for unconstrained min-max problems, the descent property cannot be guaranteed when their idea is applied to constrained problems. To address this issue, we introduce a potential function specifically designed for constrained settings, ensuring the descent property. We also use the gradient of the Moreau envelope, in place of the function $\nabla K$ used in works of Zhang and Luo, see for example Lem A.7 and A.8 and the estimations using these.  We note that projection causes extra difficulties with stochastic gradients, see for example the work of Davis&Drusvyatskiy, 2019.
>
> Furthermore, in the case of stochastic linear constraints, we propose a novel technique "pulling back the dual variable", which ensures boundedness of the dual variables. This property is critical but cannot be guaranteed by the deterministic analysis framework.
>
> We will add these descriptions in the final version.
>
> [1] Yang et al. Faster Single-loop Algorithms for Minimax Optimization without Strong Concavity, AISTATS 2022
>
> > Inequality constraints
>
> Thanks for the good question (please also see our response to Reviewer srvF). We clarify that our work can solve problems with constraint $\{x: Ax\leq b\}$ by only projecting on the set $\{x: x\geq 0\}$ and multiplying with $A, A^T$.
>
> In fact, the assumption on projectability of $X$ is without loss of generality.
>
> Suppose that ${x: Hx\le h}$ is difficult to project. We can add slack variables to reformulate this problem to make the set defined by inequality constraints easy to project. Specifically, we can rewrite the original problem as
>
> $$ \min_{x, s} f(x): Ax = b, Hx + s = h, s \geq 0 $$
> where $s$ is the slack variable.Then the set defined by inequality constraints  $\{(x,s)\mid s\ge 0\}$ is easy to project and then we can handle the equality constraints $Ax=b,Hx+s=h$  by using simple products $A, A^T, H, H^T$, due to our algorithm design.
>
> In summary, our assumption on $X$ being projectable is without loss of generality and for any inequality constrained problem, we can introduce slack variables so we only require projecting onto the simple set $\{x: x\ge 0\}$
>
> > Comparisons with Alacaoglu&Wright, 2024
>
> The sample complexity $O(1/\epsilon^4)$ achieved by our algorithm is already optimal under our assumptions and oracle model, as established by [2] ([2] shows $O(1/\epsilon^4)$ is even optimal for unconstrained problems). The improved complexity in Alacaoglu & Wright (2024) relies on a stronger stochastic oracle and additional smoothness assumptions. Specifically, our oracle assumes access to only one stochastic gradient $\nabla f(x, \xi)$ at a single query point $x$ per iteration, with $\xi$ sampled from the distribution $P$. Under this oracle, the lower bound is $\Omega(1/\epsilon^4)$ [2], matching our complexity.
>
> However, Alacaoglu&Wright, 2024 require a stronger oracle: they assume availability of two stochastic gradients $\nabla f(x;\xi)$ and $\nabla f(y;\xi)$ evaluated at two different points $x,y$ for the same sample $\xi$. Additionally, they make a stronger smoothness assumption:
>
> $\mathbb{E}_{\xi}\|\nabla f(x,\xi)-\nabla f(y,\xi)\|^2\leq L^2\|x-y\|^2$,
>
> which is not required in our analysis.
>
> [2]Arjevani et al. Lower bounds for non-convex stochastic optimization. Math. Program. 2023.
>
> In addition, under the same stochastic oracle and the additional smoothness assumption, we can achieve the complexity of $O(\varepsilon^{-3})$ by combining our ALM framework with the STORM variance reduction. This is a straightforward extension of our techniques and we can provide a proof sketch in the discussion stage if the reviewer wishes. We will add this result in the final version. The only difference in this algorithm would be in the definition of $G$:
>
> $$G(x_t, y_{t+1}, z_t) = F_t + A^{\top}(Ax_t - b) + \lambda(x_t - z_t)$$
>
> where
>
> $$F_{t+1} = \nabla f(x_{t+1}, \xi_{t+1}) + (1 - \alpha) (F_t - \nabla f(x_t, \xi_{t+1}))$$
>
> > Step sizes
>
> Our stepsizes are chosen as order $1/\sqrt{T}$, as standard in stochastic optimization. We refer to the textbook [3].
>
> [3] Lan. First-order and stochastic optimization methods for machine learning, 2020.
>
> > Choice of $M_y$
>
> Sorry, the choice of $M_y$ was given in our proof of Thm 4.2, we will add it to main text in the final version.

---

### Official Review · Reviewer_EKoh · 2025-03-12

**Overall Recommendation:** 4

**Summary:**

The paper proposes smoothed primal-dual algorithms for solving stochastic nonconvex optimization constrained by deterministic or random linear inequality constraints. This is both an important advance in theory, and could be useful practically.

**Claims And Evidence:**

The claims are correct, as far as I can tell.

**Essential References Not Discussed:**

None.

**Experimental Designs Or Analyses:**

There are no experiments described.

**Methods And Evaluation Criteria:**

There are no experiments described.

**Other Comments Or Suggestions:**

N/A

**Other Strengths And Weaknesses:**

- l. 42 (first paragraph of intro) should mention the assumption that X is a set easy to project on. Currently, it gives a misleading impression of handling all linear inequality constraints.
- on penalty methods: other penalty strategies (eg, Pen(x) = f(x) + \rho \| Ax-b \|_2) do not require $\rho \to \infty$. This part is unclear.
- Th. 3.1 gives a complexity result for all iterates (z_t). In view of the proximal point interpretation of the method, one would think that the last iterate has better properties. Is that so?
- l. 117 on the right: “...and lambda is selected accordingly”, it is a useful property of Moreau envelope? This part feels weird in the sentence.
- l. 123: missing parenthesis
- l. 143 on the left: “When we have the problem” is not followed by a clause.
- l. 160: x -> y
- l. 170 on the left: the full stop after -/infty.
- l. 182 on the right: “outputs a points”
- l. 194: augmented Lagrangian L_ρ(x_t, y_t+1, z_t) is mentioned, while in the rest of the paper, there are only two arguments used. The three-argument version (with z_t) does not make sense, because the "smoothing operation" is performed "outside" L_ρ.
- l. 217 on the right: “This post processing step… ...do not affect”
- l. 237 on the left: “Moreau envelope are critical”
- l. 254 on the right: “This which requires…”
- l. 295 on the right: “...decreases, in expectation,u p to an error term depends \tau^2 and the variance”
- l. 397 on the left: “the best-known rate have been”
- l. 422 on the left: “which is obtained with either double loop algorithms” not followed by “or…”
- l. 435 on the left: “Since these work focuses”
- l. 557: should be "u \in X"

Minor things:

- l. 143 on the left: “When we have the problem” is not followed by a clause.
- l. 117 on the right: “...and lambda is selected accordingly”, it is a useful property of Moreau envelope? This part feels weird in the sentence.
- l. 170 on the left: the full stop after -/infty.
- l. 182 on the right: “outputs a points”
- l. 217 on the right: “This post processing step… ...do not affect”
- l. 237 on the left: “Moreau envelope are critical”
- l. 254 on the right: “This which requires…”
- l. 295 on the right: “...decreases, in expectation,u p to an error term depends \tau^2 and the variance”
- l. 397 on the left: “the best-known rate have been”
- l. 422 on the left: “which is obtained with either double loop algorithms” not followed by “or…”
- l. 435 on the left: “Since these work focuses”

**Questions For Authors:**

N/A

**Relation To Broader Scientific Literature:**

The literature is fairly presented.

**Theoretical Claims:**

The proofs are correct, as far as I can tell.

---

> ### Author Rebuttal · Authors · 2025-04-01
>
> We thank the reviewer for their careful review and their detailed questions. We really appreciate their effort in evaluating our work.
>
> ---
>
> >(first paragraph of intro) should mention the assumption that $X$ is a set easy to project on. Currently, it gives a misleading impression of handling all linear inequality constraints.
>
> This is a good point! We first wish to emphasize that assuming $X$ is easy to project is without loss of generality.
> Suppose that $X=\{x\mid Hx\le h\}$ is difficult to project. We can add slack variables to reformulate this problem so that we can handle it by projecting on simple set $\{ x: x\geq 0\}$ and multiplications by $H, H^T$. Specifically, we can rewrite the original optimization problem as
> $$
> \min_{x, s} f(x): Ax = b, Hx + s = h, s \geq 0
> $$
>
> where $s$ is the slack variable. Then the set defined by inequality constraints  $\{(x,s)\mid s\ge 0\}$ is easy to project on and we can dualize the equality constraints $Ax=b,Hx+s=h$ in the Lagrangian function and use only multiplications with matrices $A, A^T, H, H^T$.
> In summary, for problems with linear equality and inequality constraints, we can assume $X$ is easy to project without loss of generality. We also refer to our response to Reviewer srvF.
>
> > Th. 3.1 gives a complexity result for all iterates ($z_t$). In view of the proximal point interpretation of the method, one would think that the last iterate has better properties. Is that so?
>
> This is a good question. However, in nonconvex stochastic optimization, the last iterate generally may not have better properties, and typically we need to sample a point from among all the iterates. For example, in [1]  (Algorithm 1 and 2), the authors also need to  randomly draw output x from $(x_t)^T_{t=1}$ at uniform. There are more examples, please see Section 6, in textbook [2]. In summary, in the stochastic case, we believe that it is hard to use the proximal point interpretation of ALM to improve the last iterate guarantees, because even without linear constraints, we are not aware of last iterate guarantees for single-loop algorithms for nonconvex stochastic optimization.
>
> [1] Lin, Jin, Jordan. On Gradient Descent Ascent for Nonconvex-Concave Minimax Problems, ICML 2020.
>
> [2] Lan. First-order and stochastic optimization methods for machine learning. Springer, 2020.
>
> > On penalty methods: other penalty strategies (e.g., $Pen(x) = f(x) + \rho \| Ax - b \|_2$) do not require $\rho \to \infty$. This part is unclear.
>
> Indeed, in principle, there are papers about exact nonsmooth penalty methods, however we are not aware of complexity results with such a method for our stochastic problem. If the reviewer knows of such a reference, we would be happy to include it. We are taking advantage of the dual variable updates to avoid large penalty parameters. Our focus on ALM instead of nonsmooth penalty methods is because ALM has been a popular method in practice and our aim is to improve its theoretical understanding. The importance of analyzing ALM-based algorithms is also emphasized in the work of Alacaoglu&Wright, 2024 and in fact, we solve an open question stated in the Section 5 of this work.
>
> > Line 117 on the right: “...and lambda is selected accordingly”, is it a useful property of the Moreau envelope? This part feels weird in the sentence.
>
> Because in line 117, we only assume $f$ is weakly convex, we must select $\lambda$ accordingly to ensure the Moreau envelope is also smooth. For example, if $f$ is $\rho$-weakly convex, then we need $\lambda < \frac{1}{\rho}$. We will revisit the sentence
>
> > Line 194: augmented Lagrangian $L_\rho(x_t, y_{t+1}, z_t)$ is mentioned, while in the rest of the paper, only two arguments are used.
>
> Thank you for pointing out this typo, which we will fix. Please see the response to reviewer WK98.
>
> Finally, we wish to thank you so much for your careful reading and suggestions about the phrasing at many places, we will carefully incorporate your suggestions in our final version.

---

### Official Review · Reviewer_srvF · 2025-03-13

**Overall Recommendation:** 2

**Summary:**

The authors introduce smoothed primal-dual algorithms for solving stochastic nonconvex optimization problems with linear inequality constraints. Their approach builds on an inexact gradient descent framework for the Moreau envelope, where the gradient is approximated using a single step of a stochastic primal-dual augmented Lagrangian algorithm. They also establish the $\mathcal{O}(\epsilon^{-4})$ sample complexity guarantee for their algorithms and provide extensions to stochastic linear constraints.

**Claims And Evidence:**

I guess the authors should avoid stating that "the algorithm is free of large batch sizes" in the abstract and elsewhere, as the post-processing step requires a batch size of $\mathcal{O}(\epsilon^{-2})$.

**Essential References Not Discussed:**

No.

**Experimental Designs Or Analyses:**

There are no experiments included. It is essential to add some to empirically validate the performance of the proposed algorithm.

**Methods And Evaluation Criteria:**

This paper primarily focuses on the convergence analysis of the proposed algorithm, without presenting any experiments. While I understand the overall idea, certain aspects remain unclear, as noted in my previous comments.

**Other Comments Or Suggestions:**

It could be better to have a table summarizing all the related algorithms, their convergence guarantees, and their complexities.

**Other Strengths And Weaknesses:**

See the summary and theoretical claims sections.

**Questions For Authors:**

Please answer my questions from other sections.

**Relation To Broader Scientific Literature:**

I do not see much connection to the broader scientific literature, as this paper focuses solely on algorithm analysis. However, the proposed algorithm has the potential to be applied to other machine learning problems.

**Theoretical Claims:**

1. The title mentions linear inequality constraints. However, in (1), the constraint is given as $Ax = b$, which represents a linear equality constraint. Then, in line 145, the constraint changes to $Ax \leq b$. This inconsistency is confusing—what is the main problem the authors aim to solve?

2. In Assumption 1.1.2, the set is defined as $X = {Hx \leq h}$. If the studied problem involves linear inequality constraints, why is $X$ treated separately from $Ax = b$?

3. The paper presents two definitions of $\epsilon$-stationary points—one in line 140 and another in (3). Are these definitions equivalent?

4. What does $L_{\rho}(x, y, z)$ represent in line 194? In line 98, $L_{\rho}$ only involves $x$ and $y$.

---

> ### Author Rebuttal · Authors · 2025-04-01
>
> We thank the reviewer for their careful review and their detailed questions. We really appreciate their effort in evaluating our work.
>
> > The reviewer asks to avoid "the algorithm is free of large batch sizes" in the abstract and elsewhere, as the post-processing step requires a batch size $\mathcal{O}(\varepsilon^{-2})$.
>
> We respond to this question in two parts.
>
> First, in our Theorem 3.1, we can get the $\varepsilon$-near stationary point without the post-processing step, hence this result is completely free of large batch sizes.
>
> Second, for the result in Corollary 3.2, we use the post-processing step **only once** to get $\epsilon$-stationary point (we refer to our answer below for the difference between stationarity and near-stationarity), whereas other works that use large batch sizes require it in **every iteration**. For example, the work [I] not only considers linear equalities but also uses increasing mini-batch sizes at every iteration (see Remark 2 of [I]). Moreover, such a post-processing step is standard and has been used in earlier works, for example, [II, Appendix H] that also use single-sample algorithms at every iteration.
>
> [I] Huang, Chen, and Huang. "Faster stochastic alternating direction method of multipliers for nonconvex optimization." ICML 2019.
>
> [II] Alacaoglu and Lyu. "Convergence of first-order methods for constrained nonconvex optimization with dependent data." ICML 2023.
>
> As a result, we still argue that our method has a much more benign batch-size requirement: we use a large-batch size for **only one** iteration, and only for the optional post-processing step. Hence, our algorithm uses a single sample at every iteration. We will clarify this in our final revision.
>
> > "The title mentions linear inequality constraints, ... This inconsistency is confusing—what is the main problem the authors aim to solve?"
>
> Sorry for the confusion!
>
> We focus on the problem with the linear inequality constraint $Ax\leq b$. Let us see why it is a special case of our problem formulation. Let us start from
> $$
> \min_x f(x) \quad \text{s.t.} \quad Ax \leq b.
> $$
>
> Specifically, we introduce a slack variable $s$, set $Ax - b = -s$, and $s \geq 0$, and write the equivalent problem as:
>
> $$
> \min_{x\in\mathbb{R}^n, s\geq 0} f(x) \quad \text{s.t.}\quad [A \; I]\begin{bmatrix} x \\ s \end{bmatrix} = b.
> $$
>
> Setting
> $$\begin{equation*}
> z=\binom{x}{s}, ~~Z = \mathbb{R}^n \times \mathbb{R}^m_+, ~~B=\begin{bmatrix} A & I \end{bmatrix}, ~~~ g(z) = f(x)
> \end{equation*}$$
> then we have
> $$\begin{equation*}
> \min_{z\in Z} g(z)\colon Bz=b,
> \end{equation*}$$
>
> which is precisely the problem in Eq (1) where $Z$ is indeed polyhedral, which is required by $X$ in Eq (1). We will clarify this in the final version.
>
> > In Assumption 1.1.2, the set is defined as $X = \{Hx \leq h\}$. If the studied problem involves linear inequality constraints, why is $X$ treated separately from $Ax = b$?
>
> Here, without loss of generality, we can always assume $X$ to be easy to project (if not, we can always use slack variables to reformulate the problem and require projecting on simple sets such as $\{x: x\geq 0\}$), please see also our response to Reviewr 1Nyx. This means each of our projections have closed-form solution. And as explained in our response above, we handle the constraint $Ax \leq b$ by reformulating it into equality constraints with slack variables.
>
> The technical reason we treat them separately is because we would like to ensure that dual variable remains unconstrained (Note that writing the Lagrangian with an inequality constraint will require the dual variable to have a sign).
>
> > The paper presents two definitions of $\varepsilon$-stationary points—one in line 140 and another in (3). Are these definitions equivalent?
>
>  These are different notions, as emphasized in our text. Line 140 defines an **$\varepsilon$-stationary** point, while Eq. (3) defines an $\varepsilon$-**near stationary** point. Intuitively, near stationarity indicates proximity to a stationary point, making it slightly weaker than stationarity. Our results show near-stationarity without the post-processing step, whereas stationarity requires post-processing. These two definitions are commonly used in the literature.,such as Davis & Drusvyatskiy (2019). We will clarify this further in our final version.
>
> > What does $L_\rho(x,y,z)$ represent in line 194? In line 98, it only involves $x$ and $y$.
>
>  Thank you for pointing out this typo, which we will fix. Please see our response to Reviewer WK98.
>
> > It is essential to add some to empirically validate the performance of the proposed algorithm.
>
> We ran some experiments to validate our theory. Please see our response to Reviewer WK98, and we will include the experimental results in the final version.
>
> > It could be better to have a table.
>
> Please see the table in our response to Reviewer WK98. We will include the table in our final version.

---

### Official Review · Reviewer_WK98 · 2025-03-17

**Overall Recommendation:** 3

**Summary:**

This work proposed stochastic algorithms for solving nonconvex optimization problems with linear inequality constraints. The main idea is to treat the original nonconvex constrained problem as a nonsmooth optimization problem, and solve it by leveraging the Moreau envelope smoothing technique.

**Claims And Evidence:**

The main claim of this paper is that the proposed method can achieve a complexity of $O(\epsilon^{-4})$ in terms of solving nonconvex optimization with linear inequality constraints. This claim is supported by the analysis.

**Essential References Not Discussed:**

There is a previous work studied the nonsmooth min-max problem and proposed a very similar algorithm. Although the ultimate goals are different, as they do not consider any constraints, I believe it still worth to discuss due to the high similarity in the method design.

Quanqi Hu, Qi Qi, Zhaosong Lu, and Tianbao Yang. Single-loop stochastic algorithms for difference of max-structured weakly convex functions. In Advances in Neural Information Processing Systems 37: Annual Conference on Neural Information Processing Systems 2024, NeurIPS 2024, 2024.

**Experimental Designs Or Analyses:**

The paper has no experiment.

**Methods And Evaluation Criteria:**

The idea behind the proposed method makes sense.

**Other Comments Or Suggestions:**

As someone who is not familiar with the constrained optimization literature, I find the motivation to be vague while reading the first section, especially when many details on existing works are deferred to the last section. It would be a lot clearer if there is a table summarizing existing and the proposed methods and highlighting the main contributions of this work.

Typo: in line 123, the right column, a ')' is missing in the gradient of Moreau envelope.

**Other Strengths And Weaknesses:**

The method design and analysis seem interesting and solid to me, but the lack of experiment is the main weakness.

**Questions For Authors:**

In line 194 (left column), the definition of the function $L_\rho (x,y_{t+1},z_t)$ seems missing. Should it be $L_\rho (z_t,y_{t+1})$ instead? The same issue exists in Algorithm 1 and Algorithm 2.

**Relation To Broader Scientific Literature:**

The main contribution of this work is to show a way of using the Moreau envelope technique in solving the family of constrained optimization problems. Similar strategy may apply to other setting.

**Theoretical Claims:**

I briefly checked all the proofs in the appendix, and I do not see any major issues.

---

> ### Author Rebuttal · Authors · 2025-04-01
>
> We thank the reviewer for their careful review and their detailed questions. We really appreciate their effort in evaluating our work.
>
> > The reviewer suggests discussing the following paper by Hu et al, 2024.
>
> Thank you for pointing out this relevant work! We agree that this paper focuses on a different problem with different goals. We will include a discussion of the algorithm design in this paper and its similarity to our work, in the final version of our paper.
>
> > It would be a lot clearer if there is a table summarizing existing and the proposed methods and highlighting the main contributions of this work.
>
> We will add this table to our paper. We also include it here for reference. For further complexity comparisons (and clarifying how we handle $Ax\leq b$), we refer to our response to Reviewer 1NyX. For the importance of showing guarantees for ALM instead of penalty methods, we also refer to [1]. We solve an open question stated in Section 5 of [1].
>
> $$
> \text{Objective: } \mathbb{E}_\xi [f(x,\xi)]
> $$
>
> | **Reference** | **Constraint**                                                                         | ** Oracle** | **Complexity**                              | **Loops** | Method  |
> | ------------- | -------------------------------------------------------------------------------------- | ----------- | ------------------------------------------- | --------- | ------- |
> | [1]           | $A x = b$                                                                              | ①           | $\widetilde{\mathcal{O}}(\varepsilon^{-3})$ | 1         | ALM     |
> | [1]           | $\mathbb{E}[c(x,\zeta)] = 0$ and $x \in X$ where $X$ is easy to project on  | ①           | $\widetilde{\mathcal{O}}(\varepsilon^{-5})$ | 1         | Penalty |
> | [2]           | $c(x)=0$  and $x \in X$ where $X$ is easy to project on                                                                              | ①           | $\mathcal{O}(\varepsilon^{-3})$             | 1         | Penalty |
> | [3]           | $\mathbb{E}[c(x,\zeta)] = 0$  and $x \in X$ where $X$ is easy to project on                                                           | ①           | $\mathcal{O}(\varepsilon^{-5})$             | 2         | Penalty |
> | This work     | $A x \leq b$                                    | ②           | $\mathcal{O}(\varepsilon^{-4})$             | 1         | ALM     |
> | This work     | $\mathbb{E}_\zeta[A(\zeta)x - b(\zeta)] \leq 0$ | ②           | $\mathcal{O}(\varepsilon^{-4})$             | 1         | ALM     |
>
> Oracle ①: for a given random seed $\xi$, we need to sample $\nabla f(x, \xi)$ and $\nabla f(y, \xi)$, satisfying:
>
> $$\mathbb{E}_\xi[\nabla f(z,\xi)]=\nabla f(z)$$
>
> and
>
> $$\mathbb{E}_\xi\|\nabla f(z,\xi)-\nabla f(z)\|^2\leq \sigma^2$$
>
> for $z=x$ and $z=y$ and,
>
>  $$\mathbb{E}_{\xi}\|\nabla f(x,\xi)-\nabla f(y,\xi)\|^2\leq L^2\|x-y\|^2$$
>
> Oracle ②: we only evaluate the stochastic gradient at a single point $x$ for seed $\xi$, satisfying
>
>  $$\mathbb{E}_\xi[\nabla f(x,\xi)]=\nabla f(x)$$
>
> and
>
> $$ \mathbb{E}_\xi\|\nabla f(x,\xi)-\nabla f(x)\|^2\leq \sigma^2$$
>
> [1] Alacaoglu, and Wright. “Complexity of Single Loop Algorithms for Nonlinear Programming with Stochastic Objective and Constraints.” AISTATS 2024
> [2] Lu, Mei, Xiao. Variance-reduced first-order methods for deterministically constrained stochastic nonconvex optimization with strong convergence guarantees arXiv, 2024.
> [3] Li et al. Stochastic inexact augmented Lagrangian method for nonconvex expectation constrained optimization. COAP 2024
>
> > Typo: in line 123, the right column, a ')' is missing
>
> Thank you! We will fix the typo.
>
> > In line 194 (left column), the definition of the function $L_\rho(x, y_{t+1}, z_t)$ seems missing. Should it be $L_\rho(z_t, y_{t+1})$ instead?
>
> It is a typo. In line 194, $L_\rho(x, y_{t+1}, z_t)$ should be $L_\rho(x, y_{t+1})$ (which is defined in line 97). And in Algorithm 1 and 2 (line 224 and 338), $L_\rho(x_t, y_{t+1}, z_t)$ should be $L_\rho(x_t, y_{t+1})$.
>
> > The method design and analysis seem interesting and solid to me, but the lack of experiment is the main weakness.
>
> We ran a preliminary experiment to validate our theory. This result also shows that our algorithm converges faster than [I]. We consider the stochastic quadratic programming problem:
>
> $$\min_{x} \mathbb{E} \frac{1}{2} x^T Q(\zeta) x + r(\zeta)^T x: \mathbb{E}  [A(\xi) x - b(\xi)] = 0, \ell_i \leq x_i \leq u_i$$
>
> where $x \in \mathbb{R}^{n}$, $Q(\zeta) \in \mathbb{R}^{n \times n}$, $r(\zeta) \in \mathbb{R}^n$, $A(\xi) \in \mathbb{R}^{m \times n}$, and $b(\xi) \in \mathbb{R}^m$.
>
> | **Iteration** | ALM(ours) | **Penalty method [I]** |
> | ------------: | --------: | ---------------------: |
> |           1e5 |    0.5036 |                 0.7050 |
> |           1e6 |    0.0454 |                 0.5017 |
> |           2e6 |    0.0219 |                 0.2656 |
> |           5e6 |    0.0152 |                 0.2025 |
> |           1e7 |    0.0082 |                 0.1755 |

---

### Decision · Program_Chairs · 2025-05-01

**Decision:**

Accept (spotlight poster)

**Comment:**

This paper proposes single-loop smoothed primal-dual algorithms for solving stochastic nonconvex optimization problems with linear inequality constraints. The proposed method achieves a convergence rate of $1/\epsilon^4$, and the analysis extends to cases with stochastic linear constraints.

Strengths:

The paper makes a solid theoretical contribution to the literature on constrained nonconvex optimization. The convergence analysis is technically sound, with all reviewers generally agreeing that the proofs are correct and the complexity result matches the worst-case lower bound under the assumption in the paper.

Weaknesses:

The lack of empirical validation is a major shortcoming noted across all reviews. In the AC's opinion, this is understandable given that the contribution of this paper clearly focuses more on the theoretical complexity analysis. The main theoretical limitation is the requirement that $X$ must be a polyhedron. We hope this can be relax in the future work.

If this work is accepted. Please include the following items in the revision:

1. Reviewers suggested adding a summary table comparing existing methods and highlighting what is new in this paper. It seems the authors have such a table in the rebuttal to begin with.
2. A review suggest citing a closely related work (Hu et al., NeurIPS 2024) that employs similar techniques in different problem settings.
3. Show how STORM can be used get the $1/\epsilon^3$ complexity under the stronger assumption.
4. Add more discussion on Lu et al, for example, like comments in your reply. It should be as important as Alacaoglu&Wright in your literature review, but it is not discussed as much in the current version.
5. Somewhere in the paper, refer readers to Li, Chen, Liu, Lu, Xu (https://arxiv.org/abs/2007.01284) for the corresponding between the
$\epsilon$-stationary point inequality constrained problem and the equality-constrained problem after adding slack variable. If you have space to explain in details in your paper, that will be even better.

AC